# Sobolev Regularized Score Difference Estimation in Diffusion Models

**Chenghan Xie** [1]   **Jose Blanchet** [1]   **Renyuan Xu** [1]

## Abstract

Estimating the difference of two Stein's score functions is a fundamental problem in generative modeling. In particular, score differences arise naturally in transfer learning, where the score difference provides the mechanism for adapting a pre-trained model to a new target distribution, and in diffusion model-based post-training methods such as discriminator guidance. Existing estimators for score differences in these settings either lack of statistical consistency or are difficult to scale up in high-dimensions. We propose a statistically consistent and scalable estimator for score differences based on Sobolev regularization, which plays a crucial role in ensuring consistency and stablizing the training in the small-sample regime. Mathematically, we establish a convergence rate of $\tilde{\mathcal{O}}(n^{-\frac{s-1}{d+2s-2}})$ where $d$ is the dimension and $s$ denotes the smoothness of the underlying densities, and provide a minimax lower bound of $\tilde{\Omega}(n^{-\frac{2(s-1)}{d+2s}})$ (in mean-squared error). Empirically, our estimator exhibits significantly improved stability in small-sample regimes compared to existing methods. We demonstrate its effectiveness on real-world tasks, including transfer learning for ECG signal generation, where it substantially outperforms non-regularized score difference estimators in downstream classification performance.

## 1. Introduction

The difference of Stein's score functions is defined as the gradient of the log-density ratio $\nabla \log q(\cdot) - \nabla \log p(\cdot)$ between a target distribution $q$ and a source distribution $p$. Conceptually, this difference represents the driving force required to transport samples from a source distribution $p$

to a target distribution $q$. As a result, the score difference emerges as a fundamental primitive in transfer learning for modern generative modeling (Liu et al., 2023; Ouyang et al., 2024; Wang et al., 2024).

This estimation problem is central in post-training of diffusion models, which adapts pre-trained generative models to align with human preferences, structural constraints, or downstream tasks. A wide range of approaches have been proposed, including RLHF (Black et al., 2023; Fan et al., 2023), stochastic control–based formulations (Tang & Zhou, 2024; Han et al., 2024b; Uehara et al., 2024), and classifier-guided or conditioning-based methods. Notably, most of these can be unified as add-on mechanisms to pre-trained dynamics:

$$\mathrm{d}Y_t = s_\theta(t, Y_t)\mathrm{d}t - h_\eta(t, Y_t)\mathrm{d}t + \sigma(t)\mathrm{d}W_t, \quad (1)$$

where $s_\theta(t, \cdot)$ is the pre-trained score function approximating $\nabla \log p_t(\cdot)$, and $h_\eta(t, \cdot)$ is an additive control term. In frameworks such as discriminator guidance and conditional generation, via Doob's $h$-transform, the term $h_\eta(t, \cdot)$ approximates $\nabla \log q_t(\cdot)$, where $q_t$ is a target distribution carrying information from constraints, classifiers, or preference signals (Denker et al., 2024; Du et al., 2024; Pidstrigach et al., 2025; Howard et al., 2025). In diffusion models, post-training is a form of transfer learning, transferring pre-training knowledge to new data-generation tasks. For simplicity, we use "transfer learning" to refer to this broader setting throughout the paper.

In transfer learning, the target task typically has far fewer samples than the source task, rendering direct estimation of the target score $\nabla q_t$ unstable. Fortunately, because the target and source tasks are closely related, the density ratio $\frac{q_t}{p_t}$ often exhibits exploitable structure. This motivates directly estimating the density ratio and its gradient, since

$$\nabla \log q_t(\cdot) - \nabla \log p_t(\cdot) = \frac{\nabla (q_t/p_t)(\cdot)}{q_t/p_t(\cdot)}. \quad (2)$$

Unlike estimating and differencing the two scores separately, this approach exploits shared geometry and is more structure-aware and data-efficient in low-sample regimes (see Figure 1).

Despite arising naturally in many settings, obtaining a reliable estimator for this score difference remains challenging.

Code is available at the GitHub repo. [1]Department of Management Science and Engineering, Stanford University. Correspondence to: Renyuan Xu <renyuanxu@stanford.edu>.

Proceedings of the 43$^{rd}$ International Conference on Machine Learning, Seoul, South Korea. PMLR 306, 2026. Copyright 2026 by the author(s).

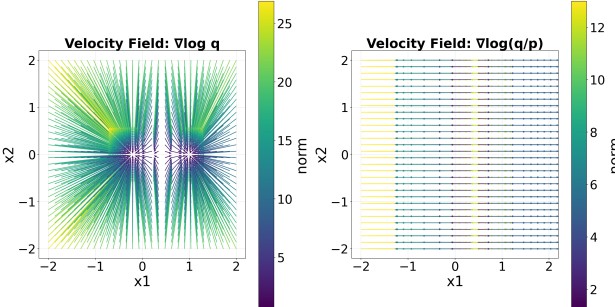

Figure 1. **Visualization of the score functions**. The score difference field (right) is markedly smoother and less variable than the absolute target score (left), demonstrating the structural benefit of directly estimating the score difference.

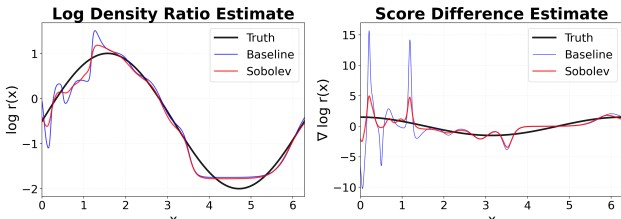

Figure 2. **Impact of high-frequency noise.** Comparison of $\log r$ (left) and $\nabla \log r$ (right) estimates. Standard classification (blue) captures the function value but fails on the gradient due to overfitting. Our Sobolev-regularized method (red) enforces smoothness, recovering the gradient accurately by filtering out noise.

The most prevalent large-scale approaches (Ouyang et al., 2024; Kim et al., 2022) adopt a "classify-then-differentiate" pipeline: first estimating the log-density ratio via binary classification, and then differentiating the fitted model. However, this approach lacks theoretical guarantees on gradient convergence. As illustrated in Figure 2, standard classifiers tend to overfit high-frequency but low-amplitude noise components in the data. This overfitting yields small ratio errors (left) but catastrophic oscillations after differentiation (right).

We address this challenge via Sobolev regularization, which controls the smoothness of the log-density ratio gradient and prevents overfitting to spurious high-frequency noise. With $n$ samples from both the source $p$ and target $q$, we prove a convergence rate of $\tilde{\mathcal{O}}(n^{-\frac{s-1}{d+2s-2}})$ for the score difference error (Theorem 3.1, 5.2), where $d$ is the dimension and $s$ is the smoothness parameter. We also provide a minimax lower bound of $\tilde{\Omega}(n^{-\frac{2(s-1)}{d+2s}})$ (Theorem 4.1), showing near-optimality. Experiments verify these findings across diverse tasks, including joint domain adaptation via Wasserstein gradient flow (Sec 6.2.1) and transfer learning for diffusion models (Sec 6.2.2), confirming that Sobolev regularization yields stable and consistent score difference estimators.

**Closely related literature.** Our work is closely related to two lines of literature: score difference estimation and Sobolev regularization in machine learning.

Score difference estimation underlies a broad class of applications related to diffusion models. For example, (Ouyang et al., 2024) propose using score difference estimation for transfer learning tasks in diffusion models, which is later refined by (Wang et al., 2024) with an additional residual fine-tuning step. For post-training problems, the discriminator guidance framework of (Kim et al., 2022) highlights the central role of score difference estimation. In this approach, a discriminator is trained to distinguish samples from the target distribution and the pre-trained model, and its gradient provides an estimate of the score difference $\nabla \log q_t - \nabla \log p_t$, which is then injected as a control term into the diffusion dynamics. This framework was further extended to noisy data settings in (Cong et al., 2025). However, the above papers primarily emphasize methodological development and empirical validation, while theoretical convergence guarantees remain unestablished. On the other hand, several alternative approaches provide convergence guarantees but may still suffer from practical limitations such as scalability and numerical stability. For example, assuming that the ground-truth score difference lies in a reproducing kernel Hilbert space (RKHS), (Srikanth et al.) derive a closed-form kernel-based solution. However, this approach incurs prohibitive computational costs, as it requires evaluating kernel distances against the entire training dataset at each inference step, rendering it impractical at scale. Similarly, (Liu et al., 2023) propose a consistent score difference estimator for Wasserstein gradient flow (WGF) methods, but their local kernel approach faces the same scalability challenges. More recently, (Verine et al., 2025) address objective bias in discriminator guidance by adopting a score-matching objective in which the regression target is the residual error of a pre-trained score. However, without explicit regularity constraints, the resulting estimator can be dominated by variance, particularly in low-density regions or when the pre-trained score is already relatively accurate. In contrast to these approaches, our Sobolev-regularized score difference estimator retains the computational efficiency of classification-based frameworks, while offering enhanced consistency and robustness via the introduction of a Sobolev regularization term.

Highlighting the central role of regularization, (Husain & Nock) point out that discriminators in guidance frameworks must be well regularized to ensure generalization, a perspective that closely aligns with our work. Sobolev-type regularization, often realized through gradient penalties, has been widely studied in the literature on generative adversarial networks (GANs) (Lin et al., 2025; Roth et al., 2017; Mescheder et al., 2018), whereas its role in diffusion models remains comparatively underexplored. In these settings, the discriminator can be interpreted as estimating a time-varying density ratio. Moreover, the underlying motivations differ substantially. In those works, gradient penalties

are introduced to promote dynamical stability by shifting the eigenvalues of the Jacobian and preventing oscillations around local Nash equilibria. In contrast, our motivation is rooted in statistical estimation: we employ Sobolev regularization to constrain the hypothesis space, ensuring that the estimated score difference remains accurate and robust to overfitting to high-frequency noise. Finally, beyond generative modeling, (Ding et al., 2025) study Sobolev-penalized deep semi-supervised regression for the joint estimation of a regression function and its gradient, leveraging a large set of unlabeled covariates to approximate the Sobolev penalty.

## 2. Problem setup and proposed method

Our goal is to formalize transfer learning from a source $p$ to a target $q$, by estimating the score difference in a way that is structure-aware, data-efficient, and stable.

To focus on the essential building block of our framework, we first consider a bounded state space. The extension to unbounded domains and the inclusion of a time component, tailored to diffusion models, are deferred to Section 5, where the generalization is straightforward.

Let $[0,1]^d \subseteq \Omega \subseteq \mathbb{R}^d$, $d \in \mathbb{N}$ be a bounded and connected domain with sufficiently smooth boundary $\partial\Omega$. Let $P, Q \in \mathcal{P}(\Omega)$ be absolute continuous with respect to Lebesgue measure with densities $p, q$. For any $x \in \Omega$ let

$$\rho(x) := \tfrac{1}{2}p(x) + \tfrac{1}{2}q(x), \qquad \mu(\mathrm{d}x) := \rho(x)\,\mathrm{d}x,$$

and define the log-density ratio

$$f^\star(x) := \log \frac{q(x)}{p(x)}. \tag{3}$$

For a multi-index $\alpha = (\alpha_0, \alpha_1, \ldots, \alpha_d) \in \mathbb{N}^{1+d}$, denote $|\alpha| := \alpha_0 + \cdots + \alpha_d$ and $D^\alpha f := \frac{\partial^{|\alpha|} f}{\partial t^{\alpha_0} \partial x_1^{\alpha_1} \ldots \partial x_d^{\alpha_d}}$, interpreted in the weak (distributional) sense. For an integer $s \geq 0$, the weighted Sobolev space $H^s(\mu)$ is given by

$$H^s(\mu) := \left\{ f \in L^2(\mu) \mid \begin{array}{l} D^\alpha f \in L^2(\mu), \text{for all} \\ \text{multi-indices } \alpha \text{ with } |\alpha| \leq s \end{array} \right\}.$$

For the supremum norm, we denote the $L^\infty(\mu)$ norm as the essential supremum with respect to the measure $\mu$:

$$\|f\|_{L^\infty(\mu)} := \inf\{C \geq 0 : \mu(\{x \in \Omega : |f(x)| > C\}) = 0\}.$$

Let $\mu_0$ be the uniform distribution on $\Omega$. We assume the following outstanding assumptions.

**Assumption 2.1** (Density functions). Assume $p$ and $q$ satisfy the following conditions:

1. $p, q \in C^2(\Omega)$ and are strictly positive on $\Omega$.

2. The following Neumann boundary condition holds:

$$\rho(x)\,\partial_n f^\star(x) = 0 \qquad \text{for all } x \in \partial\Omega,$$

where $n(x)$ is the outward unit normal to $\partial\Omega$.

3. Fix $s \geq 2$. We assume $f^\star \in H^s(\mu_0)$.

Given that $p, q$ are positive and $C^2$ on $\Omega$, we have $f^\star \in C^2(\Omega)$. Hence, there exists $M^\star > 0$ such that

$$\max\{\|f^\star\|_{L^\infty(\mu_0)}, \|\nabla f^\star\|_{L^\infty(\mu_0)}\} \leq M^\star.$$

Motivated by the boundedness of the true density ratio $f^\star$, we restrict the function class to ensure stability in the optimization. Specially, fix a constant $M \geq M^\star$, we define a bounded Sobolev subset of $\mathcal{H}^s(\mu_0)$ as follows:

$$\mathcal{H}_M := \left\{ f \in H^1(\mu_0) : \|f\|_{L^\infty(\mu_0)} \leq M \right\}. \tag{4}$$

By construction, this set is closed and convex in $H^1(\mu_0)$, and it contains the ground truth $f^\star$. Since $\mu$ and $\mu_0$ are equivalent under Assumption 2.1, we keep the notation $\mathcal{H}_M$ when risks and norms are evaluated with respect to $\mu$.

**Learning objective.** Our objective is to estimate the score difference

$$\nabla_x f^\star(x) := \nabla_x \log \frac{q(x)}{p(x)},$$

given access only to finite samples from both source and target datasets:

$$\mathcal{S} = \{X_i^p\}_{i=1}^n \overset{\text{i.i.d.}}{\sim} p, \ \mathcal{T} = \{X_i^q\}_{i=1}^n \overset{\text{i.i.d.}}{\sim} q. \tag{5}$$

When $p$ and $q$ share similar geometric structure, the vector field $\nabla \log \frac{q(x)}{p(x)}$ often concentrates on a low-dimensional manifold, making it easier to learn than the full score of $q$. The standard approach first estimates the log-density ratio and then differentiates it directly (Ouyang et al., 2024; Kim et al., 2022). Usually, the estimation of log-density ratio is trained via classification loss.

**Binary classification reformulation.** We introduce a label-augmented mixture model, offering a convenient probabilistic representation for learning the log-density ratio. Specifically, denote by $\mu_{p,q}$ the joint law of $(X, Y)$ with

$$\begin{aligned} &Y \sim \text{Bernoulli}(1/2), \\ &X \mid (Y = 1) \sim q, \ X \mid (Y = 0) \sim p. \end{aligned} \tag{6}$$

Under such setting, the posterior classification probability is, for $x \in \Omega$, $\eta(x) := \mathbb{P}(Y = 1 \mid X = x) = \frac{q(x)}{p(x)+q(x)}$ The corresponding Bayes logit is $\log \frac{\eta(x)}{1-\eta(x)} = \log \frac{q(x)}{p(x)} = f^\star(x)$. Therefore, if we let $\sigma(u) := \frac{1}{1+e^{-u}}$ and define the cross-entropy loss

$$\ell_{\text{CE}}(y, u) := -y \log \sigma(u) - (1-y) \log(1 - \sigma(u)),$$

then the log-density ratio $f^\star$ in (3) minimizes the population cross-entropy risk

$$L_{\mathrm{CE}}(f) := \mathbb{E}_{(X,Y)\sim\mu_{p,q}}\big[\,\ell_{\mathrm{CE}}(Y, f(X))\,\big]$$

among all measurable functions $f : \Omega \to \mathbb{R}$. Using the fixed source–target design in (5), we augment deterministic labels and write $\mathcal{D} := \{(X_i^p, 0)\}_{i=1}^n \cup \{(X_i^q, 1)\}_{i=1}^n$. The corresponding fixed source–target empirical cross-entropy risk is

$$\widehat{L}_{\mathrm{CE},\mathcal{D}}(f) := \frac{1}{2n}\sum_{i=1}^n \ell_{\mathrm{CE}}(0, f(X_i^p)) + \frac{1}{2n}\sum_{i=1}^n \ell_{\mathrm{CE}}(1, f(X_i^q)).$$

**Functional class: Function class: clipped sparse neural networks with bounded gradients.** We consider a sparse neural network class (Schmidt-Hieber, 2020; Suzuki, 2018; Farrell et al., 2021) with the $\mathrm{ReLU}^3$ activation $\eta_3(x) := \max\{x^3, 0\}$, applied componentwise for vector inputs. This activation is useful for representing spline-type approximants and is supported by the corresponding approximation theory (Lu et al., 2021). Fix $M > M^\star$. Let $T_M : \mathbb{R} \to [-M, M]$ be a nondecreasing $C^1$ truncation map such that $|T_M(u)| \le M$ and $|T_M'(u)| \le 1$ for all $u \in \mathbb{R}$, and such that $T_M(u) = u$ whenever $|u| \le M_0$, for some $M_0 \in (M^\star, M)$. Thus $T_M$ clips only outside a range strictly containing the target range.

Given the depth $L$, width $W$, sparsity level $S$, and parameter bound $B$, define

$$\mathcal{F}_M(L, W, S, B) :=$$
$$\Big\{ g(x) = T_M\Big[\big(\mathcal{W}^{(L)}\eta_3(\cdot) + b^{(L)}\big) \circ \cdots \circ \big(\mathcal{W}^{(1)}x + b^{(1)}\big)\Big],$$
$$\mathcal{W}^{(1)} \in \mathbb{R}^{W\times d}, \mathcal{W}^{(2,\ldots,L-1)} \in \mathbb{R}^{W\times W}, \mathcal{W}^{(L)} \in \mathbb{R}^{1\times W},$$
$$\sum_{l=1}^L \Big( \|\mathcal{W}^{(l)}\|_0 + \|b^{(l)}\|_0 \Big) \le S,$$
$$\max_l \|\mathcal{W}^{(l)}\|_\infty \vee \|b^{(l)}\|_\infty \le B,$$
$$\|g\|_{L^\infty(\mu)} \le M, \|\nabla g\|_{L^\infty(\mu)} \le M \Big\}.$$

Here $\circ$ denotes function composition, $\|\cdot\|_0$ counts nonzero entries, and $\|\cdot\|_\infty$ denotes the entrywise maximum norm.

**Proposed learning method.** To control both function values and gradients, we define the Sobolev seminorm

$$\mathcal{R}(f) := \|\nabla f\|_{L^2(\mu)}^2,$$

for any $s \ge 1$ and $f \in H^s(\mu)$. For $\lambda > 0$, the population Sobolev-regularized risk is defined as

$$J_\lambda(f) := L_{\mathrm{CE}}(f) + \lambda\,\mathcal{R}(f).$$

The existence of the unique minimizer $f_\lambda$ is provided in Lemma A.1. We estimate this population penalty using the

same fixed source–target design. Accordingly, define the fixed source–target empirical Sobolev penalizer:

$$\widehat{\mathcal{R}}_{\mathcal{D}}(f) := \frac{1}{2n}\sum_{j=1}^n \|\nabla f(X_j^p)\|_2^2 + \frac{1}{2n}\sum_{j=1}^n \|\nabla f(X_j^q)\|_2^2.$$

The fixed source–target empirical energy functional is

$$\widehat{J}_{\lambda,\mathcal{D}}(f) := \widehat{L}_{\mathrm{CE},\mathcal{D}}(f) + \lambda\widehat{\mathcal{R}}_{\mathcal{D}}(f).$$

Our method proceeds by first estimating the log–density ratio as the minimizer of an empirical energy functional,

$$\widehat{f}_{\lambda,\mathcal{F}} \in \arg\min_{f\in\mathcal{F}} \widehat{J}_{\lambda,\mathcal{D}}(f), \tag{8}$$

and then taking its gradient to obtain the desired score difference. We assume that the minimum is attained in $\mathcal{F}$.

## 3. Upper bound analysis

In this section, we provide a statistical convergence rate for our Soblev regularized estimator, in terms of the number of samples used in optimization, for which the proof is deferred to Appendix A.4.

**Theorem 3.1** (Convergence rate for Sobolev-regularized logistic empirical risk minimization (ERM))**.** *Suppose Assumption 2.1 hold and the smoothness index satisfies $s \le 4$. Consider neural network $\mathcal{F}_M(L, W, S, B)$ with $N_n \asymp n^{\frac{d}{d+2s-2}}$, $L = \mathcal{O}(1)$, $W, S, B \asymp N_n, \lambda \asymp N_n^{-\frac{s-1}{d}}$. Then the soblev-penalized estimator $\widehat{f}_{\lambda,\mathcal{F}}$ in (8) satisfies the following upper bound with probability $1 - 2n^{-2}$:*

$$\|\widehat{f}_{\lambda,\mathcal{F}} - f^\star\|_{H^1(\mu)}^2 \lesssim n^{-\frac{s-1}{d+2s-2}} \log n.$$

*Remark* 3.2. (a). The restriction $s \le 4$ is technical and follows from the cubic quasi-interpolant and $\mathrm{ReLU}^3$ approximation used in the proof, following the construction in (Lu et al., 2021). Higher smoothness can be handled by higher-order quasi-interpolants and $\mathrm{ReLU}^r$ networks with spline order $r + 1 \ge s$.

(b). Since $\mu$ admits a $C^2$ density $\rho$ on $\Omega$, it is equivalent the uniform distribution $\mu_0$. Consequently, the above bound can also be equivalently expressed as

$$\|\widehat{f}_{\lambda,\mathcal{F}} - f^\star\|_{H^1(\mu_0)}^2 \lesssim n^{-\frac{s-1}{d+2s-2}} \log n.$$

(c). Moreover, since $J_\lambda$ is strongly convex over $\mathcal{H}_M$, a local Rademacher complexity argument yields a sharper rate than the $\mathcal{O}(n^{-\frac{s}{d+4s}})$ rate obtained for the Sobolev-regularized estimators in (Ding et al., 2025), where they used global Rademacher complexity for regression problems.

**Proof outline.** The proof proceeds in four steps. First, we establish key analytic properties of the population energy

functional $J_\lambda$, including strong convexity and smoothness (see Lemma 3.3). Second, we quantify the regularization-induced bias at the population level (see Lemma 3.4). Third, we relate the generalized error to the excess energy via localized Rademacher complexity (see Theorem 3.5). Finally, combining these ingredients yields the desired rate.

For any functional $J : H^s(\mu) \to \mathbb{R}$, let $DJ(g)$ be the Fréchet derivative of $J$ at $g$. We have the following result.

**Lemma 3.3** (Strong convexity and smoothness of the penalized risk). *Define* $c_{\min} := \frac{1}{4\cosh^2(M/2)}$. *Then the penalized risk* $J_\lambda(f) := L_{\mathrm{CE}}(f) + \lambda\mathcal{R}(f)$ *is Fréchet differentiable on* $\mathcal{H}_M$ *and satisfies, for all* $f, g \in \mathcal{H}_M$,

$$
\begin{aligned}
& J_\lambda(f) - J_\lambda(g) - DJ_\lambda(g)[f - g] \\
\geq\ & \frac{c_{\min}}{2}\|f - g\|_{L^2(\mu)}^2 + \lambda\|\nabla(f - g)\|_{L^2(\mu)}^2, \quad (9)
\end{aligned}
$$

*and*

$$
\begin{aligned}
& J_\lambda(f) - J_\lambda(g) - DJ_\lambda(g)[f - g] \\
\leq\ & \|f - g\|_{L^2(\mu)}^2 + \lambda\|\nabla(f - g)\|_{L^2(\mu)}^2. \quad (10)
\end{aligned}
$$

Note that the parameter $M > 0$ used in $c_{\min}$ is the uniform logit bound, i.e. $|f(x)| \leq M$ for all $f \in \mathcal{F} \cup f^\star$ and $\mu$-a.e. $x$ (see the details in Proposition A.2).

Our argument follows the general outline of (Ding et al., 2025), with an important modification: because the cross-entropy loss fails to be globally strongly convex, uniqueness cannot be obtained directly. Instead, we establish conditional strong convexity restricted to $\mathcal{H}_M$. Our proof proceeds by (i) deriving strictly positive pointwise curvature bounds for the logistic loss within $[-M, M]$, (ii) lifting the curvature bounds to the functional space via Taylor expansions, and (iii) combining the bounds with the quadratic structure of the Sobolev regularizer.

Denote $\Delta$ as the Laplacian operator $\Delta f(x) := \sum_{j=1}^d \partial_{x_j x_j} f(x)$, interpreted in the weak sense for $f \in H^1(\mu)$. Define the weighted elliptic operator $\mathcal{K}h := \Delta h + \nabla h \cdot \nabla\log\mu$. Then $\mathcal{K}f^\star \in L^2(\mu)$, and define

$$
\beta := 64\,\|\mathcal{K}f^\star\|_{L^2(\mu)}^2\,\cosh^4\!\left(\tfrac{M}{2}\right).
$$

Next we quantify the regularization-induced bias.

**Lemma 3.4** (Bias of the population Sobolev solution). *For all* $\lambda > 0$, *the unique minimizer* $f_\lambda$ *of* $J_\lambda$ *satisfies*

$$
\|f_\lambda - f^\star\|_{L^2(\mu)}^2 \leq \beta\,\lambda^2, \quad \|\nabla(f_\lambda - f^\star)\|_{L^2(\mu)}^2 \leq \beta\,\lambda.
$$

The proof relies on testing the Euler-Lagrange (EL) equation with error $\delta_\lambda := f_\lambda - f^\star$. A crucial insight is to use Green's identity to relate the regularization part in EL equation to the Laplacian of $f^\star$, which controls the bias magnitude.

**Theorem 3.5** (Shifted oracle inequality for the clipped sieve). *Fix* $0 < \lambda < 1$. *Let* $\mathcal{F} := \mathcal{F}_M(L, W, S, B)$ *be the clipped and gradient-bounded neural-network sieve, where* $L = \mathcal{O}(1)$, $W = \mathcal{O}(N)$, $S = \mathcal{O}(N)$, *and* $B = \mathcal{O}(N)$. *Let* $f_0 \in \mathcal{F}$ *be any fixed comparator independent of the data, and* $f_\lambda$ *as the population minimizer of* $J_\lambda$. *Let*

$$
\widehat{f}_{\lambda,\mathcal{F}} \in \arg\min_{f\in\mathcal{F}} \widehat{J}_{\lambda,\mathcal{D}}(f).
$$

*Then for any* $t > 0$, *with probability at least* $1 - e^{-t}$,

$$
J_\lambda(\widehat{f}_{\lambda,\mathcal{F}}) - J_\lambda(f_\lambda) \lesssim J_\lambda(f_0) - J_\lambda(f_\lambda) + r^\star + \frac{t}{n}, \quad (11)
$$

*where* $r^\star$ *is the critical radius of the sub-root function*

$$
\phi(r) := C_0\left[\frac{1}{n} + \sqrt{\frac{S\,3^L r}{n}\log(BWn)}\right].
$$

*Here* $C_0$ *depends only on the fixed envelope* $M$ *and* $c_{\min}^{-1}$, *but not on* $n, N, W, S, B$.

The proof is deferred to Appendix A.3. The key property facilitating our fast rate analysis is the structural constraint of the hypothesis space $\mathcal{H}_M$. Since optimization is performed over a bounded, convex domain, the regularized loss $J_\lambda$ is strongly convex and the estimator remains bounded.

Strong convexity and boundedness ensure that the variance of the error diminishes as the estimator approaches the optimum. This motivates the use of localized, rather than global, complexity measures. In the proof, we formalize this intuition using the framework of (Lu et al., 2021).

## 4. Minimax lower bound

This section provides a minimax lower bound for the ratio estimation, for which the proof is deferred to Appendix B.

Let $\mathcal{C}_{\mathrm{pair}}$ be the class of pairs $(p, q)$ satisfying Assumption 2.1, and let $\mathbb{E}_{p,q}$ denote expectation under the fixed source–target design $X_i^p \overset{\text{i.i.d.}}{\sim} p$, $X_i^q \overset{\text{i.i.d.}}{\sim} q$.

**Theorem 4.1** (Minimax lower bound for score difference estimation). *For any estimator* $\psi : (\mathbb{R}^d)^n \times (\mathbb{R}^d)^n \to H^1(\mu_0)$, *we have the minimax lower bound*

$$
\begin{aligned}
&\inf_\psi \sup_{(p,q)\in\mathcal{C}_{\mathrm{pair}}} \mathbb{E}_{p,q}\left[\big\|\psi(\mathcal{S}, \mathcal{T}) - f_{p,q}\big\|_{H^1(\mu_0)}^2\right] \\
&\gtrsim\ n^{-2(s-1)/(2s+d)}. \quad (12)
\end{aligned}
$$

**Technical novelties.** Our lower-bound analysis builds on the Local Fano method, but requires several new ingredients to address three structural obstacles specific to our setting. First, we confront a regularity mismatch between the observation and the target quantity: unlike (Lu et al., 2021), where the observation involves a differential operator that smooths the estimation problem, here we must recover high-order $H^1$ information from lower-order $L^2$ observations.

Second, we characterize the information-theoretic limits under a heteroscedastic observation model induced by the joint sampling of the source and target densities. Finally, we handle the global normalization constraint intrinsic to density estimation, showing that local perturbations remain statistically indistinguishable even after enforcing normalization through global partition functions.

**Discussion on the optimality gap.** Comparing Theorem 3.1 with Theorem 4.1, we observe a gap between the achievable upper rate $\tilde{\mathcal{O}}(n^{-\frac{s-1}{d+2s-2}})$ and the minimax lower bound $\tilde{\Omega}(n^{-\frac{2(s-1)}{d+2s}})$. We conjecture that this suboptimality is an artifact of the saturation phenomenon associated with single-step Tikhonov-type regularization (Bauer et al., 2007; Engl et al., 1996). Within our framework, the Sobolev penalty $\lambda\|\nabla f\|^2$ induces a bias–variance trade-off in which the regularization parameter $\lambda$ simultaneously governs approximation error and statistical stability. In particular, enforcing conditional strong convexity requires $\lambda$ to be sufficiently large to control the stochastic error–manifested through the $1/\lambda$ factor in the gradient stability bounds. This constraint limits how rapidly the bias can decay, thereby preventing the estimator from attaining the optimal nonparametric rate.

Theoretically, the gap between the upper and lower bounds could be closed using iterative regularization schemes, such as iterated Tikhonov regularization (Engl et al., 1996), which are known to attain optimal convergence rates. However, implementing such methods in a deep learning setting would require training a sequence of neural networks where each subsequent network relies on the previous one, leading to prohibitive computational and memory costs for large-scale applications such as diffusion model and transfer learning. Consequently, while our single-step estimator is not minimax optimal, it is computationally efficient and practically scalable, and already exhibits strong empirical performance and stability in the small-sample regime (see Table 2).

## 5. Generalization to diffusion models: time dependence and unbounded domains

In this section, we extend our framework to transfer learning for diffusion models. Compared to the static setting in Section 2, two additional challenges arise: (i) the density ratio becomes time dependent; and (ii) Gaussian perturbations render the data support unbounded. We address both issues by reformulating the problem on an augmented time-space domain, applying a truncation argument and proposing a projected-rescaled algorithm.

**Augmented time-space formulation under the VP forward model.** To formalize the diffusion-model setting, we focus on the variance-preserving (VP) forward noising model. Let $X_0 \sim p_0$ and $Y_0 \sim q_0$ denote the source and target initial variables. For $t \in [0, T]$, assume

$$X_t = \alpha_t X_0 + \sigma_t \xi, \ Y_t = \alpha_t Y_0 + \sigma_t \xi', \ \xi, \xi' \sim \mathcal{N}(0, I_d),$$

where $\alpha_t \in \mathbb{R}$ and $\sigma_t > 0$ are deterministic scalar VP coefficients.

Fix $t_0 \in (0, T)$ as the early-stopping time, commonly used in the diffusion model literature (Han et al., 2024a). In this context, the goal is to estimate the score difference $\nabla_x f^\star(t, x) := \nabla_x \log(q_t(x)/p_t(x))$ for $t \in [t_0, T]$. We consider the corresponding learning problem on the augmented space:

$$\tilde{\Omega} := [t_0, T] \times \mathbb{R}^d, \quad \tilde{z} := (t, x) \in \tilde{\Omega}.$$

We define the time-dependent mixture measure as $\mu_t := \frac{1}{2}(p_t + q_t)$, which serves as our reference measure on each time slice.

The objective is to learn a function $f : \tilde{\Omega} \to \mathbb{R}$ that minimizes a time-averaged risk. Since the temporal domain $[t_0, T]$ is compact, the main difficulty arises from the unbounded spatial domain $\mathbb{R}^d$. In particular, strong convexity fails globally on $\mathbb{R}^d$. To address this issue, we impose suitable assumptions on the tail behavior of the data distribution, which is common and reasonable assumptions to assume in the diffusion model literature (Han et al., 2024a; Li et al., 2023; Kong et al., 2024).

**Assumption 5.1** (Compact initial support and VP coefficients)**.** The initial source and target distributions are compactly supported: there exists $R_0 < \infty$ such that

$$\text{supp}(p_0) \cup \text{supp}(q_0) \subseteq B_{R_0}.$$

Moreover, the VP coefficients satisfy $\alpha, \sigma \in C^s([t_0, T])$, and there exist constants $0 < \underline{\sigma} \leq \overline{\sigma} < \infty, \omega_0 < \infty$, such that for all $t \in [t_0, T]$,

$$\underline{\sigma} \leq \sigma_t \leq \overline{\sigma}, \qquad \max_{0 \leq a \leq s} (|\partial_t^a \alpha_t| + |\partial_t^a \sigma_t|) \leq \omega_0.$$

**Truncation-dependent clipped network class.** For $R > 0$, the network is trained on the rescaled domain $[t_0, T] \times B_1$ with input $(t, \bar{x}) \in \mathbb{R}^{d+1}$. We use

$$\mathcal{F}_R := \mathcal{F}_{M_R}^{(d+1)}(L, W, S, B), \qquad M_R := C_M(1 + R),$$

where $\mathcal{F}_{M_R}^{(d+1)}$ is the clipped sparse ReLU[3] class from Section 3, now with input dimension $d + 1$. The constant $C_M$ is fixed independently of $R, n, N, \lambda$ and is chosen large enough so that the envelope requirements in Appendix C.2 hold. In particular, for every $R \geq 1$,

$$\|f_R^\star\|_{L^\infty([t_0,T]\times B_1)} + \|\nabla_{\bar{x}} f_R^\star\|_{L^\infty([t_0,T]\times B_1)} \leq M_R.$$

This follows from the growth bounds under Assumption 5.1.

**Projected-rescaled empirical objective.** Define the Euclidean projection onto $B_{2R}$ by

$$\text{Proj}_{2R}(x) := \frac{x}{\max\{1, \|x\|/(2R)\}}.$$

Then $\bar{x} := \frac{\text{Proj}_{2R}(x)}{2R} \in B_1$. To approximate the time-integrated source–target classification risk, we augment the source and target samples with independent time draws,

$$\mathcal{D}_{\text{ext}} := \{(t_i^p, X_i^p, 0)\}_{i=1}^n \cup \{(t_i^q, X_i^q, 1)\}_{i=1}^n,$$

where $t_i^p, t_i^q \overset{\text{i.i.d.}}{\sim} \text{Unif}([t_0, T])$, $X_i^p \mid t_i^p \sim p_{t_i^p}$, $X_i^q \mid t_i^q \sim q_{t_i^q}$, with all variables independent across the $p$-sample and $q$-sample blocks and across indices, and $\bar{X}_i^p := \frac{\text{Proj}_{2R}(X_i^p)}{2R}, \bar{X}_i^q := \frac{\text{Proj}_{2R}(X_i^q)}{2R}$. we estimate the rescaled log-density ratio by

$$\widehat{f}_R \in \arg\min_{f \in \mathcal{F}_R} \widehat{J}_{\lambda, \mathcal{D}_{\text{ext}}, 2R}^{\text{proj}}(f),$$

where

$$\widehat{J}_{\lambda, \mathcal{D}_{\text{ext}}, 2R}^{\text{proj}}(f) :=$$
$$\frac{1}{2n} \sum_{i=1}^n \left[ \ell_{\text{CE}}\big(0, f(t_i^p, \bar{X}_i^p)\big) + \frac{\lambda}{4R^2} \|\nabla_{\bar{x}} f(t_i^p, \bar{X}_i^p)\|_2^2 \right]$$
$$+ \frac{1}{2n} \sum_{i=1}^n \left[ \ell_{\text{CE}}\big(1, f(t_i^q, \bar{X}_i^q)\big) + \frac{\lambda}{4R^2} \|\nabla_{\bar{x}} f(t_i^q, \bar{X}_i^q)\|_2^2 \right],$$
$$(13)$$

The factor $1/(4R^2)$ accounts for the spatial rescaling $x = 2R\bar{x}$.

**Cutoff extension and core-tail decomposition.** Let $\chi_R : \mathbb{R}^d \to [0, 1]$ be a smooth cutoff satisfying

$$\chi_R \equiv 1 \text{ on } B_R, \chi_R \equiv 0 \text{ on } \mathbb{R}^d \setminus B_{2R}, \|\nabla \chi_R\|_\infty \lesssim R^{-1}.$$

Using the optimizer $\widehat{f}_R$ from (13), define the global estimator on $\widetilde{\Omega} = [t_0, T] \times \mathbb{R}^d$ by

$$\widetilde{f}^{(R)}(t, x) := \chi_R(x) \widehat{f}_R\left(t, \frac{\text{Proj}_{2R}(x)}{2R}\right). \quad (14)$$

Define the local loss density associated with the $H^1(\mu_t)$ norm by

$$\mathcal{L}(f, g) := |f - g|^2 + \|\nabla_x f - \nabla_x g\|_2^2, \quad (15)$$

where all terms are evaluated pointwise. We decompose the time-averaged global error as

$$\int_{t_0}^T \|\widetilde{f}_t^{(R)} - f_t^\star\|_{H^1(\mu_t)}^2 \, dt = \underbrace{\int_{t_0}^T \int_{B_R} \mathcal{L}(\widetilde{f}^{(R)}, f^\star) \, d\mu_t \, dt}_{\textbf{Main error}}$$
$$+ \underbrace{\int_{t_0}^T \int_{B_R^c} \mathcal{L}(\widetilde{f}^{(R)}, f^\star) \, d\mu_t \, dt}_{\textbf{Tail error}}.$$
$$(16)$$

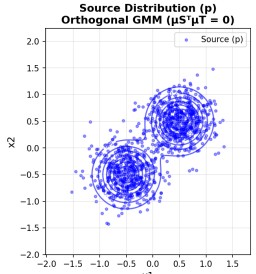
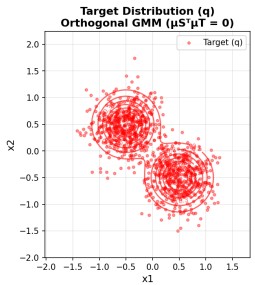

*Figure 3.* One example of the simulation distributions (All distribution pairs can be found in Appendix D.1).

The main error is controlled by applying the compact-domain analysis to the pulled-back estimator on $[t_0, T] \times B_{2R}$. The tail error is controlled by the sub-Gaussian decay of $\mu_t$ under Assumption 5.1, together with the growth bounds for $f^\star$ implied by the VP structure. Choosing $R = R(n)$ then balances these two errors and yields Theorem 5.2.

**Theorem 5.2** (Convergence for VP diffusion models on unbounded domains). *Suppose Assumption 5.1 holds and the smoothness index satisfies $s \leq 4$. For each $R > 0$, consider the $R$-dependent clipped and spatial-gradient-bounded time-space class $\mathcal{F}_R = \mathcal{F}_{M_R}^{(d+1)}(L, W, S, B)$ with $M_R = C_M(1 + R)$. Let $N_n^{\text{ext}} \asymp n^{\frac{d+1}{d+1+2s-2}}, L = \mathcal{O}(1), W, S, B \asymp N_n^{\text{ext}}, \lambda \asymp (N_n^{\text{ext}})^{-\frac{s-1}{d+1}}$. Choose $R = A\sqrt{\log n}$, where $A > 0$ is sufficiently large depending only on $t_0, T, R_0, \omega_0, \underline{\sigma}, \overline{\sigma}$. Then, with probability at least $1 - 3n^{-2}$,*

$$\int_{t_0}^T \|\widetilde{f}_t^{(R)} - f_t^\star\|_{H^1(\mu_t)}^2 \, dt \leq C e^{C\sqrt{\log n}} (\log n)^K n^{-\frac{s-1}{d+1+2s-2}},$$

*where $\widetilde{f}_t^{(R)}$ is defined in (14) and $C, K < \infty$ are independent of $n$. As a consequence, for every $\varepsilon > 0$, there exists $C_\varepsilon < \infty$ such that*

$$\int_{t_0}^T \|\widetilde{f}_t^{(R)} - f_t^\star\|_{H^1(\mu_t)}^2 \, dt \leq C_\varepsilon n^{-\frac{s-1}{d+1+2s-2} + \varepsilon}.$$

# 6. Experiments

This section presents experiments on both synthetic environments and real-world datasets, demonstrating the promising performance of our proposed method.

### 6.1. Estimation error in simulation environment

In this simulation study, we show the effectiveness of our regularization, in terms of the accuracy of estimating the gradient of the log density ratio $\nabla \log \frac{q}{p}$.

We construct three distinct pairs of two–dimensional source and target distributions $(p, q)$, for which the quantity

$\nabla \log \frac{q}{p}$ can be computed in closed form. Detailed descriptions of the three distribution pairs are provided in Appendix D.1. For each pair we train a three–layer MLP $f_\theta(x)$ to approximate the log density ratio.

We consider two training objectives: a standard classification-based objective, and our Sobolev-penalized classification objective. For each $(p, q)$ pair we train the network with $N \in \{10, 100, 1000\}$ samples drawn independently from both $p$ and $q$. After training, we set the gradient estimator directly as $g_\theta(x) := \nabla_x f_\theta(x)$, and estimate its mean squared error using 1000 new samples from both $p$ and $q$.

The errors are reported in Table 2. We first observe that the Sobolev-penalized estimator (*Cls w sob*) consistently outperforms the naive classification baseline (*Cls w/o sob*) across all training sizes. Notably, this relative improvement is most pronounced in the small-sample regime (*e.g.*, more than a $50\%$ improvement for $N = 20$), confirming that the Sobolev penalty effectively stabilizes the estimator when data are scarce.

### 6.2. Transfer learning with real-world datasets

In transfer learning, the goal is to leverage labeled samples from a source distribution to improve prediction on a related target distribution. We observe source data $\mathcal{D}_p := \{(x_p^{(i)}, z_p^{(i)})\}_{i=1}^{n_p}$ drawn from a joint distribution $P$ and target samples $\mathcal{D}_q := \{(x_q^{(j)}, z_q^{(j)})\}_{j=1}^{n_q}$ drawn from a different joint distribution $Q$. In practice, labeled target samples are often scarce, making direct training unreliable. Our objective is therefore to infer a labeling function for $\mathcal{D}_q$ by transferring information from $\mathcal{D}_p$.

#### 6.2.1. WGF METHODS

When many unlabeled target samples are available, following prior optimal-transport-based adaptation methods(Liu et al., 2023), we use WGF to transport labeled source particles toward the target distribution and then train a classifier on the transported samples.

WGF requires the score difference $\nabla \log q - \nabla \log p_t$. We therefore evaluate the proposed estimators by implementing WGF on the Office–Caltech-10 benchmark (Saenko et al., 2010), which contains four visual datasets: Amazon, Caltech, DSLR, and Webcam. Following standard practice, all samples are projected onto a 100-dimensional PCA subspace. We compare target classification accuracy across four settings: a source-only RBF SVM baseline, and RBF SVMs trained on WGF-transported source samples using (i) kernel-based estimators (LL; (Liu et al., 2023)), (ii) unregularized classification-based estimators, and (iii) our Sobolev-regularized classification estimator.

The results (Table 3) show that directly reusing source classifiers can lead to severe performance degradation (e.g., Amazon → DSLR), whereas joint distribution–based adaptation substantially alleviates this issue. Moreover, Sobolev regularization enables the classification-based method to achieve the strongest overall performance.

While kernel-based estimators attain accuracy comparable to our Sobolev-regularized approach in low-dimensional settings, they incur substantially higher computational cost. The kernel-based WGF baseline is intrinsically local, requiring a new gradient estimator to be trained at each WGF iteration through kernel optimization, which repeatedly evaluates interactions with the training samples (see Table 4). In contrast, the classification-based estimator amortizes gradient evaluation: once trained, the score difference at a new point is obtained via a single forward and backward pass of a fixed network. This amortization accounts for the significant speedup and enables scalability to large-scale settings, including diffusion-based generation.

#### 6.2.2. DIFFUSION MODELS

When the amount of unlabeled target data is extremely limited, directly training a generative model on the target task is often unreliable. A more effective alternative is to leverage a large source domain and adapt a pre-trained diffusion model to the target distribution via transfer. In particular, recent work (Ouyang et al., 2024) proposes to guide a source-trained diffusion model using an estimated score difference between the source and target distributions.

To evaluate the effectiveness of such transfer-based diffusion model methods under limited target data, we consider a benchmark task in electrocardiogram (ECG) generation. Following (Ouyang et al., 2024), we use the PTB-XL dataset (Wagner et al., 2020) as the source task and the ICBEB2018 dataset (Liu et al., 2018) as the target task, whose details can be found on Appendix D.3.

**Evaluation protocol.** We assess different methods through their ability to generate target-task samples for downstream classification. (More results on computing time and generation quality can be found on Appendix D.3) Specifically, each method is used to generate a sufficient number of synthetic ECG samples, which are then combined with the limited target samples to train a classifier. Performance is evaluated on the target test set. We compare four approaches: (1) *Vanilla Diffusion*, which trains a diffusion model directly on limited target data; (2) *Finetune Generator*, which adapts a source-trained diffusion model to generate target-label samples; (3) *TGDP* (Ouyang et al., 2024), which trains the guidance function using standard classification objectives; and (4) *TGDP-SoB*, which incorporates Sobolev regularization into the TGDP framework.

*Table 1.* Accuracy for gradient of logdensity ratio estimation. Best results are marked in bold. Improvement indicates the relative reduction in error from Cls w/o sob to Cls w sob.

| Train Size | Cls w/o sob (A) | Cls w sob (B) | Kernel | Improvement $(A - B)/A$ [%] |
|---|---|---|---|---|
| **ROTATED_RIDGE** | | | | |
| 20 | $61.94 \pm 7.98$ | $30.96 \pm 6.92$ | $\mathbf{7.65 \pm 1.24}$ | 50.02 |
| 200 | $5.91 \pm 0.92$ | $\mathbf{4.29 \pm 0.61}$ | $6.44 \pm 0.43$ | 27.36 |
| 2000 | $1.68 \pm 0.21$ | $\mathbf{0.96 \pm 0.12}$ | $5.80 \pm 0.06$ | 42.96 |
| **ORTHOGONAL_GMM** | | | | |
| 20 | $24.10 \pm 2.73$ | $\mathbf{11.89 \pm 0.80}$ | $25.87 \pm 1.42$ | 50.66 |
| 200 | $8.90 \pm 2.74$ | $\mathbf{6.31 \pm 0.44}$ | $23.78 \pm 0.63$ | 29.19 |
| 2000 | $5.03 \pm 0.34$ | $\mathbf{3.83 \pm 0.17}$ | $22.80 \pm 0.21$ | 23.86 |
| **BOUNDED** | | | | |
| 20 | $103.46 \pm 7.16$ | $48.77 \pm 2.62$ | $\mathbf{7.73 \pm 0.59}$ | 52.86 |
| 200 | $15.68 \pm 2.45$ | $9.86 \pm 1.19$ | $\mathbf{3.68 \pm 0.09}$ | 37.12 |
| 2000 | $2.42 \pm 0.17$ | $\mathbf{1.96 \pm 0.13}$ | $3.55 \pm 0.01$ | 18.88 |

*Table 2.* Accuracy for gradient of log-density ratio estimation. Values are reported as mean (standard deviation). Best results are marked in bold.

| Train Size | Cls w/o sob (A) | Cls w sob (B) | Improve $(A - B)/A$ [%] |
|---|---|---|---|
| **ROTATED_RIDGE** | | | |
| 20 | 61.94(7.98) | **30.96(6.92)** | 50.02 |
| 200 | 5.91(0.92) | **4.29(0.61)** | 27.36 |
| 2000 | 1.68(0.21) | **0.96(0.12)** | 42.96 |
| **ORTHOGONAL_GMM** | | | |
| 20 | 24.10(2.73) | **11.89(0.80)** | 50.66 |
| 200 | 8.90(2.74) | **6.31(0.44)** | 29.19 |
| 2000 | 5.03(0.34) | **3.83(0.17)** | 23.86 |
| **BOUNDED** | | | |
| 20 | 103.46(7.16) | **48.77(2.62)** | 52.86 |
| 200 | 15.68(2.45) | **9.86(1.19)** | 37.12 |
| 2000 | 2.42(0.17) | **1.96(0.13)** | 18.88 |

*Table 3.* Comparison of classification accuracy on Office-Caltech-10 Dataset

| $\mathcal{D}_p \to \mathcal{D}_q$ | Base | Cls w/o sob | Kernel | Cls w sob |
|---|---|---|---|---|
| amz.→cal. | 0.7115 | 0.7863 | 0.8379 | **0.8504** |
| amz.→dslr | 0.2675 | 0.7580 | **0.7962** | **0.7962** |
| amz.→web. | 0.3932 | 0.7390 | **0.8678** | 0.8203 |
| cal.→amz. | **0.9081** | 0.8716 | **0.9081** | 0.8977 |
| cal.→dslr | 0.2420 | 0.8535 | 0.8344 | **0.8623** |
| cal.→web. | 0.3797 | 0.7763 | **0.8203** | 0.8034 |
| dslr→amz. | 0.7035 | 0.8017 | **0.8716** | 0.8006 |
| dslr→cal. | 0.6572 | 0.7489 | **0.8094** | 0.7427 |
| dslr→web. | 0.9492 | 0.9492 | 0.9492 | **0.9525** |
| web.→amz. | 0.6294 | 0.6002 | 0.5877 | **0.7046** |
| web.→cal. | 0.3954 | 0.7070 | **0.7640** | 0.6456 |
| web.→dslr | 0.8535 | 0.9618 | 0.9554 | **0.9682** |

*Table 4.* Average gradient computation time across the 12 transfer tasks shown in Table 3

| Method | Gradient Calculation Time |
|---|---|
| Baseline | N/A |
| Kernel | 2.0617 s |
| Cls w/o sob | **0.0004 s** |
| Cls w sob | **0.0004 s** |

Following the ECG benchmark protocol in (Strodthoff et al., 2020), we report the Macro-averaged area under the ROC curve (AUC), macro-averaged $F_\beta$-score with $\beta = 2$, and macro-averaged $G_\beta$-score with $\beta = 2$; where $F_\beta = \frac{(1+\beta^2) \cdot \text{TP}}{(1+\beta^2) \cdot \text{TP} + \beta^2 \cdot \text{FN} + \text{FP}}, G_\beta = \frac{\text{TP}}{\text{TP} + \text{FP} + \beta \cdot \text{FN}}$. As shown in Table 5, TGDP substantially outperforms the baseline methods across all evaluation metrics. In addition, the Sobolev-regularized variant (TGDP-SoB) consistently yields further improvements, highlighting the benefit of regularized score difference estimation for diffusion-based transfer under limited target data.

| Method | AUC | $F_{\beta=2}$ | $G_{\beta=2}$ |
|---|---|---|---|
| Vanilla Diffusion | 0.844(07) | 0.590(09) | 0.331(08) |
| Finetune Generator | 0.862(05) | 0.604(09) | 0.351(09) |
| TGDP | 0.905(04) | 0.662(10) | 0.436(12) |
| **TGDP-SoB** | **0.915(05)** | **0.693(11)** | **0.453(11)** |

*Table 5.* Results on ECG benchmark for downstream classification task. (90% confidence intervals are provided via empirical bootstrapping (Strodthoff et al., 2020); 0.915(04) stands for 0.915 ± 0.004.)

# Impact Statement

This paper presents work whose goal is to advance the field of machine learning. There are many potential societal consequences of our work, none of which we feel must be specifically highlighted here.

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

## A. Proof for Bounded Case (Section 3)

**Notation**   For the bounded-domain fixed source–target design in (5), recall the labeled sample $\mathcal{D}$ defined in the main text. For any measurable function $\varphi : \Omega \times \{0,1\} \to \mathbb{R}$, define

$$\overline{P}\varphi := \frac{1}{2}\mathbb{E}_{X\sim p}[\varphi(X,0)] + \frac{1}{2}\mathbb{E}_{X\sim q}[\varphi(X,1)],$$

and

$$\overline{P}_n\varphi := \frac{1}{2n}\sum_{i=1}^{n}\varphi(X_i^p,0) + \frac{1}{2n}\sum_{i=1}^{n}\varphi(X_i^q,1).$$

The population operator $\overline{P}$ agrees with expectation under $\mu_{p,q}$. Hence

$$\widehat{L}_{\mathrm{CE},\mathcal{D}}(f) := \overline{P}_n\{\ell_{\mathrm{CE}}(y,f(x))\}.$$

We first show that the minimizer for our energy functional exists and is unique.

**Lemma A.1** (Existence and uniqueness of the population risk minimizer in $\mathcal{H}_M$). *For any regularization parameter $\lambda > 0$, the define the Sobolev-regularized population risk*

$$J_\lambda(f) := L_{CE}(f) + \lambda\mathcal{R}(f).$$

*Then $J_\lambda$ admits a unique minimizer $f_\lambda \in \mathcal{H}_M$:*

$$f_\lambda := \arg\min_{f\in\mathcal{H}_M} J_\lambda(f).$$

*Proof.* We proceed by establishing uniqueness via strong convexity and existence via the properties of the constraint set $\mathcal{H}_M$.

From Lemma 3.3, the functional $J_\lambda$ satisfies the following strong convexity inequality for any $f,g \in \mathcal{H}_M$:

$$J_\lambda(f) - J_\lambda(g) - DJ_\lambda(g)[f-g] \geq \frac{c_{min}}{2}\|f-g\|_{L^2(\mu)}^2 + \lambda\|\nabla(f-g)\|_{L^2(\mu)}^2. \tag{17}$$

Since $\lambda > 0$ and $c_{min} > 0$, the right-hand side is strictly positive for any $f \neq g$ (in the $H^1(\mu)$ norm sense). Suppose there exist two distinct minimizers $f_1, f_2 \in \mathcal{H}_M$ with $J_\lambda(f_1) = J_\lambda(f_2) = \inf_{f\in\mathcal{H}_M} J_\lambda(f) = m$. By the strict convexity, for any $t \in (0,1)$, we have:

$$J_\lambda(tf_1 + (1-t)f_2) < tJ_\lambda(f_1) + (1-t)J_\lambda(f_2) = m.$$

This implies we have found an element with a risk strictly lower than the infimum $m$, which is a contradiction. Thus, the minimizer must be unique.

To apply the Direct Method on the constrained set, we first establish that the feasible set $\mathcal{H}_M$ is a weakly closed subset of $H^1(\mu)$.

First, $\mathcal{H}_M$ is convex. For any $f,g \in \mathcal{H}_M$ and $t \in [0,1]$, by the triangle inequality:

$$\|tf + (1-t)g\|_{L^\infty} \leq t\|f\|_{L^\infty} + (1-t)\|g\|_{L^\infty} \leq tM + (1-t)M = M.$$

Thus, the convex combination remains in $\mathcal{H}_M$. Second, $\mathcal{H}_M$ is closed in the strong topology of $H^1(\mu)$. Let $\{f_n\} \subset \mathcal{H}_M$ be a sequence converging to $f$ in $H^1(\mu)$. Convergence in $H^1$ implies convergence in $L^2$, which in turn implies the existence of a subsequence converging pointwise almost everywhere. Since $|f_n(x)| \leq M$ a.e., the pointwise limit must satisfy $|f(x)| \leq M$ a.e. Thus, $f \in \mathcal{H}_M$. Since $\mathcal{H}_M$ is a closed and convex subset of a Banach space, it is weakly closed.

Next we show coercivity. Although functions in $\mathcal{H}_M$ are bounded in $L^\infty$, they are not a priori bounded in $H^1(\mu)$ (as gradients can be arbitrarily large). However, fixing a reference $g = 0 \in \mathcal{H}_M$, the inequality (17) implies:

$$J_\lambda(f) \geq J_\lambda(0) + DJ_\lambda(0)[f] + C(\lambda)\|f\|_{H^1(\mu)}^2.$$

Since the quadratic term dominates the linear functional as $\|f\|_{H^1} \to \infty$, $J_\lambda$ is coercive. This ensures that any minimizing sequence $\{f_n\}_{n=1}^\infty \subset \mathcal{H}_M$ such that $J_\lambda(f_n) \to \inf_{f \in \mathcal{H}_M} J_\lambda(f)$ is bounded in the $H^1(\mu)$ norm.

Finaly, we take limit via weak lower semicontinuity. Since the minimizing sequence $\{f_n\} \subset \mathcal{H}_M$ is bounded in the reflexive space $H^1(\mu)$, by the Banach-Alaoglu theorem, there exists a subsequence $\{f_{n_k}\}$ that converges weakly to some limit $f_\lambda \in H^1(\mu)$. Crucially, because $\mathcal{H}_M$ is weakly closed, the limit must satisfy the constraints: $f_\lambda \in \mathcal{H}_M$. Finally, since $J_\lambda$ is continuous and convex, it is weakly lower semicontinuous. Therefore:

$$J_\lambda(f_\lambda) \le \liminf_{k \to \infty} J_\lambda(f_{n_k}) = \inf_{f \in \mathcal{H}_M} J_\lambda(f).$$

Thus, the minimum is attained by $f_\lambda$ within the set $\mathcal{H}_M$. $\qquad\square$

Then we provide some properties about sparse neural network $\mathcal{F}$.

**Proposition A.2** (Boundedness of DNN). *The gradients and function value of $f \in \mathcal{F}$ are uniformly bounded: $\exists M \ge M^\star$, s.t*

$$\max \left\{ \sup_{f \in \mathcal{F}(\Omega)} \|f\|_{L^\infty(\mu)}, \sup_{f \in \mathcal{F}(\Omega)} \|\nabla f\|_{L^\infty(\mu)}, \|f^\star\|_{L^\infty(\mu)}, \|\nabla f^\star\|_{L^\infty(\mu)}, \right\} \le M.$$

*Proof.* Proof can be found on the proof of Lemma A.15 and Lemma A.19 in (Lu et al., 2021). $\qquad\square$

**Lemma A.3** (Tensor-product B-spline quasi-interpolation). *Fix integers $k, l \in \mathbb{N}$. Let*

$$0 = t_0^{(l)} < t_1^{(l)} < \cdots < t_l^{(l)} = 1, \qquad t_i^{(l)} := \frac{i}{l},$$

*be the uniform partition of $[0, 1]$. Define the extended knot sequence*

$$t_{-k+1}^{(l)} = \cdots = t_{-1}^{(l)} = t_0^{(l)} = 0, \qquad t_l^{(l)} = t_{l+1}^{(l)} = \cdots = t_{l+k-1}^{(l)} = 1.$$

*Let*

$$I_{l,k} := \{-k+1, -k+2, \ldots, l-1\}.$$

*For $i \in I_{l,k}$, define the univariate B-spline of order $k$ by*

$$N_{l,i}^{(k)}(x) := (-1)^k \big(t_{i+k}^{(l)} - t_i^{(l)}\big) \big[t_i^{(l)}, t_{i+1}^{(l)}, \ldots, t_{i+k}^{(l)}\big] \left\{ (x - t)_+^{k-1} \right\}, \qquad x \in [0, 1], \tag{18}$$

*where $[\cdot]$ denotes the divided-difference operator with respect to the variable $t$, and $(a)_+ := \max\{a, 0\}$.*

*Equivalently, for interior indices $0 \le i \le l - k + 1$,*

$$N_{l,i}^{(k)}(x) = \frac{l^{k-1}}{(k-1)!} \sum_{j=0}^{k} (-1)^j \binom{k}{j} \left(x - \frac{i+j}{l}\right)_+^{k-1}. \tag{19}$$

*The boundary splines are defined by the repeated endpoint knots above.*

*For a multi-index*

$$\mathbf{i} = (i_1, \ldots, i_d) \in I_{l,k}^d,$$

*define the tensor-product B-spline*

$$N_{l,\mathbf{i}}^{(k)}(x) := \prod_{m=1}^{d} N_{l,i_m}^{(k)}(x_m), \qquad x = (x_1, \ldots, x_d) \in [0, 1]^d. \tag{20}$$

*Let $\{\Lambda_{\mathbf{i}}\}_{\mathbf{i} \in I_{l,k}^d}$ be the standard local B-spline quasi-interpolation functionals associated with the tensor basis $\{N_{l,\mathbf{i}}^{(k)}\}_{\mathbf{i} \in I_{l,k}^d}$. Define the quasi-interpolation operator*

$$Q_{k,l} g := \sum_{\mathbf{i} \in I_{l,k}^d} \Lambda_{\mathbf{i}}(g) N_{l,\mathbf{i}}^{(k)}. \tag{21}$$

*Then $Q_{k,l}$ is local and stable. In particular, there exists a constant $C_{\mathrm{qi}} < \infty$, depending only on $k$ and $d$, but independent of $l$, such that for every $g \in W^{1,\infty}([0,1]^d)$,*

$$\|Q_{k,l}g\|_{L^\infty} \leq C_{\mathrm{qi}}\|g\|_{L^\infty}, \qquad \|\nabla Q_{k,l}g\|_{L^\infty} \leq C_{\mathrm{qi}}\|\nabla g\|_{L^\infty}. \tag{22}$$

*Moreover, for every $g \in H^s([0,1]^d)$, $s \in \mathbb{N}$, and every $0 \leq r \leq s$, if $k \geq s$, then*

$$\|Q_{k,l}g - g\|_{H^r} \leq Cl^{-(s-r)}\|g\|_{H^s}, \tag{23}$$

*where $C$ depends only on $k, s, r, d$.*

*Proof.* The definitions (18)–(20) are the standard univariate and tensor-product B-splines with repeated endpoint knots. The operator $Q_{k,l}$ in (21) is the standard local B-spline quasi-interpolation operator.

We first recall its stability. Since the B-splines have compact support, form a partition of unity, and have uniformly bounded overlap, each point $x \in [0,1]^d$ belongs to the support of only $O_{k,d}(1)$ tensor-product basis functions. The local functionals $\Lambda_{\mathbf{i}}$ depend only on $g$ in a fixed-size neighborhood of $\mathrm{supp}(N_{l,\mathbf{i}}^{(k)})$, and satisfy the standard coefficient stability bounds

$$|\Lambda_{\mathbf{i}}(g)| \leq C_{k,d}\|g\|_{L^\infty(\Omega_{\mathbf{i}})},$$

where $\Omega_{\mathbf{i}}$ is the local support neighborhood associated with $N_{l,\mathbf{i}}^{(k)}$. Therefore,

$$\|Q_{k,l}g\|_{L^\infty} \leq C_{\mathrm{qi}}\|g\|_{L^\infty}.$$

For the derivative bound, differentiating

$$Q_{k,l}g = \sum_{\mathbf{i}} \Lambda_{\mathbf{i}}(g) N_{l,\mathbf{i}}^{(k)}$$

produces derivatives of the B-spline basis of size $\mathcal{O}(l)$. The locality and polynomial-reproduction property of the quasi-interpolant imply the standard first-order coefficient-difference bound

$$|\Lambda_{\mathbf{i}}(g) - \Lambda_{\mathbf{j}}(g)| \leq C_{k,d}l^{-1}\|\nabla g\|_{L^\infty}$$

for neighboring indices $\mathbf{i}, \mathbf{j}$. The factor $l^{-1}$ in the coefficient difference cancels the $\mathcal{O}(l)$ factor from differentiating the B-splines. Using again the uniformly bounded overlap of the tensor basis yields

$$\|\nabla Q_{k,l}g\|_{L^\infty} \leq C_{\mathrm{qi}}\|\nabla g\|_{L^\infty}.$$

This proves (22).

Finally, the approximation estimate (23) is the standard Sobolev approximation property of tensor-product B-spline quasi-interpolation. It follows from the Bramble–Hilbert lemma and the polynomial reproduction of $Q_{k,l}$ up to degree $k-1$. This gives

$$\|Q_{k,l}g - g\|_{H^r} \leq Cl^{-(s-r)}\|g\|_{H^s},$$

whenever $k \geq s$ and $0 \leq r \leq s$. $\qquad\square$

**Proposition A.4** (Approximation by the clipped and gradient-bounded sieve). *Let $\Omega \subset \mathbb{R}^d$ be a bounded domain with sufficiently smooth boundary, and let $\mu_{p,q}$ be the mixture measure corresponding to a pair $(p,q)$ satisfying Assumption 2.1. Suppose $f^\star \in H^s(\mu_{p,q})$, $s \leq 4$, and*

$$\max\left\{\|f^\star\|_{L^\infty(\mu_{p,q})}, \|\nabla f^\star\|_{L^\infty(\mu_{p,q})}\right\} \leq M^\star.$$

*Let $\mathcal{C}_\Omega \subset \mathbb{R}^d$ be a cube containing $\Omega$. Since $\Omega$ has sufficiently smooth boundary, there exists a bounded extension operator*

$$\mathcal{E}: H^s(\Omega) \to H^s(\mathcal{C}_\Omega)$$

*such that*

$$\|\mathcal{E}f\|_{H^s(\mathcal{C}_\Omega)} \leq C_{\mathrm{ext}}\|f\|_{H^s(\Omega)}.$$

*Moreover, for functions with bounded first derivatives, we use the extension so that*

$$\|\mathcal{E}f\|_{W^{1,\infty}(\mathcal{C}_\Omega)} \le C_{\text{ext},\infty}\|f\|_{W^{1,\infty}(\Omega)}.$$

*Let $C_{\Omega,\text{ext,qi}}$ be a constant depending only on $\Omega$, the extension operator, the affine rescaling from $\mathcal{C}_\Omega$ to $[0,1]^d$, and the quasi-interpolation stability constant in Lemma A.3. Choose*

$$M > C_{\Omega,\text{ext,qi}}M^\star,$$

*and choose the clipping map $T_M$ so that it is the identity on $[-M_0, M_0]$ for some*

$$C_{\Omega,\text{ext,qi}}M^\star < M_0 < M.$$

*Then, for $L = \mathcal{O}(1)$, $W = \mathcal{O}(N)$, $S = \mathcal{O}(N)$, and $B = \mathcal{O}(N)$, there exists*

$$f_N^{\text{app}} \in \mathcal{F}_M(L, W, S, B)$$

*such that*

$$\|f_N^{\text{app}} - f^\star\|_{H^1(\mu_{p,q})}^2 \lesssim N^{-\frac{2(s-1)}{d}}\|f^\star\|_{H^s(\mu_{p,q})}^2. \tag{24}$$

*Proof.* Under Assumption 2.1, the weighted Sobolev norm induced by $\mu_{p,q}$ is equivalent to the usual Sobolev norm on $\Omega$. Thus it is enough to prove the approximation bound in the usual $H^1(\Omega)$ norm.

Let

$$\bar{f}^\star := \mathcal{E}f^\star$$

be the extension of $f^\star$ to $\mathcal{C}_\Omega$. By the boundedness of the extension operator,

$$\|\bar{f}^\star\|_{H^s(\mathcal{C}_\Omega)} \le C_{\text{ext}}\|f^\star\|_{H^s(\Omega)} \lesssim \|f^\star\|_{H^s(\mu_{p,q})}.$$

Moreover,

$$\|\bar{f}^\star\|_{W^{1,\infty}(\mathcal{C}_\Omega)} \le C_{\text{ext},\infty}M^\star.$$

Let

$$A : \mathcal{C}_\Omega \to [0,1]^d$$

be an invertible affine map, and define the rescaled function on the unit cube by

$$\tilde{f}^\star(z) := \bar{f}^\star(A^{-1}z), \qquad z \in [0,1]^d.$$

Since $A$ is fixed and depends only on $\Omega$, Sobolev norms before and after rescaling are equivalent up to constants depending only on $\Omega$. Hence

$$\|\tilde{f}^\star\|_{H^s([0,1]^d)} \lesssim \|\bar{f}^\star\|_{H^s(\mathcal{C}_\Omega)} \lesssim \|f^\star\|_{H^s(\mu_{p,q})}.$$

Similarly,

$$\|\tilde{f}^\star\|_{W^{1,\infty}([0,1]^d)} \lesssim \|\bar{f}^\star\|_{W^{1,\infty}(\mathcal{C}_\Omega)} \lesssim M^\star.$$

Let

$$l := \left\lceil N^{1/d} \right\rceil.$$

On the unit cube $[0,1]^d$, take the cubic B-spline quasi-interpolant

$$Q_{4,l}\tilde{f}^\star = \sum_{\mathbf{i} \in I_{l,4}^d} \Lambda_{\mathbf{i}}(\tilde{f}^\star)N_{l,\mathbf{i}}^{(4)}.$$

By Lemma A.3, with $k = 4$ and $r = 1$,

$$\|Q_{4,l}\tilde{f}^\star - \tilde{f}^\star\|_{H^1([0,1]^d)} \lesssim l^{-(s-1)}\|\tilde{f}^\star\|_{H^s([0,1]^d)}.$$

Define the approximating network on $\Omega$ by

$$F_N(x) := (Q_{4,l}\tilde{f}^\star)(Ax), \qquad x \in \Omega.$$

Using the equivalence of Sobolev norms under the fixed affine map and the identity $\bar{f}^\star|_\Omega = f^\star$, we obtain

$$\|F_N - f^\star\|_{H^1(\Omega)} \lesssim \|Q_{4,l}\tilde{f}^\star - \tilde{f}^\star\|_{H^1([0,1]^d)} \lesssim l^{-(s-1)}\|\tilde{f}^\star\|_{H^s([0,1]^d)}.$$

Therefore, since $l \asymp N^{1/d}$,

$$\|F_N - f^\star\|_{H^1(\Omega)} \lesssim N^{-\frac{s-1}{d}}\|f^\star\|_{H^s(\mu_{p,q})}.$$

By norm equivalence between $H^1(\Omega)$ and $H^1(\mu_{p,q})$, this also gives

$$\|F_N - f^\star\|_{H^1(\mu_{p,q})} \lesssim N^{-\frac{s-1}{d}}\|f^\star\|_{H^s(\mu_{p,q})}.$$

We next verify the fixed envelope. By the $W^{1,\infty}$-stability in Lemma A.3 on $[0,1]^d$, together with the affine rescaling bounds,

$$\|F_N\|_{L^\infty(\Omega)} \leq \|Q_{4,l}\tilde{f}^\star\|_{L^\infty([0,1]^d)} \leq C_{\mathrm{qi}}\|\tilde{f}^\star\|_{L^\infty([0,1]^d)} \leq C_{\Omega,\mathrm{ext},\mathrm{qi}}M^\star < M_0.$$

For the gradient, using

$$\nabla_x F_N(x) = A^\top \nabla_z(Q_{4,l}\tilde{f}^\star)(Ax),$$

we get

$$\|\nabla F_N\|_{L^\infty(\Omega)} \leq C_\Omega\|\nabla Q_{4,l}\tilde{f}^\star\|_{L^\infty([0,1]^d)} \leq C_{\Omega,\mathrm{ext},\mathrm{qi}}M^\star < M.$$

It remains to realize $F_N$ as a sparse ReLU$^3$ neural network. The quasi-interpolant $Q_{4,l}\tilde{f}^\star$ is a linear combination of tensor-product cubic B-splines on $[0,1]^d$. By the explicit formula for cubic B-splines,

$$N_{l,i}^{(4)}(z) = \frac{l^3}{3!}\sum_{j=0}^{4}(-1)^j\binom{4}{j}\left(z - \frac{i+j}{l}\right)_+^3 = \frac{1}{3!}\sum_{j=0}^{4}(-1)^j\binom{4}{j}(lz - (i+j))_+^3$$

for interior indices, and by the analogous endpoint formulas for boundary splines, every univariate cubic B-spline can be implemented by a ReLU$^3$ subnetwork. Products of nonnegative spline factors can be implemented by ReLU$^3$ networks using

$$xy = \frac{1}{2}\big[(x+y)^2 - x^2 - y^2\big],$$

together with the exact representation of $x^2$ by ReLU$^3$ units. Thus each tensor-product spline can be implemented by a ReLU$^3$ subnetwork, and summing over $\mathbf{i} \in I_{l,4}^d$ gives a sparse ReLU$^3$ network representing $Q_{4,l}\tilde{f}^\star$ on $[0,1]^d$.

Composing this network with the fixed affine map $A$ only modifies the first layer and changes constants depending on $\Omega$. Hence $F_N$ can be represented by a sparse ReLU$^3$ network with

$$L = \mathcal{O}(1), \qquad W = \mathcal{O}(N), \qquad S = \mathcal{O}(N), \qquad B = \mathcal{O}(N).$$

Here the $\mathcal{O}(\cdot)$ constants may depend on $\Omega$ and fixed problem parameters, but not on $N$.

Define

$$f_N^{\mathrm{app}} := T_M \circ F_N.$$

Since

$$\|F_N\|_{L^\infty(\Omega)} < M_0,$$

and $T_M$ is the identity on $[-M_0, M_0]$, the clipping is inactive on $\Omega$:

$$f_N^{\mathrm{app}} = F_N \quad \text{on } \Omega.$$

Therefore

$$\|f_N^{\mathrm{app}}\|_{L^\infty(\mu_{p,q})} \leq M, \qquad \|\nabla f_N^{\mathrm{app}}\|_{L^\infty(\mu_{p,q})} \leq M,$$

so

$$f_N^{\mathrm{app}} \in \mathcal{F}_M(L, W, S, B).$$

Finally,

$$\|f_N^{\mathrm{app}} - f^\star\|_{H^1(\mu_{p,q})} = \|F_N - f^\star\|_{H^1(\mu_{p,q})} \lesssim N^{-\frac{s-1}{d}}\|f^\star\|_{H^s(\mu_{p,q})}.$$

Squaring both sides proves (24). $\qquad\square$

## A.1. Proof of Lemma 3.3

*Proof.* We first analyze the cross-entropy loss. For any $x$, let $G_x(u) = -\eta(x) \log \sigma(u) - (1 - \eta(x)) \log(1 - \sigma(u))$ be the conditional risk. A direct calculation yields the second derivative:

$$G_x''(u) = \sigma(u)(1 - \sigma(u)) = \frac{1}{4 \cosh^2(u/2)}.$$

Since any $f \in \mathcal{H}_M$ satisfies $|f(x)| \leq M$ almost everywhere, the curvature is uniformly bounded from below and above:

$$c_{\min} \ \leq \ G_x''(u) \ \leq \ \frac{1}{4}, \quad \forall u \in [-M, M]. \tag{25}$$

Since $G_x(u)$ is smooth with bounded derivatives on this compact interval, standard arguments from the calculus of variations imply that $L_{\mathrm{CE}}$ is Fréchet differentiable with $DL_{\mathrm{CE}}(g)[h] = \mathbb{E}[G_X'(g(X))h(X)]$. Applying Taylor's theorem with integral remainder to $G_x$, we obtain:

$$G_x(u) - G_x(v) - G_x'(v)(u - v) = (u - v)^2 \int_0^1 (1 - t) G_x''\big(v + t(u - v)\big) \, \mathrm{d}t.$$

Using the bounds on $G_x''$, this implies

$$\frac{c_{\min}}{2}(u - v)^2 \ \leq \ G_x(u) - G_x(v) - G_x'(v)(u - v) \ \leq \ \frac{1}{8}(u - v)^2.$$

Integrating with respect to $\mu$ (setting $u = f(x), v = g(x)$) yields the strong convexity and smoothness estimates for $L_{\mathrm{CE}}$:

$$\frac{c_{\min}}{2}\|f - g\|_{L^2}^2 \ \leq \ L_{\mathrm{CE}}(f) - L_{\mathrm{CE}}(g) - DL_{\mathrm{CE}}(g)[f - g] \ \leq \ \frac{1}{2}\|f - g\|_{L^2}^2. \tag{26}$$

We then analyze the regularizer. The functional $\mathcal{R}(f) = \|\nabla f\|_{L^2(\mu)}^2$ is a standard quadratic form on the Hilbert space $H^1(\mu)$. It is strictly convex and Fréchet differentiable with $D\Omega(g)[h] = 2\langle \nabla g, \nabla h \rangle_{L^2}$. The polarization identity yields the exact expansion:

$$\mathcal{R}(f) - \Omega(g) - D\Omega(g)[f - g] = \|\nabla(f - g)\|_{L^2(\mu)}^2. \tag{27}$$

Finally, we put together above results. Recall $J_\lambda(f) = L_{\mathrm{CE}}(f) + \lambda\mathcal{R}(f)$. By linearity, $DJ_\lambda = DL_{\mathrm{CE}} + \lambda D\Omega$. Combining the lower bound from (26) and the identity (27), we obtain:

$$J_\lambda(f) - J_\lambda(g) - DJ_\lambda(g)[f - g] \ \geq \ \frac{c_{\min}}{2}\|f - g\|_{L^2}^2 + \lambda\|\nabla(f - g)\|_{L^2}^2,$$

which proves (10). Similarly, combining the upper bound from (26) with (27) yields (10). $\qquad\square$

## A.2. Proof of Lemma 3.4

*Proof.* Denote $\delta_\lambda := f_\lambda - f^\star$. We proceed in three steps.

Step 1: A pointwise calibration inequality along $f_\lambda - f^\star$. From (25) we have

$$c_{\min} \ \leq \ G_x''(u) \ \leq \ \frac{1}{4} \quad \text{for all } u \in [-M, M] \text{ and all } x.$$

By the fundamental theorem of calculus,

$$G_x'(f(x)) - G_x'(f^\star(x)) = \int_0^1 G_x''\big(f^\star(x) + t(f(x) - f^\star(x))\big) \, (f(x) - f^\star(x)) \, \mathrm{d}t.$$

Since $f^\star$ is the Bayes logit, we have $G_x'(f^\star(x)) = \sigma(f^\star(x)) - \eta(x) = 0$, so

$$G_x'(f(x)) \, (f(x) - f^\star(x)) = \left( \int_0^1 G_x''\big(f^\star(x) + t(f(x) - f^\star(x))\big) \, \mathrm{d}t \right)(f(x) - f^\star(x))^2.$$

Define
$$w(x) := \int_0^1 G_x''\big(f^\star(x) + t(f(x) - f^\star(x))\big)\,\mathrm{d}t.$$

By the bounds on $G_x''$, we have $c_{\min} \leq w(x) \leq c_{\max}$ for all $x$. Moreover $G_x'(f(x)) = \sigma(f(x)) - \eta(x)$, so
$$(\sigma(f(x)) - \eta(x))\,(f(x) - f^\star(x)) = w(x)\,(f(x) - f^\star(x))^2.$$

Taking expectation with respect to $X \sim \mu$ gives
$$\mathbb{E}\big[(\sigma(f(X)) - \eta(X))\,(f(X) - f^\star(X))\big] = \mathbb{E}\big[w(X)\,(f(X) - f^\star(X))^2\big],$$

and hence
$$c_{\min}\|f - f^\star\|_{L^2(\mu)}^2 \;\leq\; \mathbb{E}\big[(\sigma(f(X)) - \eta(X))\,(f(X) - f^\star(X))\big] \;\leq\; \frac{1}{4}\|f - f^\star\|_{L^2(\mu)}^2. \tag{28}$$

This is the desired "gradient-distance" inequality along the direction $f - f^\star$.

Step 2: First-order optimality of $f_\lambda$ and Green's identity. Since $f_\lambda$ minimizes $J_\lambda(f)$ over the convex set $\mathcal{H}_M$, and noting that $f^\star \in \mathcal{H}_M$ (which holds by the assumption $M \geq M^\star$), $f_\lambda$ satisfies the first-order variational inequality:
$$DJ_\lambda(f_\lambda)[f^\star - f_\lambda] \geq 0.$$

Let $\delta_\lambda := f_\lambda - f^\star$. The inequality above is equivalent to $DJ_\lambda(f_\lambda)[\delta_\lambda] \leq 0$, which expands to:
$$\mathbb{E}\big[(\sigma(f_\lambda) - \eta)\delta_\lambda\big] + 2\lambda\,(\nabla f_\lambda, \nabla \delta_\lambda)_{L^2(\mu)} \;\leq\; 0. \tag{29}$$

Decomposing $\nabla f_\lambda = \nabla f^\star + \nabla \delta_\lambda$ gives
$$(\nabla f_\lambda, \nabla \delta_\lambda)_{L^2(\mu)} = \|\nabla \delta_\lambda\|_{L^2(\mu)}^2 + (\nabla f^\star, \nabla \delta_\lambda)_{L^2(\mu)}.$$

By the Neumann boundary condition in Assumption 2.1 and Green's Identity, we have
$$-(\nabla \delta_\lambda, \nabla f^\star)_{L^2(\mu)} = (\Delta f^\star + \nabla f^\star \cdot \nabla \log \mu,\ \delta_\lambda)_{L^2(\mu)}.$$

Recall
$$\mathcal{K}f^\star := \Delta f^\star + \nabla f^\star \cdot \nabla \log \mu \in L^2(\mu).$$

Then
$$(\nabla f^\star, \nabla \delta_\lambda)_{L^2(\mu)} = -(\mathcal{K}f^\star, \delta_\lambda)_{L^2(\mu)}.$$

Substituting into (29) yields
$$\mathbb{E}\big[(\sigma(f_\lambda) - \eta)\delta_\lambda\big] + 2\lambda\|\nabla \delta_\lambda\|_{L^2(\mu)}^2 - 2\lambda(\mathcal{K}f^\star, \delta_\lambda)_{L^2(\mu)} \;\leq\; 0,$$

or equivalently
$$\mathbb{E}\big[(\sigma(f_\lambda) - \eta)\delta_\lambda\big] + 2\lambda\|\nabla \delta_\lambda\|_{L^2(\mu)}^2 \;\leq\; 2\lambda(\mathcal{K}f^\star, \delta_\lambda)_{L^2(\mu)}. \tag{30}$$

Step 3: Combining the sandwich inequality and Cauchy–Schwarz. Applying the lower bound in (28) with $f = f_\lambda$, we obtain
$$c_{\min}\|\delta_\lambda\|_{L^2(\mu)}^2 \;\leq\; \mathbb{E}\big[(\sigma(f_\lambda) - \eta)\delta_\lambda\big].$$

Combining this with (30), we get
$$c_{\min}\|\delta_\lambda\|_{L^2(\mu)}^2 + 2\lambda\|\nabla \delta_\lambda\|_{L^2(\mu)}^2 \;\leq\; 2\lambda(\mathcal{K}f^\star, \delta_\lambda)_{L^2(\mu)} \leq 2\lambda\,\|\mathcal{K}f^\star\|_{L^2(\mu)}\,\|\delta_\lambda\|_{L^2(\mu)}. \tag{31}$$

We now extract the desired bounds from (31).

*(i) Value bias.* Dropping the nonnegative gradient term on the left-hand side gives

$$c_{\min} \|\delta_\lambda\|_{L^2(\mu)}^2 \leq 2\lambda \|\mathcal{K}f^\star\|_{L^2(\mu)} \|\delta_\lambda\|_{L^2(\mu)}.$$

If $\delta_\lambda \equiv 0$, the conclusion is trivial. Otherwise, dividing both sides by $\|\delta_\lambda\|_{L^2(\mu)}$,

$$\|\delta_\lambda\|_{L^2(\mu)} \leq \frac{2 \|\mathcal{K}f^\star\|_{L^2(\mu)}}{c_{\min}} \lambda.$$

Hence

$$\|f_\lambda - f^\star\|_{L^2(\mu)}^2 = \|\delta_\lambda\|_{L^2(\mu)}^2 \leq \Big(\frac{2 \|\mathcal{K}f^\star\|_{L^2(\mu)}}{c_{\min}}\Big)^2 \lambda^2. \tag{32}$$

*(ii) Gradient bias.* Substituting the bound $\|\delta_\lambda\|_{L^2(\mu)} \leq \big(2 \|\mathcal{K}f^\star\|_{L^2(\mu)}/c_{\min}\big)\lambda$ back into (31), we obtain

$$2\lambda\|\nabla\delta_\lambda\|_{L^2(\mu)}^2 \leq 2\lambda \|\mathcal{K}f^\star\|_{L^2(\mu)} \|\delta_\lambda\|_{L^2(\mu)} \leq 2\lambda \|\mathcal{K}f^\star\|_{L^2(\mu)} \cdot \frac{2 \|\mathcal{K}f^\star\|_{L^2(\mu)}}{c_{\min}} \lambda = \frac{4 \|\mathcal{K}f^\star\|_{L^2(\mu)}^2}{c_{\min}} \lambda^2.$$

Dividing by $2\lambda$ (recall $\lambda > 0$) yields

$$\|\nabla(f_\lambda - f^\star)\|_{L^2(\mu)}^2 = \|\nabla\delta_\lambda\|_{L^2(\mu)}^2 \leq \frac{2 \|\mathcal{K}f^\star\|_{L^2(\mu)}^2}{c_{\min}} \lambda. \tag{33}$$

Using the explicit value $c_{\min} = 1/(4\cosh^2(M/2))$, we can write

$$\Big(\frac{2 \|\mathcal{K}f^\star\|_{L^2(\mu)}}{c_{\min}}\Big)^2 = 64 \|\mathcal{K}f^\star\|_{L^2(\mu)}^2 \cosh^4(M/2), \qquad \frac{2 \|\mathcal{K}f^\star\|_{L^2(\mu)}^2}{c_{\min}} = 8 \|\mathcal{K}f^\star\|_{L^2(\mu)}^2 \cosh^2(M/2).$$

Thus we may take $\beta = 64 \|\mathcal{K}f^\star\|_{L^2(\mu)}^2 \cosh^4\big(\frac{M}{2}\big)$ so that

$$\max\Big\{64 \|\mathcal{K}f^\star\|_{L^2(\mu)}^2 \cosh^4\big(\tfrac{M}{2}\big),\ 8 \|\mathcal{K}f^\star\|_{L^2(\mu)}^2 \cosh^2\big(\tfrac{M}{2}\big)\Big\} \leq \beta.$$

$\square$

## A.3. Proof of Theorem 3.5

Let $\{\sigma_i^p\}_{i=1}^n$ and $\{\sigma_i^q\}_{i=1}^n$ be independent i.i.d. Rademacher variables. For the fixed source–target design, define the empirical Rademacher complexity of a function class $\mathcal{G}$ with respect to $\mathcal{D} = \{(X_i^p, 0)\}_{i=1}^n \cup \{(X_i^q, 1)\}_{i=1}^n$ by

$$\mathfrak{R}_n(\mathcal{G}; \mathcal{D}) := \mathbb{E}_\sigma\left[\sup_{g\in\mathcal{G}}\left\{\frac{1}{2n}\sum_{i=1}^n \sigma_i^p g(X_i^p, 0) + \frac{1}{2n}\sum_{i=1}^n \sigma_i^q g(X_i^q, 1)\right\}\,\Big|\,\mathcal{D}\right].$$

*Proof.* Fix $t > 0$. Define the shifted excess-loss function

$$g_f(x, y) := \ell_{\mathrm{CE}}(y, f(x)) - \ell_{\mathrm{CE}}(y, f_0(x)) + \lambda\left(\|\nabla f(x)\|_2^2 - \|\nabla f_0(x)\|_2^2\right). \tag{34}$$

Then, with $\overline{P}$ and $\overline{P}_n$ defined above,

$$\overline{P}g_f = J_\lambda(f) - J_\lambda(f_0),$$

and

$$\overline{P}_n g_f = \widehat{J}_{\lambda,\mathcal{D}}(f) - \widehat{J}_{\lambda,\mathcal{D}}(f_0).$$

Step 1: Local Rademacher complexity bound. For $r > 0$, define the localized set

$$\mathcal{U}_r := \{f \in \mathcal{F} : J_\lambda(f) - J_\lambda(f_\lambda) + J_\lambda(f_0) - J_\lambda(f_\lambda) \leq r\},$$

and define the shifted localized loss class

$$\mathcal{G}_r := \{g_f : \Omega \times \{0,1\} \to \mathbb{R} \mid f \in \mathcal{U}_r\}.$$

Here $g_f$ is the shifted excess loss in (34). We claim that for $r \gtrsim n^{-2}$,

$$\mathfrak{R}_n(\mathcal{G}_r; \mathcal{D}) \le \phi(r) := C_0 \left[ \frac{1}{n} + \sqrt{\frac{S \, 3^L r}{n} \log(BWn)} \right], \qquad (35)$$

and $\phi(4r) \le 2\phi(r)$.

We first prove the Lipschitz property of the shifted excess loss. Since

$$\partial_u \ell_{\mathrm{CE}}(y, u) = \sigma(u) - y,$$

the cross-entropy loss is 1-Lipschitz in $u$. For any $f_1, f_2 \in \mathcal{U}_r$, using the fixed clipped-gradient envelope

$$\|f_j\|_{L^\infty(\mu)} \le M, \qquad \|\nabla f_j\|_{L^\infty(\mu)} \le M, \qquad j = 1, 2,$$

we obtain

$$
\begin{aligned}
|g_{f_1}(x, y) - g_{f_2}(x, y)| &= \left| \ell_{\mathrm{CE}}(y, f_1(x)) - \ell_{\mathrm{CE}}(y, f_2(x)) + \lambda \left( \|\nabla f_1(x)\|_2^2 - \|\nabla f_2(x)\|_2^2 \right) \right| \\
&\le |f_1(x) - f_2(x)| + \lambda \left( \|\nabla f_1(x)\|_2 + \|\nabla f_2(x)\|_2 \right) \|\nabla f_1(x) - \nabla f_2(x)\|_2 \\
&\le |f_1(x) - f_2(x)| + 2M\lambda \|\nabla f_1(x) - \nabla f_2(x)\|_2.
\end{aligned}
$$

Thus $g_f$ is Lipschitz in $(f, \lambda \nabla f)$, with Lipschitz constant depending only on $M$.

Next we relate the shifted localization

$$J_\lambda(f) - J_\lambda(f_\lambda) + J_\lambda(f_0) - J_\lambda(f_\lambda) \le r$$

to a radius constraint around $f_0$. By Lemma 3.3, for all $f \in \mathcal{H}_M$,

$$J_\lambda(f) - J_\lambda(f_\lambda) \ge \frac{c_{\min}}{2} \|f - f_\lambda\|_{L^2(\mu)}^2 + \lambda \|\nabla(f - f_\lambda)\|_{L^2(\mu)}^2.$$

Similarly,

$$J_\lambda(f_0) - J_\lambda(f_\lambda) \ge \frac{c_{\min}}{2} \|f_0 - f_\lambda\|_{L^2(\mu)}^2 + \lambda \|\nabla(f_0 - f_\lambda)\|_{L^2(\mu)}^2.$$

Therefore, if $f \in \mathcal{U}_r$, then

$$\|f - f_0\|_{L^2(\mu)}^2 \le 2\|f - f_\lambda\|_{L^2(\mu)}^2 + 2\|f_0 - f_\lambda\|_{L^2(\mu)}^2 \le \frac{4}{c_{\min}} r,$$

and

$$\|\lambda \nabla(f - f_0)\|_{L^2(\mu)}^2 \le 2\lambda^2 \|\nabla(f - f_\lambda)\|_{L^2(\mu)}^2 + 2\lambda^2 \|\nabla(f_0 - f_\lambda)\|_{L^2(\mu)}^2 \le 2\lambda r \le 2r,$$

where we used $0 < \lambda < 1$.

Applying the Ledoux–Talagrand contraction lemma (Ledoux & Talagrand, 1991, Theorem 4.12) separately to the $p$-sample block and the $q$-sample block, and using $\sup_f(a_f + b_f) \le \sup_f a_f + \sup_f b_f$, we first have

$$
\begin{aligned}
\mathfrak{R}_n(\mathcal{G}_r; \mathcal{D}) &= \mathbb{E}_\sigma \sup_{f \in \mathcal{U}_r} \left\{ \frac{1}{2n} \sum_{i=1}^n \sigma_i^p g_f(X_i^p, 0) + \frac{1}{2n} \sum_{i=1}^n \sigma_i^q g_f(X_i^q, 1) \right\} \\
&\le \mathbb{E}_{\sigma^p} \sup_{f \in \mathcal{U}_r} \frac{1}{2n} \sum_{i=1}^n \sigma_i^p g_f(X_i^p, 0) + \mathbb{E}_{\sigma^q} \sup_{f \in \mathcal{U}_r} \frac{1}{2n} \sum_{i=1}^n \sigma_i^q g_f(X_i^q, 1).
\end{aligned}
$$

The same derivative-augmented covering-number argument as in Lemma A.26 of (Lu et al., 2021) gives, for all $r \gtrsim n^{-2}$,

$$\begin{aligned}
\Re_n(\mathcal{G}_r; \mathcal{D}) &\lesssim \Re_n(\{f - f_0 : f \in \mathcal{U}_r\}; \mathcal{D}) \\
&\quad + \Re_n(\{\lambda \nabla f - \lambda \nabla f_0 : f \in \mathcal{U}_r\}; \mathcal{D}) \\
&\lesssim \frac{1}{n} + \frac{1}{\sqrt{n}} \int_{1/n}^{C\sqrt{r}} \sqrt{S\Big[\log(\delta^{-1}) + 3^L \log(WB)\Big]} \, d\delta \\
&\lesssim \frac{1}{n} + \sqrt{\frac{Sr}{n}\Big[\log n + 3^L \log(WB)\Big]} \\
&\lesssim \frac{1}{n} + \sqrt{\frac{S\,3^L r}{n} \log(BWn)}.
\end{aligned}$$

The first inequality follows from the Ledoux–Talagrand contraction lemma applied to the pointwise Lipschitz bound

$$|g_{f_1}(x,y) - g_{f_2}(x,y)| \leq |f_1(x) - f_2(x)| + 2M\lambda \|\nabla f_1(x) - \nabla f_2(x)\|_2.$$

Indeed, the logistic cross-entropy loss is 1-Lipschitz in the logit argument, and

$$\Big|\|\nabla f_1(x)\|_2^2 - \|\nabla f_2(x)\|_2^2\Big| \leq \big(\|\nabla f_1(x)\|_2 + \|\nabla f_2(x)\|_2\big)\|\nabla f_1(x) - \nabla f_2(x)\|_2 \leq 2M\|\nabla f_1(x) - \nabla f_2(x)\|_2.$$

Thus the Rademacher complexity of the shifted loss class is reduced to the complexities of the localized function class and the localized $\lambda$-scaled gradient class.

The second inequality is the Dudley entropy integral after using the shifted localization

$$J_\lambda(f) - J_\lambda(f_\lambda) + J_\lambda(f_0) - J_\lambda(f_\lambda) \leq r.$$

By the strong convexity of $J_\lambda$, for $0 < \lambda \leq 1$,

$$\|f - f_0\|_{L^2(\mu)}^2 + \|\lambda \nabla(f - f_0)\|_{L^2(\mu)}^2 \lesssim r.$$

Hence both localized classes have $L^2(\mu)$-radius of order $\sqrt{r}$.

Since $\mu = (p + q)/2$, for every measurable $h$,

$$\|h\|_{L^2(p)}^2 \leq 2\|h\|_{L^2(\mu)}^2, \qquad \|h\|_{L^2(q)}^2 \leq 2\|h\|_{L^2(\mu)}^2.$$

Therefore the same $C\sqrt{r}$ localized radius is valid for the two empirical blocks $\{X_i^p\}_{i=1}^n$ and $\{X_i^q\}_{i=1}^n$.

Combining this radius bound with the derivative-augmented covering-number estimate for sparse ReLU$^3$ networks gives

$$\log \mathcal{N}(\delta, \mathcal{F}, \|\cdot\|_\infty) \vee \log \mathcal{N}(\delta, \nabla \mathcal{F}, \|\cdot\|_\infty) \lesssim S\Big[\log(\delta^{-1}) + 3^L \log(WB)\Big].$$

Since the empirical $L^2$-metric is dominated by the sup-norm, Dudley's integral yields the displayed entropy integral with upper limit $C\sqrt{r}$.

Finally, evaluating the integral gives

$$\frac{1}{\sqrt{n}} \int_{1/n}^{C\sqrt{r}} \sqrt{S\Big[\log(\delta^{-1}) + 3^L \log(WB)\Big]} \, d\delta \lesssim \sqrt{\frac{Sr}{n}\Big[\log n + 3^L \log(WB)\Big]},$$

which is further bounded by

$$\sqrt{\frac{S\,3^L r}{n} \log(BWn)}.$$

This is exactly the sub-root function $\phi$ stated in Theorem 3.5:

$$\phi(r) := C_0 \left[\frac{1}{n} + \sqrt{\frac{S\,3^L r}{n} \log(BWn)}\right].$$

This function is sub-root: it is nonnegative, nondecreasing, and $\phi(r)/\sqrt{r}$ is nonincreasing on $r > 0$. Consequently, its critical radius satisfies

$$r^\star \lesssim \frac{1}{n} + \frac{S\,3^L \log(BWn)}{n}.$$

In particular, when $L = \mathcal{O}(1)$, $S = \mathcal{O}(N)$, and $B, W = \mathcal{O}(N)$, we obtain

$$r^\star \lesssim \frac{N(\log N + \log n)}{n}.$$

Step 2: Peeling and the normalized empirical process. The local Rademacher bound in Step 1 controls the class only on each localized shell

$$\mathcal{G}_s = \{g_f : J_\lambda(f) - J_\lambda(f_\lambda) + J_\lambda(f_0) - J_\lambda(f_\lambda) \le s\}.$$

Here and below, $g_f$ denotes the shifted loss defined in (34). To turn these shell-wise bounds into a uniform bound over the whole class $\mathcal{F}$, we introduce the normalized class

$$\overline{\mathcal{G}}_r := \left\{ \widehat{g}_f : \widehat{g}_f(x,y) = \frac{g_f(x,y)}{J_\lambda(f) - J_\lambda(f_\lambda) + J_\lambda(f_0) - J_\lambda(f_\lambda) + r}, \ f \in \mathcal{F} \right\}.$$

The numerator uses the same shifted loss $g_f$ from (34). The denominator is positive because both excess risks are nonnegative and $r > 0$. The role of the denominator is to rescale each function according to its own local excess radius. Thus, functions lying farther from the population minimizer are placed in larger shells and are normalized more strongly.

Applying the Peeling Lemma (Lemma A.7 in (Lu et al., 2021)), with the localization functional

$$f \mapsto J_\lambda(f) - J_\lambda(f_\lambda) + J_\lambda(f_0) - J_\lambda(f_\lambda),$$

and using the sub-root bound

$$\mathfrak{R}_n(\mathcal{G}_s; \mathcal{D}) \le \phi(s),$$

we obtain

$$\mathfrak{R}_n(\overline{\mathcal{G}}_r; \mathcal{D}) \le \frac{4\phi(r)}{r}.$$

In words, peeling converts the local Rademacher bounds on the shells

$$\{J_\lambda(f) - J_\lambda(f_\lambda) + J_\lambda(f_0) - J_\lambda(f_\lambda) \le s\}$$

into a single Rademacher bound for the normalized global class.

We now pass from the normalized loss class to the centered normalized empirical process. Define

$$\widetilde{\mathcal{G}}_r := \left\{ \widetilde{g}_f : \Omega \times \{0,1\} \to \mathbb{R} \ \middle| \ \widetilde{g}_f(x,y) = \frac{\overline{P}g_f - g_f(x,y)}{J_\lambda(f) - J_\lambda(f_\lambda) + J_\lambda(f_0) - J_\lambda(f_\lambda) + r}, \ f \in \mathcal{F} \right\}.$$

This is the centered version of the normalized shifted loss in (34). Let $\{X_i^{p'}\}_{i=1}^n$ and $\{X_i^{q'}\}_{i=1}^n$ be auxiliary independent copies, independent of the data, with $X_i^{p'} \sim p$ and $X_i^{q'} \sim q$. By the Symmetrization Lemma (Lemma A.3 in (Lu et al., 2021)) applied separately to the $p$-sample and $q$-sample blocks and then summed,

$$\sup_{\widetilde{g} \in \widetilde{\mathcal{G}}_r} \mathbb{E} \left[ \frac{1}{2n} \sum_{i=1}^n \widetilde{g}(X_i^{p'}, 0) + \frac{1}{2n} \sum_{i=1}^n \widetilde{g}(X_i^{q'}, 1) \right] \le 2\mathfrak{R}_n(\overline{\mathcal{G}}_r; \mathcal{D})$$

$$\le \frac{8\phi(r)}{r}.$$

Therefore,

$$\sup_{\widetilde{g} \in \widetilde{\mathcal{G}}_r} \mathbb{E} \left[ \frac{1}{2n} \sum_{i=1}^n \widetilde{g}(X_i^{p'}, 0) + \frac{1}{2n} \sum_{i=1}^n \widetilde{g}(X_i^{q'}, 1) \right] \le \frac{8\phi(r)}{r}. \tag{36}$$

Step 3: Verifying the Talagrand conditions. For any $f \in \mathcal{F}$, since both $f$ and $f_0$ lie in the clipped and gradient-bounded sieve,

$$\|f\|_\infty, \|f_0\|_\infty, \|\nabla f\|_\infty, \|\nabla f_0\|_\infty \leq M.$$

Hence, for every $(x, y)$,

$$\begin{aligned} |g_f(x, y)| &\leq |\ell_{\mathrm{CE}}(y, f(x))| + |\ell_{\mathrm{CE}}(y, f_0(x))| + \lambda \left( \|\nabla f(x)\|_2^2 + \|\nabla f_0(x)\|_2^2 \right) \\ &\leq 2(\log(1 + e^M) + M) + 2\lambda M^2 \\ &\leq 2(\log(1 + e^M) + M) + 2M^2 =: M_\infty. \end{aligned}$$

Therefore

$$\|\widetilde{g}_f\|_\infty \leq \frac{2M_\infty}{r} =: \beta_{\mathrm{Tal}}. \tag{37}$$

We next bound the second moment. From the Lipschitz estimate in Step 1,

$$|g_f(x, y)| \leq |f(x) - f_0(x)| + 2M\lambda \|\nabla f(x) - \nabla f_0(x)\|_2.$$

Thus

$$\overline{P} g_f^2 \leq C_0 \left( \|f - f_0\|_{L^2(\mu)}^2 + \|\lambda\nabla(f - f_0)\|_{L^2(\mu)}^2 \right).$$

Using the strong-convexity bounds from Step 1,

$$\overline{P} g_f^2 \leq C_0 \left( J_\lambda(f) - J_\lambda(f_\lambda) + J_\lambda(f_0) - J_\lambda(f_\lambda) \right).$$

Consequently,

$$\overline{P} \widetilde{g}_f^2 = \frac{\overline{P}[(g_f - \overline{P} g_f)^2]}{(J_\lambda(f) - J_\lambda(f_\lambda) + J_\lambda(f_0) - J_\lambda(f_\lambda) + r)^2} \leq \frac{\overline{P} g_f^2}{(J_\lambda(f) - J_\lambda(f_\lambda) + J_\lambda(f_0) - J_\lambda(f_\lambda) + r)^2} \leq \frac{C_0}{r} =: \sigma_{\mathrm{Tal}}^2.$$

Moreover,

$$\overline{P} \widetilde{g}_f = 0.$$

Step 4: Talagrand concentration and choosing the radius. Apply Talagrand's inequality to the independent product variables $(X_i^p, X_i^q)$, with the normalized centered function

$$(x^p, x^q) \mapsto \frac{1}{2} \widetilde{g}(x^p, 0) + \frac{1}{2} \widetilde{g}(x^q, 1).$$

With probability at least $1 - e^{-t}$,

$$\begin{aligned} \sup_{\widetilde{g} \in \widetilde{\mathcal{G}}_r} \overline{P}_n \widetilde{g} &\leq 2 \sup_{\widetilde{g} \in \widetilde{\mathcal{G}}_r} \mathbb{E} \left[ \frac{1}{2n} \sum_{i=1}^n \widetilde{g}(X_i^{p\prime}, 0) + \frac{1}{2n} \sum_{i=1}^n \widetilde{g}(X_i^{q\prime}, 1) \right] + \sqrt{\frac{2t\sigma_{\mathrm{Tal}}^2}{n}} + \frac{2t\beta_{\mathrm{Tal}}}{n} \\ &\leq \frac{16\phi(r)}{r} + C_0 \sqrt{\frac{t}{nr}} + \frac{C_0 t}{nr} =: \psi(r). \end{aligned}$$

Choose

$$r_0 := C_0' \max \left\{ r^\star, \frac{t}{n} \right\},$$

where $C_0'$ is chosen so that

$$C_0' \geq \max \left\{ 128^2, 36C_0^2, 6C_0 \right\}.$$

Since $\phi$ is sub-root and $r^\star$ is its critical radius, $r \mapsto \phi(r)/\sqrt{r}$ is nonincreasing and $\phi(r^\star) = r^\star$. Hence $r_0 \geq C_0' r^\star$ implies

$$\frac{\phi(r_0)}{r_0} \leq \left( \frac{r^\star}{r_0} \right)^{1/2} \leq \frac{1}{\sqrt{C_0'}}.$$

Thus the first term satisfies

$$\frac{16\phi(r_0)}{r_0} \le \frac{1}{8}.$$

Moreover, $r_0 \ge C_0' t/n$, and therefore

$$C_0\sqrt{\frac{t}{nr_0}} \le \frac{1}{6}, \qquad \frac{C_0 t}{nr_0} \le \frac{1}{6},$$

where the two inequalities follow respectively from $C_0' \ge 36C_0^2$ and $C_0' \ge 6C_0$.

Therefore,

$$\psi(r_0) \le \frac{1}{8} + \frac{1}{6} + \frac{1}{6} < \frac{1}{2}.$$

Step 5: Concluding the shifted oracle bound. Pick $r = r_0$. On the event above, for every $f \in \mathcal{F}$,

$$\frac{\left[J_\lambda(f) - \widehat{J}_{\lambda,\mathcal{D}}(f)\right] - \left[J_\lambda(f_0) - \widehat{J}_{\lambda,\mathcal{D}}(f_0)\right]}{J_\lambda(f) - J_\lambda(f_\lambda) + J_\lambda(f_0) - J_\lambda(f_\lambda) + r_0} = \overline{P}_n\widetilde{g}_f \le \frac{1}{2},$$

which implies

$$\left[J_\lambda(f) - \widehat{J}_{\lambda,\mathcal{D}}(f)\right] - \left[J_\lambda(f_0) - \widehat{J}_{\lambda,\mathcal{D}}(f_0)\right] \le \frac{1}{2}\left[J_\lambda(f) - J_\lambda(f_\lambda)\right] + \frac{1}{2}\left[J_\lambda(f_0) - J_\lambda(f_\lambda)\right] + \frac{1}{2}r_0.$$

Now take $f = \widehat{f}_{\lambda,\mathcal{F}}$. By empirical optimality,

$$\widehat{J}_{\lambda,\mathcal{D}}(\widehat{f}_{\lambda,\mathcal{F}}) \le \widehat{J}_{\lambda,\mathcal{D}}(f_0).$$

Therefore

$$J_\lambda(\widehat{f}_{\lambda,\mathcal{F}}) - J_\lambda(f_\lambda) \le \left[J_\lambda(\widehat{f}_{\lambda,\mathcal{F}}) - \widehat{J}_{\lambda,\mathcal{D}}(\widehat{f}_{\lambda,\mathcal{F}})\right] - \left[J_\lambda(f_0) - \widehat{J}_{\lambda,\mathcal{D}}(f_0)\right] + J_\lambda(f_0) - J_\lambda(f_\lambda)$$

$$\le \frac{1}{2}\left[J_\lambda(\widehat{f}_{\lambda,\mathcal{F}}) - J_\lambda(f_\lambda)\right] + \frac{3}{2}\left[J_\lambda(f_0) - J_\lambda(f_\lambda)\right] + \frac{1}{2}r_0.$$

Rearranging gives

$$J_\lambda(\widehat{f}_{\lambda,\mathcal{F}}) - J_\lambda(f_\lambda) \le 3\left[J_\lambda(f_0) - J_\lambda(f_\lambda)\right] + r_0.$$

Since $r_0 \lesssim \max\{r^\star, t/n\}$, we conclude that

$$J_\lambda(\widehat{f}_{\lambda,\mathcal{F}}) - J_\lambda(f_\lambda) \lesssim J_\lambda(f_0) - J_\lambda(f_\lambda) + r^\star + \frac{t}{n}.$$

This proves the theorem. $\qquad\square$

## A.4. Proof of Theorem 3.1

*Proof.* Let

$$\mathcal{F} := \mathcal{F}_M(L_n, W_n, S_n, B_n).$$

By Proposition A.4, there exists

$$f_0 = f_{\mathcal{F}}^{\mathrm{app}} \in \mathcal{F}$$

such that

$$\|f_0 - f^\star\|_{H^1(\mu)}^2 \le C_{\mathrm{app}} N^{-\frac{2(s-1)}{d}}. \tag{38}$$

Define

$$\delta_N := N^{-\frac{s-1}{d}}.$$

Then

$$\|f_0 - f^\star\|_{L^2(\mu)} \le C_{\mathrm{app}}^{1/2}\delta_N, \qquad \|\nabla(f_0 - f^\star)\|_{L^2(\mu)} \le C_{\mathrm{app}}^{1/2}\delta_N.$$

We first bound the comparator excess $J_\lambda(f_0) - J_\lambda(f_\lambda)$. Since $f_\lambda$ minimizes $J_\lambda$, this quantity is nonnegative. Moreover,

$$J_\lambda(f_0) - J_\lambda(f_\lambda) \leq J_\lambda(f_0) - J_\lambda(f^\star) + J_\lambda(f^\star) - J_\lambda(f_\lambda).$$

Step 1: Bound $J_\lambda(f_0) - J_\lambda(f^\star)$. Since $f^\star$ minimizes the population cross-entropy risk and

$$\partial_u^2 \ell_{\mathrm{CE}}(y, u) = \sigma(u)(1 - \sigma(u)) \leq \frac{1}{4},$$

we have

$$L_{\mathrm{CE}}(f_0) - L_{\mathrm{CE}}(f^\star) \leq \frac{1}{8}\|f_0 - f^\star\|_{L^2(\mu)}^2 \leq \frac{1}{8}C_{\mathrm{app}}\delta_N^2.$$

For the Sobolev penalty, using

$$\|\nabla f_0\|_{L^\infty(\mu)} \leq M, \qquad \|\nabla f^\star\|_{L^\infty(\mu)} \leq M,$$

we obtain

$$\lambda \left| \|\nabla f_0\|_{L^2(\mu)}^2 - \|\nabla f^\star\|_{L^2(\mu)}^2 \right| \leq \lambda \int \left( \|\nabla f_0\|_2 + \|\nabla f^\star\|_2 \right) \|\nabla(f_0 - f^\star)\|_2 \, d\mu$$
$$\leq 2M\lambda\|\nabla(f_0 - f^\star)\|_{L^2(\mu)}$$
$$\leq 2MC_{\mathrm{app}}^{1/2}\lambda\delta_N.$$

Therefore

$$J_\lambda(f_0) - J_\lambda(f^\star) \leq \frac{1}{8}C_{\mathrm{app}}\delta_N^2 + 2MC_{\mathrm{app}}^{1/2}\lambda\delta_N. \tag{39}$$

Step 2: Bound $J_\lambda(f^\star) - J_\lambda(f_\lambda)$. Since $f^\star$ minimizes $L_{\mathrm{CE}}$,

$$L_{\mathrm{CE}}(f^\star) - L_{\mathrm{CE}}(f_\lambda) \leq 0.$$

Thus

$$J_\lambda(f^\star) - J_\lambda(f_\lambda) \leq \lambda \left( \|\nabla f^\star\|_{L^2(\mu)}^2 - \|\nabla f_\lambda\|_{L^2(\mu)}^2 \right).$$

Then

$$\|\nabla f^\star\|_{L^2(\mu)}^2 - \|\nabla f_\lambda\|_{L^2(\mu)}^2 = -2\langle \nabla f^\star, \nabla(f_\lambda - f^\star)\rangle_{L^2(\mu)} - \|\nabla(f_\lambda - f^\star)\|_{L^2(\mu)}^2.$$

Hence

$$J_\lambda(f^\star) - J_\lambda(f_\lambda) \leq 2\lambda \left| \langle \nabla f^\star, \nabla(f_\lambda - f^\star)\rangle_{L^2(\mu)} \right|.$$

Using the same Green identity as in Lemma 3.4,

$$\left| \langle \nabla f^\star, \nabla(f_\lambda - f^\star)\rangle_{L^2(\mu)} \right| = \left| \langle \mathcal{K}f^\star, f_\lambda - f^\star\rangle_{L^2(\mu)} \right| \leq \|\mathcal{K}f^\star\|_{L^2(\mu)}\|f_\lambda - f^\star\|_{L^2(\mu)}.$$

By Lemma 3.4,

$$\|f_\lambda - f^\star\|_{L^2(\mu)} \leq \sqrt{\beta}\lambda.$$

Therefore

$$J_\lambda(f^\star) - J_\lambda(f_\lambda) \leq 2\|\mathcal{K}f^\star\|_{L^2(\mu)}\sqrt{\beta}\,\lambda^2. \tag{40}$$

Combining (39) and (40), we obtain

$$J_\lambda(f_0) - J_\lambda(f_\lambda) \leq \frac{1}{8}C_{\mathrm{app}}\delta_N^2 + 2MC_{\mathrm{app}}^{1/2}\lambda\delta_N + 2\|\mathcal{K}f^\star\|_{L^2(\mu)}\sqrt{\beta}\lambda^2. \tag{41}$$

Step 3: Apply the shifted oracle inequality. Apply Theorem 3.5 with $t = 2\log n$. With probability at least $1 - n^{-2}$,

$$J_\lambda(\widehat{f}_{\lambda,\mathcal{F}}) - J_\lambda(f_\lambda) \leq C_{\mathrm{or}} \left( J_\lambda(f_0) - J_\lambda(f_\lambda) + r^\star + \frac{\log n}{n} \right).$$

Using (41), we get

$$J_\lambda(\widehat{f}_{\lambda,\mathcal{F}}) - J_\lambda(f_\lambda) \leq C_{\mathrm{or}} \left[ \frac{1}{8} C_{\mathrm{app}} \delta_N^2 + 2M C_{\mathrm{app}}^{1/2} \lambda \delta_N + 2\|\mathcal{K}f^\star\|_{L^2(\mu)} \sqrt{\beta} \lambda^2 + r^\star + \frac{\log n}{n} \right]. \tag{42}$$

Step 4: Convert excess risk to $L^2$ and gradient errors. By Lemma 3.3, for any $f, g \in \mathcal{H}_M$,

$$J_\lambda(f) - J_\lambda(g) - DJ_\lambda(g)[f - g] \geq \frac{c_{\min}}{2} \|f - g\|_{L^2(\mu)}^2 + \lambda \|\nabla(f - g)\|_{L^2(\mu)}^2.$$

Since $f_\lambda$ is the population minimizer, the first-order optimality condition gives

$$DJ_\lambda(f_\lambda)[\widehat{f}_{\lambda,\mathcal{F}} - f_\lambda] \geq 0.$$

Therefore

$$\frac{c_{\min}}{2} \|\widehat{f}_{\lambda,\mathcal{F}} - f_\lambda\|_{L^2(\mu)}^2 + \lambda \|\nabla(\widehat{f}_{\lambda,\mathcal{F}} - f_\lambda)\|_{L^2(\mu)}^2 \leq J_\lambda(\widehat{f}_{\lambda,\mathcal{F}}) - J_\lambda(f_\lambda).$$

Combining this with (42), we obtain

$$\|\widehat{f}_{\lambda,\mathcal{F}} - f_\lambda\|_{L^2(\mu)}^2 \leq \frac{2C_{\mathrm{or}}}{c_{\min}} \left[ \frac{1}{8} C_{\mathrm{app}} \delta_N^2 + 2M C_{\mathrm{app}}^{1/2} \lambda \delta_N + 2\|\mathcal{K}f^\star\|_{L^2(\mu)} \sqrt{\beta} \lambda^2 + r^\star + \frac{\log n}{n} \right], \tag{43}$$

and

$$\|\nabla(\widehat{f}_{\lambda,\mathcal{F}} - f_\lambda)\|_{L^2(\mu)}^2 \leq \frac{C_{\mathrm{or}}}{\lambda} \left[ \frac{1}{8} C_{\mathrm{app}} \delta_N^2 + 2M C_{\mathrm{app}}^{1/2} \lambda \delta_N + 2\|\mathcal{K}f^\star\|_{L^2(\mu)} \sqrt{\beta} \lambda^2 + r^\star + \frac{\log n}{n} \right]. \tag{44}$$

Step 5: Add the regularization bias. By Lemma 3.4,

$$\|f_\lambda - f^\star\|_{L^2(\mu)}^2 \leq \beta \lambda^2, \qquad \|\nabla(f_\lambda - f^\star)\|_{L^2(\mu)}^2 \leq \beta \lambda.$$

Using the triangle inequality,

$$\|\widehat{f}_{\lambda,\mathcal{F}} - f^\star\|_{L^2(\mu)}^2 \leq 2\|\widehat{f}_{\lambda,\mathcal{F}} - f_\lambda\|_{L^2(\mu)}^2 + 2\|f_\lambda - f^\star\|_{L^2(\mu)}^2,$$

and

$$\|\nabla(\widehat{f}_{\lambda,\mathcal{F}} - f^\star)\|_{L^2(\mu)}^2 \leq 2\|\nabla(\widehat{f}_{\lambda,\mathcal{F}} - f_\lambda)\|_{L^2(\mu)}^2 + 2\|\nabla(f_\lambda - f^\star)\|_{L^2(\mu)}^2.$$

Therefore

$$\|\widehat{f}_{\lambda,\mathcal{F}} - f^\star\|_{H^1(\mu)}^2 \leq \left( \frac{4C_{\mathrm{or}}}{c_{\min}} + \frac{2C_{\mathrm{or}}}{\lambda} \right) \left[ \frac{1}{8} C_{\mathrm{app}} \delta_N^2 + 2M C_{\mathrm{app}}^{1/2} \lambda \delta_N + 2\|\mathcal{K}f^\star\|_{L^2(\mu)} \sqrt{\beta} \lambda^2 + r^\star + \frac{\log n}{n} \right]$$
$$+ 2\beta \lambda^2 + 2\beta \lambda. \tag{45}$$

Step 6: Choose $\lambda$ and $N$. The critical radius satisfies

$$r^\star \leq C_{\mathrm{rad}} \frac{N(\log N + \log n)}{n},$$

where $C_{\mathrm{rad}}$ depends only on the fixed sieve envelope $M$ and fixed depth $L = \mathcal{O}(1)$.

Choose

$$\lambda \asymp \delta_N = N^{-\frac{s-1}{d}}.$$

Since $c_{\min}$, $M$, $\|\mathcal{K}f^\star\|_{L^2(\mu)}$, $\beta$, $C_{\mathrm{app}}$, and $C_{\mathrm{or}}$ are independent of $n$, the leading terms in (45) are bounded by

$$C \left[ \delta_N + \frac{r^\star}{\delta_N} + \frac{\log n}{n\delta_N} \right].$$

Since $r^\star \gtrsim N/n$ up to logarithmic factors, the term

$$\frac{\log n}{n\delta_N}$$

is dominated by $r^\star/\delta_N$. Hence

$$\|\widehat{f}_{\lambda,\mathcal{F}} - f^\star\|_{H^1(\mu)}^2 \leq C\left[N^{-\frac{s-1}{d}} + \frac{N^{1+\frac{s-1}{d}}(\log N + \log n)}{n}\right].$$

Balancing the two terms gives

$$N^{-\frac{s-1}{d}} \asymp \frac{N^{1+\frac{s-1}{d}}}{n},$$

and therefore

$$N \asymp n^{\frac{d}{d+2s-2}}.$$

Consequently,

$$\lambda \asymp N^{-\frac{s-1}{d}} \asymp n^{-\frac{s-1}{d+2s-2}}.$$

Substituting this choice yields

$$\|\widehat{f}_{\lambda,\mathcal{F}} - f^\star\|_{H^1(\mu)}^2 \leq Cn^{-\frac{s-1}{d+2s-2}}\log n.$$

This proves the theorem. $\qquad\square$

## B. Proof of Theorem 4.1: Minimax lower bound

*Proof.* Step 1: Local packing on $\mathcal{C}_{\mathrm{pair}}$ with enough separation. We construct a finite subset of $\mathcal{C}_{\mathrm{pair}}$ whose log-density ratios have non-trivial $H^1$-separation.

Let $\eta : \mathbb{R} \to \mathbb{R}$ be the one-dimensional $C^\infty$ bump

$$\eta(t) := \begin{cases} \exp\left(-\frac{1}{t(1-t)}\right), & t \in (0,1), \\ 0, & t \notin (0,1), \end{cases}$$

and define

$$\varphi(x) := \prod_{i=1}^d \eta(x_i), \qquad x = (x_1, \ldots, x_d) \in \mathbb{R}^d.$$

Then $\varphi \in C^\infty(\mathbb{R}^d)$, $\varphi \geq 0$, $\nabla\varphi \not\equiv 0$, and $\mathrm{supp}(\varphi) \subset [0,1]^d$.

For an integer $m \geq 1$, choose points $\{x_j\}_{j\in\{1,\ldots,m\}^d} \subset \mathbb{R}^d$ such that the cubes $x_j + [0, (3m)^{-1}]^d$ are disjoint and contained in $(0,1)^d$. Define

$$\varphi_{m,j}(x) := \varphi\big(3m(x - x_j)\big).$$

By the change of variables $u = 3m(x - x_j)$ and standard Sobolev scaling,

$$\|\varphi_{m,j}\|_{L^2(\mu_0)}^2 \asymp m^{-d}, \qquad \|\nabla\varphi_{m,j}\|_{L^2(\mu_0)}^2 \asymp m^{2-d}, \qquad \|\varphi_{m,j}\|_{H^s(\mu_0)}^2 \asymp m^{2s-d}, \tag{46}$$

where the implicit constants depend only on $\varphi$ and the density of $\mu_0$.

By the Varshamov–Gilbert bound (Takezawa, 2005), there exist $\tau^{(1)}, \ldots, \tau^{(N)} \in \{0,1\}^{m^d}$ such that

$$N \geq 2^{m^d/8}, \qquad \|\tau^{(v)} - \tau^{(v')}\|_2^2 \geq \frac{m^d}{8} \quad \text{for all } v \neq v'.$$

Fix an amplitude $a_m > 0$. For each $v$, choose the unique scalar $c_v$ such that

$$\int_\Omega \sigma\left(a_m \sum_{j\in\{1,\ldots,m\}^d} \tau_j^{(v)}\varphi_{m,j}(x) - c_v\right)d\mu_0(x) = \frac{1}{2},$$

and set

$$f_v(x) := a_m \sum_{j\in\{1,\ldots,m\}^d} \tau_j^{(v)}\varphi_{m,j}(x) - c_v.$$

The scalar $c_v$ exists and is unique because the left-hand side is a continuous strictly decreasing function of $c_v$, with limits $1$ and $0$ as $c_v \to -\infty$ and $c_v \to +\infty$. Since the bump sum is nonnegative,

$$0 \le c_v \le a_m \sup_{x \in \Omega} \sum_{j \in \{1,\ldots,m\}^d} \tau_j^{(v)} \varphi_{m,j}(x) \lesssim a_m.$$

Writing the constant density of $\mu_0$ as $\rho_0 = |\Omega|^{-1}$, define

$$q_v(x) := 2\sigma(f_v(x))\rho_0, \qquad p_v(x) := 2(1 - \sigma(f_v(x)))\rho_0.$$

Then

$$\int_\Omega q_v(x)\,dx = 1, \qquad \int_\Omega p_v(x)\,dx = 1, \qquad \frac{p_v(x) + q_v(x)}{2} = \rho_0,$$

and

$$\log \frac{q_v(x)}{p_v(x)} = f_v(x).$$

We next verify that $(p_v, q_v) \in \mathcal{C}_{\mathrm{pair}}$. Since the supports of $\varphi_{m,j}$ are disjoint,

$$\sup_{x \in \Omega} |f_v(x)| \lesssim a_m, \tag{47}$$

$$\|f_v\|_{H^s(\mu_0)}^2 \lesssim a_m^2 \sum_{j \in \{1,\ldots,m\}^d} \|\varphi_{m,j}\|_{H^s(\mu_0)}^2 + c_v^2 \lesssim a_m^2 m^{2s} + a_m^2. \tag{48}$$

Choosing $a_m \asymp m^{-s}$ with a sufficiently small implicit constant ensures that $\|f_v\|_{H^s(\mu_0)} < \infty$ and $\sup_{x \in \Omega} |f_v(x)| \le M$. Since $f_v$ is bounded, smooth, and constant in a neighborhood of $\partial\Omega$, it satisfies condition 3 in Assumption 2.1. Moreover, $p_v$ and $q_v$ are strictly positive and $C^2$, and $\partial_n f_v = 0$ on $\partial\Omega$. Since $(p_v + q_v)/2 = \rho_0$, the Neumann condition also holds.

For $v \ne v'$, disjoint supports and (46) give

$$\|\nabla f_v - \nabla f_{v'}\|_{L^2(\mu_0)}^2 = a_m^2 \sum_{j \in \{1,\ldots,m\}^d} \left(\tau_j^{(v)} - \tau_j^{(v')}\right)^2 \|\nabla \varphi_{m,j}\|_{L^2(\mu_0)}^2$$

$$\gtrsim a_m^2 m^d m^{2-d} \asymp m^{-2(s-1)}.$$

Thus, for some constant $c > 0$,

$$\|f_v - f_{v'}\|_{H^1(\mu_0)}^2 \ge c\, m^{-2(s-1)}, \qquad v \ne v'. \tag{49}$$

Step 2: KL upper bound for the fixed source–target observation law. Under the fixed source–target design, candidate $v$ induces $p_v^n \otimes q_v^n$.

For two candidates $v$ and $v'$,

$$\mathrm{KL}(p_v^n \otimes q_v^n \| p_{v'}^n \otimes q_{v'}^n) = n\mathrm{KL}(p_v \| p_{v'}) + n\mathrm{KL}(q_v \| q_{v'})$$

$$= 2n \int_\Omega \mathrm{KL}(\mathrm{Bern}(\sigma(f_v(x))) \,\|\, \mathrm{Bern}(\sigma(f_{v'}(x)))) \, d\mu_0(x).$$

For each fixed $x$,

$$\mathrm{KL}(\mathrm{Bern}(\sigma(f_v(x))) \,\|\, \mathrm{Bern}(\sigma(f_{v'}(x))))$$

$$= \mathbb{E}_{Y \sim \mathrm{Bern}(\sigma(f_v(x)))}[\ell_{\mathrm{CE}}(Y, f_{v'}(x)) - \ell_{\mathrm{CE}}(Y, f_v(x))].$$

The derivative of $u \mapsto \mathbb{E}_{Y \sim \mathrm{Bern}(\sigma(f_v(x)))}[\ell_{\mathrm{CE}}(Y, u)]$ vanishes at $u = f_v(x)$, and its second derivative is $\sigma(u)(1 - \sigma(u))$. Since $\|f_v\|_\infty \le M$ uniformly in $v$, Taylor's theorem gives

$$\mathrm{KL}(p_v^n \otimes q_v^n \| p_{v'}^n \otimes q_{v'}^n) \lesssim n \|f_v - f_{v'}\|_{L^2(\mu_0)}^2. \tag{50}$$

Moreover,

$$\|f_v - f_{v'}\|_{L^2(\mu_0)}^2 \lesssim a_m^2 \left( \sum_{j \in \{1,\ldots,m\}^d} |\tau_j^{(v)} - \tau_j^{(v')}|^2 \right) m^{-d} + |c_v - c_{v'}|^2$$

$$\lesssim a_m^2 \asymp m^{-2s}. \tag{51}$$

Combining (50) and (51),

$$\mathrm{KL}(p_v^n \otimes q_v^n \| p_{v'}^n \otimes q_{v'}^n) \lesssim n\,m^{-2s}. \tag{52}$$

Step 3: Local Fano reduction and minimax lower bound. Let $V$ be uniformly distributed on $\{1,\dots,N\}$. Conditional on $V = v$, draw $\{X_i^p\}_{i=1}^n \overset{\text{i.i.d.}}{\sim} p_v$ and $\{X_i^q\}_{i=1}^n \overset{\text{i.i.d.}}{\sim} q_v$, independently. Any estimator $\widehat{f} = \psi(\{X_i^p\}_{i=1}^n, \{X_i^q\}_{i=1}^n)$ induces the testing rule

$$\widehat{V} := \underset{1 \le v \le N}{\arg\min} \left\| \psi(\{X_i^p\}_{i=1}^n, \{X_i^q\}_{i=1}^n) - f_v \right\|_{H^1(\mu_0)}.$$

By (49), if $\widehat{V} \ne V$, then

$$\|\widehat{f} - f_V\|_{H^1(\mu_0)} \ge \frac{1}{2} \|f_V - f_{\widehat{V}}\|_{H^1(\mu_0)} \gtrsim m^{-(s-1)}.$$

Therefore

$$\sup_{1 \le v \le N} \mathbb{E}_{p_v^n \otimes q_v^n}\left[ \|\widehat{f} - f_v\|_{H^1(\mu_0)}^2 \right] \gtrsim m^{-2(s-1)} \mathbb{P}(\widehat{V} \ne V). \tag{53}$$

It remains to lower bound $\mathbb{P}(\widehat{V} \ne V)$. Choose $m \asymp n^{1/(2s+d)}$, with a sufficiently large implicit constant. Then (52) gives

$$\mathrm{KL}(p_v^n \otimes q_v^n \| p_{v'}^n \otimes q_{v'}^n) \lesssim m^d.$$

Since $\log N \asymp m^d$, the local Fano inequality (Takezawa, 2005) gives

$$\mathbb{P}(\widehat{V} \ne V) \ge 1 - \frac{I(V; \{X_i^p\}_{i=1}^n, \{X_i^q\}_{i=1}^n) + \log 2}{\log N}$$

$$\ge 1 - \frac{N^{-2} \sum_{v,v'} \mathrm{KL}(p_v^n \otimes q_v^n \| p_{v'}^n \otimes q_{v'}^n) + \log 2}{\log N} \ge \frac{1}{2},$$

where the last inequality holds by taking the implicit constant in $a_m \asymp m^{-s}$ sufficiently small. Combining this bound with (53) and taking the infimum over all estimators $\psi$,

$$\inf_{\psi} \sup_{1 \le v \le N} \mathbb{E}_{p_v^n \otimes q_v^n}\left[ \|\psi(\{X_i^p\}_{i=1}^n, \{X_i^q\}_{i=1}^n) - f_v\|_{H^1(\mu_0)}^2 \right] \gtrsim n^{-\frac{2(s-1)}{2s+d}}.$$

Since all $(p_v, q_v)$ belong to $\mathcal{C}_{\mathrm{pair}}$, the same lower bound holds for the minimax risk over $\mathcal{C}_{\mathrm{pair}}$. $\qquad\square$

# C. Proof for the extension to diffusion models

Throughout this appendix, $\alpha_t$ denotes the VP coefficient, and we write

$$\kappa := \frac{s-1}{d+1+2s-2}.$$

for the nonparametric rate exponent. In this section we prove Theorem 5.2. We work on the bounded time-space cylinder $[t_0, T] \times B_{2R}$, where

$$B_{2R} := \{x \in \mathbb{R}^d : \|x\| \le 2R\},$$

and use a projection-rescaling argument so that the neural network is trained on the fixed compact spatial domain $B_1$.

## C.1. Notation

Throughout Appendix C, for each truncation radius $R \ge 1$, we use the $R$-dependent clipped-gradient time-space class

$$\mathcal{F}_R := \mathcal{F}_{M_R}^{(d+1)}(L, W, S, B), \qquad M_R := C_M(1+R),$$

on the fixed input domain

$$[t_0, T] \times B_1 \subset \mathbb{R}^{d+1}.$$

The input variable is $(t, \bar{x})$, and time is treated as an ordinary network input. By definition of $\mathcal{F}_R$,

$$\|f\|_{L^\infty([t_0,T] \times B_1)} \le M_R, \qquad \|\nabla_{\bar{x}} f\|_{L^\infty([t_0,T] \times B_1)} \le M_R, \qquad f \in \mathcal{F}_R.$$

**Time-dependent label-augmented model and truncated population risk.** For each $t \in [t_0, T]$, define a joint distribution $\mu_t^{x,y}$ on $\mathbb{R}^d \times \{0, 1\}$ by

$$Y \sim \text{Bernoulli}(1/2), \qquad X \mid (Y = 1) \sim q_t, \qquad X \mid (Y = 0) \sim p_t. \tag{54}$$

We write

$$\eta_t(x) := \mathbb{P}(Y = 1 \mid X = x) = \frac{q_t(x)}{p_t(x) + q_t(x)}.$$

Recall the pointwise $H^1$-loss density in (15). For any function $h : [t_0, T] \times \mathbb{R}^d \to \mathbb{R}$, define the truncated population functional on $[t_0, T] \times B_{2R}$ by

$$J_{\lambda, 2R}(h) := \int_{t_0}^{T} \mathbb{E}_{(X,Y) \sim \mu_t^{x,y}} \Big[ \mathbf{1}\{X \in B_{2R}\} \Big( \ell_{\text{CE}}(Y, h(t, X)) + \lambda \| \nabla_x h(t, X) \|^2 \Big) \Big] \, dt. \tag{55}$$

**Projection-rescaling and pulled-back class.** Recall that the projected-rescaled input is $\bar{x} = \text{Proj}_{2R}(x)/(2R) \in B_1$. For a function $f : [t_0, T] \times B_1 \to \mathbb{R}$, define its pull-back to $[t_0, T] \times B_{2R}$ by

$$(\pi_R f)(t, x) := f\Big(t, \frac{x}{2R}\Big), \qquad (t, x) \in [t_0, T] \times B_{2R}. \tag{56}$$

Then

$$\nabla_x (\pi_R f)(t, x) = \frac{1}{2R} \nabla_{\bar{x}} f\Big(t, \frac{x}{2R}\Big).$$

Similarly, for the rescaled target $f_R^\star(t, \bar{x}) := f^\star(t, 2R\bar{x})$, we have, for $(t, x) \in [t_0, T] \times B_{2R}$,

$$f^\star(t, x) = f_R^\star\Big(t, \frac{x}{2R}\Big), \qquad \nabla_x f^\star(t, x) = \frac{1}{2R} \nabla_{\bar{x}} f_R^\star\Big(t, \frac{x}{2R}\Big).$$

Define the pulled-back network class

$$\mathcal{F}_{2R} := \{ \pi_R f : f \in \mathcal{F}_R \}.$$

By the envelope definition of $\mathcal{F}_R$, every $h = \pi_R f \in \mathcal{F}_{2R}$ satisfies

$$\|h\|_{L^\infty([t_0,T] \times B_{2R})} \le M_R, \qquad \|\nabla_x h\|_{L^\infty([t_0,T] \times B_{2R})} \le \frac{M_R}{2R} \le C_M, \qquad R \ge 1.$$

**Norms.** Set $I := [t_0, T]$. On $I \times B_{2R}$, unless otherwise stated, we use the $\mu_t$-weighted spatial energy norm

$$\|g\|_{L^2(I; H^1(B_{2R}, \mu_t))}^2 := \int_I \int_{B_{2R}} \Big( |g(t, x)|^2 + \|\nabla_x g(t, x)\|^2 \Big) \, d\mu_t(x) \, dt.$$

Here $H^1(B_{2R}, \mu_t)$ is only in the spatial variable $x$.

We also use its $L^2$-part

$$\|g\|_{L^2(I \times B_{2R}, \mu_t)}^2 := \int_I \int_{B_{2R}} |g(t, x)|^2 \, d\mu_t(x) \, dt.$$

Since $p_t$ and $q_t$ are Gaussian convolutions, $\rho_t = (p_t + q_t)/2$ is smooth and strictly positive on the compact cylinder $I \times B_{2R}$; hence $\| \cdot \|_{L^2(I; H^1(B_{2R}, \mu_t))}$ is equivalent, for fixed $R$, to the corresponding Lebesgue spatial energy norm on $I \times B_{2R}$. On the fixed cylinder $I \times B_1$, we use the Lebesgue norms

$$\|u\|_{L^2(I; H^1(B_1))}^2 := \int_I \int_{B_1} \Big( |u(t, \bar{x})|^2 + \|\nabla_{\bar{x}} u(t, \bar{x})\|^2 \Big) \, d\bar{x} \, dt,$$

and, for integer $s \ge 1$,

$$\|u\|_{H^s(I \times B_1)}^2 := \sum_{a + |\gamma| \le s} \int_I \int_{B_1} \big| \partial_t^a D_{\bar{x}}^\gamma u(t, \bar{x}) \big|^2 \, d\bar{x} \, dt.$$

Here $L^2(I; H^1(B_1))$ is spatial Sobolev only in $\bar{x}$, while $H^s(I \times B_1)$ is the full time-space Sobolev norm of order $s$ on the same fixed cylinder.

For the rescaled target $f_R^\star(t, \bar{x}) = f^\star(t, 2R\bar{x})$, define

$$\mathcal{A}_s(R) := (2R)^d \|f_R^\star\|_{H^s(I \times B_1)}^2.$$

**Bounded Sobolev classes on truncated cylinders.** For $R > 0$ and $M > 0$, define the bounded Sobolev class on $[t_0, T] \times B_{2R}$ by

$$\mathcal{H}_M([t_0,T]\times B_{2R}) := \left\{ h : [t_0,T]\times B_{2R} \to \mathbb{R} \,\middle|\, \|h\|_{L^2(I;H^1(B_{2R},\mu_t))} < \infty, \ |h(t,x)| \leq M \text{ for a.e. } (t,x) \in [t_0,T]\times B_{2R} \right\}. \tag{57}$$

## C.2. Uniform bounds on the pulled-back class and auxiliary lemmas

We first collect the Gaussian-convolution bounds.

**Lemma C.1** (Basic consequences of the regular VP assumption). *Suppose Assumption 5.1 holds. Set $R_1 := \omega_0 R_0$. Then, if $X_0 \sim p_0$ and $Y_0 \sim q_0$, the laws of $\alpha_t X_0$ and $\alpha_t Y_0$ are supported in $B_{R_1}$.*

*Moreover, there exist constants $C, c, K < \infty$, depending only on*

$$d, s, t_0, T, R_0, \underline{\sigma}, \overline{\sigma}, \omega_0,$$

*such that the following bounds hold uniformly over $t \in [t_0, T]$.*

*First, the mixture measure has sub-Gaussian tails: for every $u \geq 0$,*

$$\mu_t(\{x : \|x\| > u\}) \leq C \exp\{-c(u - R_1)_+^2\}. \tag{58}$$

*More generally, for every $m \geq 0$,*

$$\int_{\{\|x\|>u\}} (1 + \|x\|)^m \, d\mu_t(x) \leq C_m(1 + u)^m \exp\{-c_m(u - R_1)_+^2\}. \tag{59}$$

*Second, the log-density ratio has at most linear growth:*

$$|f^\star(t,x)| \leq C(1 + \|x\|). \tag{60}$$

*Its spatial gradient is uniformly bounded:*

$$\|\nabla_x f^\star(t,x)\| \leq C. \tag{61}$$

*Finally, for every integer $a \geq 0$ and multi-index $b \in \mathbb{N}^d$ satisfying $a + |b| \leq s$*

$$\left| \partial_t^a D_x^b f^\star(t,x) \right| \leq C(1 + \|x\|)^K. \tag{62}$$

*Proof.* The support statement follows directly from the VP representation. If $X_0 \sim p_0$ and $Y_0 \sim q_0$, then $X_0, Y_0 \in B_{R_0}$ almost surely. Since

$$|\alpha_t| \leq \omega_0,$$

we have

$$\|\alpha_t X_0\| \leq \omega_0 R_0, \qquad \|\alpha_t Y_0\| \leq \omega_0 R_0 \quad \text{a.s.},$$

so both laws are supported in

$$B_{\omega_0 R_0} = B_{R_1}.$$

We first prove the tail bounds. If $Z_0$ is supported in $B_{R_0}$, then

$$Z_t = \alpha_t Z_0 + \sigma_t \xi$$

satisfies

$$\|Z_t\| \leq R_1 + \overline{\sigma}\|\xi\|.$$

Therefore

$$\mathbb{P}(\|Z_t\| > u) \leq \mathbb{P}\left( \|\xi\| > \frac{(u - R_1)_+}{\overline{\sigma}} \right) \leq C \exp\{-c(u - R_1)_+^2\}.$$

This holds for both $p_t$ and $q_t$, hence for their mixture $\mu_t$. The polynomial tail-moment bound (59) follows by integration by parts. Indeed, for any nonnegative random variable $W$,

$$\mathbb{E}\big[(1+W)^m \mathbf{1}\{W > u\}\big] \leq (1+u)^m \mathbb{P}(W > u) + m \int_u^\infty (1+r)^{m-1} \mathbb{P}(W > r)\, dr,$$

and applying the preceding sub-Gaussian tail bound gives (59), after adjusting constants.

Next we study the density ratio. Since $p_t$ is the law of $\alpha_t X_0 + \sigma_t \xi$ with $X_0 \sim p_0$ and $\xi \sim \mathcal{N}(0, I)$, we have the Gaussian mixture representation

$$p_t(x) = \int (2\pi\sigma_t^2)^{-d/2} \exp\left(-\frac{\|x - \alpha_t z\|^2}{2\sigma_t^2}\right) dp_0(z),$$

and similarly $q_t$ is represented by replacing $p_0$ with $q_0$. Expanding the square gives

$$p_t(x) = (2\pi\sigma_t^2)^{-d/2} \exp\left(-\frac{\|x\|^2}{2\sigma_t^2}\right) I_p(t, x),$$

where

$$I_p(t, x) := \int \exp\left(\frac{\alpha_t \langle x, z\rangle}{\sigma_t^2} - \frac{\alpha_t^2 \|z\|^2}{2\sigma_t^2}\right) dp_0(z).$$

Similarly,

$$q_t(x) = (2\pi\sigma_t^2)^{-d/2} \exp\left(-\frac{\|x\|^2}{2\sigma_t^2}\right) I_q(t, x),$$

where $I_q$ is defined with $q_0$. Hence the common Gaussian factor cancels in the log-density ratio:

$$f^\star(t, x) = \log I_q(t, x) - \log I_p(t, x).$$

Since $z \in B_{R_0}$, $\alpha_t$ is bounded, and $\sigma_t \geq \underline{\sigma}$, the exponent inside $I_p$ is bounded above and below by affine functions of $\|x\|$. Therefore

$$|\log I_p(t, x)| \leq C(1 + \|x\|), \qquad |\log I_q(t, x)| \leq C(1 + \|x\|),$$

which proves (60).

For the gradient, define the tilted probability measure

$$d\pi_{p,t,x}(z) := \frac{\exp\left(\frac{\alpha_t \langle x, z\rangle}{\sigma_t^2} - \frac{\alpha_t^2 \|z\|^2}{2\sigma_t^2}\right)}{I_p(t, x)}\, dp_0(z).$$

Then

$$\nabla_x \log I_p(t, x) = \frac{\alpha_t}{\sigma_t^2} \int z\, d\pi_{p,t,x}(z).$$

Since $\|z\| \leq R_0$, we obtain

$$\|\nabla_x \log I_p(t, x)\| \leq C.$$

The same bound holds for $I_q$. Therefore

$$\nabla_x f^\star(t, x) = \nabla_x \log I_q(t, x) - \nabla_x \log I_p(t, x)$$

is uniformly bounded, proving (61).

It remains to show (62). We prove it for $\log I_p$; the argument for $\log I_q$ is identical. Define

$$\Phi(t, x, z) := \frac{\alpha_t \langle x, z\rangle}{\sigma_t^2} - \frac{\alpha_t^2 \|z\|^2}{2\sigma_t^2}.$$

For every integer $a \geq 0$ and multi-index $b \in \mathbb{N}^d$ with $a + |b| \leq s$, Assumption 5.1, compactness of $[t_0, T]$, and $\|z\| \leq R_0$ imply

$$\left|\partial_t^a D_x^b \Phi(t, x, z)\right| \leq C_{a,b}(1 + \|x\|),$$

uniformly in $t \in [t_0, T]$ and $z \in B_{R_0}$. We now pass the derivative bound from $\Phi$ to $\log I_p$. Let $\nu = (a, b)$ be a time–space multi-index and write $D^\nu = \partial_t^a D_x^b$. Since $p_0$ is supported on $B_{R_0}$, the bound

$$|D^\eta \Phi(t, x, z)| \leq C_\eta (1 + \|x\|), \qquad 1 \leq |\eta| \leq s,$$

holds uniformly over $t \in [t_0, T]$ and $z \in B_{R_0}$. Moreover, on every compact set in $x$, the derivatives $D^\eta e^{\Phi(t,x,z)}$, $|\eta| \leq s$, are uniformly bounded on $[t_0, T] \times K \times B_{R_0}$. Hence differentiation under the integral is justified by dominated convergence.

For every $1 \leq |\nu| \leq s$, the multivariate Faà di Bruno formula gives

$$D^\nu I_p(t, x) = \int e^{\Phi(t,x,z)} P_\nu \Big( \{ D^\eta \Phi(t, x, z) : 1 \leq |\eta| \leq |\nu| \} \Big) \, dp_0(z),$$

where $P_\nu$ is a polynomial depending only on $\nu$. The preceding bound on the derivatives of $\Phi$ implies

$$\left| P_\nu \Big( \{ D^\eta \Phi(t, x, z) : 1 \leq |\eta| \leq |\nu| \} \Big) \right| \leq C_\nu (1 + \|x\|)^{K_\nu}.$$

Since $I_p(t, x) = \int e^{\Phi(t,x,z)} \, dp_0(z) > 0$, it follows directly that

$$\left| \frac{D^\nu I_p(t, x)}{I_p(t, x)} \right| \leq \frac{\int e^{\Phi(t,x,z)} C_\nu (1 + \|x\|)^{K_\nu} \, dp_0(z)}{\int e^{\Phi(t,x,z)} \, dp_0(z)} = C_\nu (1 + \|x\|)^{K_\nu}.$$

Applying the multivariate chain rule to $\log I_p$, for every $1 \leq |\nu| \leq s$, $D^\nu \log I_p$ is a finite sum of products of terms of the form

$$\frac{D^\eta I_p(t, x)}{I_p(t, x)}, \qquad 1 \leq |\eta| \leq |\nu|.$$

Therefore the ratio bound above yields

$$|D^\nu \log I_p(t, x)| \leq C_\nu (1 + \|x\|)^{K_\nu}, \qquad 1 \leq |\nu| \leq s.$$

Together with the zeroth-order bound for $\log I_p$ established above, we obtain

$$|D^\nu \log I_p(t, x)| \leq C_\nu (1 + \|x\|)^{K_\nu}, \qquad |\nu| \leq s.$$

The same argument applies to $\log I_q$. Since $f^\star = \log I_q - \log I_p$, enlarging $C$ and $K$ over the finitely many time–space multi-indices with order at most $s$ proves

$$\left| \partial_t^a D_x^b f^\star(t, x) \right| \leq C (1 + \|x\|)^K, \qquad a + |b| \leq s.$$

$\square$

**Global choice of the envelope constant.** Throughout Appendix C.2, the constant $C_M$ is a deterministic constant fixed once and for all, independently of $R$, $n$, $N$, and $\lambda$. We now specify its choice.

By Lemma C.1, there exists a constant $C_\star < \infty$, independent of $R$, such that, for all $R \geq 1$,

$$\sup_{(t,\bar{x}) \in [t_0, T] \times B_1} |f_R^\star(t, \bar{x})| + \sup_{(t,\bar{x}) \in [t_0, T] \times B_1} \|\nabla_{\bar{x}} f_R^\star(t, \bar{x})\| \leq C_\star (1 + R). \tag{63}$$

Indeed, $f_R^\star(t, \bar{x}) = f^\star(t, 2R\bar{x})$, and

$$\nabla_{\bar{x}} f_R^\star(t, \bar{x}) = 2R \, \nabla_x f^\star(t, 2R\bar{x}).$$

Thus the linear-growth bound for $f^\star$ and the uniform bound for $\nabla_x f^\star$ imply (63).

Let $C_{qi}$ denote the $W^{1,\infty}$-stability constant of the spline quasi-interpolation operator used in Proposition A.4. Since the approximation is performed on the fixed cylinder $[t_0, T] \times B_1 \subset \mathbb{R}^{d+1}$, $C_{qi}$ depends only on the dimension, the smoothness index, and the spline order, and is independent of $R$, $n$, $N$, and $\lambda$.

We choose $C_M$ large enough so that

$$C_M \geq \max\left\{2C_\star,\ 2C_{\mathrm{qi}}C_\star,\ C_\star + 1\right\}. \tag{64}$$

Set $M_R = C_M(1 + R)$. Then the choice (64) has the following consequences, uniformly for all $R \geq 1$.

First, the rescaled target satisfies

$$\sup_{[t_0,T]\times B_1} |f_R^\star| + \sup_{[t_0,T]\times B_1} \|\nabla_{\bar{x}} f_R^\star\| \leq M_R. \tag{65}$$

Equivalently, on the original cylinder $[t_0, T] \times B_{2R}$,

$$\sup_{(t,x)\in[t_0,T]\times B_{2R}} |f^\star(t,x)| \leq M_R.$$

Moreover, the choice $C_M \geq C_\star + 1$ provides the strict slack

$$M_R - \sup_{(t,x)\in[t_0,T]\times B_{2R}} |f^\star(t,x)| \geq 1. \tag{66}$$

This slack ensures that $f^\star$ lies in the interior of the clipping envelope; in particular, clipping is inactive on $f^\star$ and on any approximant that is uniformly within distance 1 of $f^\star$.

Second, the $W^{1,\infty}$-stability constant in the spline quasi-interpolation construction is absorbed into the same envelope. Namely, if $Qf_R^\star$ is the spline quasi-interpolant used in Proposition A.4, then

$$\|Qf_R^\star\|_{L^\infty([t_0,T]\times B_1)} + \|\nabla_{\bar{x}} Qf_R^\star\|_{L^\infty([t_0,T]\times B_1)} \leq M_R. \tag{67}$$

Hence the clipped approximation construction on $[t_0, T] \times B_1$ is compatible with the class $\mathcal{F}_R = \mathcal{F}_{M_R}^{(d+1)}(L, W, S, B)$ for every $R \geq 1$, without changing $C_M$.

Consequently,

$$\mathcal{F}_{2R} \subseteq \mathcal{H}_{M_R}([t_0,T]\times B_{2R}), \qquad f^\star \in \mathcal{H}_{M_R}([t_0,T]\times B_{2R}). \tag{68}$$

Indeed, if $h = \pi_R f \in \mathcal{F}_{2R}$, then

$$\|h\|_{L^\infty([t_0,T]\times B_{2R})} = \sup_{(t,x)\in[t_0,T]\times B_{2R}} \left| f\left(t, \frac{x}{2R}\right) \right| \leq M_R,$$

because $f \in \mathcal{F}_R$. The inclusion $f^\star \in \mathcal{H}_{M_R}([t_0,T]\times B_{2R})$ follows from (65).

Finally, throughout the truncated-cylinder analysis we use the logistic curvature lower bound

$$c_{\min}(R) := \frac{1}{4\cosh^2(M_R/2)}. \tag{69}$$

Since $M_R = C_M(1 + R)$, there exists a constant $C < \infty$, independent of $R$, such that

$$c_{\min}(R)^{-1} \lesssim e^{CR}. \tag{70}$$

**Lemma C.2** (Existence, uniqueness, and clipping on $[t_0, T] \times B_{2R}$). *Suppose Assumption 5.1 holds, and let $C_M$ be fixed by the global choice in Appendix C.2. For every $R \geq 1$ and every $\lambda > 0$, the population functional $J_{\lambda,2R}$ admits a unique minimizer over $\mathcal{H}_{M_R}([t_0,T]\times B_{2R})$ with respect to the finite-energy equivalence class, i.e. unique up to $dt\,dx$-a.e. equivalence. We denote it by*

$$f_{\lambda,2R} := \arg\min_{h\in\mathcal{H}_{M_R}([t_0,T]\times B_{2R})} J_{\lambda,2R}(h).$$

*Moreover,*

$$\|f_{\lambda,2R}\|_{L^\infty([t_0,T]\times B_{2R})} \leq \|f^\star\|_{L^\infty([t_0,T]\times B_{2R})} \leq M_R. \tag{71}$$

*Proof.* For the direct-method argument, the finite-energy functions are viewed in the Hilbert space $L^2([t_0, T]; H^1(B_{2R}))$, equipped with the equivalent weighted energy norm $\|\cdot\|_{L^2(I;H^1(B_{2R},\mu_t))}$ introduced above. For fixed $R$, the density $\rho_t$ is

bounded above and below by positive constants on $[t_0, T] \times B_{2R}$, so weak compactness in this weighted norm is the same as in the corresponding Lebesgue space.

The feasible set $\mathcal{H}_{M_R}([t_0, T] \times B_{2R})$ is convex. Let $\{h_m\}_{m \geq 1} \subset \mathcal{H}_{M_R}([t_0, T] \times B_{2R})$ be a minimizing sequence for $J_{\lambda,2R}$. Since $|h_m| \leq M_R$ a.e. on $[t_0, T] \times B_{2R}$, the $L^2([t_0, T] \times B_{2R})$-part is uniformly bounded. Moreover, the Sobolev penalty in $J_{\lambda,2R}$ controls $\|\nabla_x h_m\|_{L^2([t_0,T] \times B_{2R})}$. Hence $\{h_m\}$ is bounded in $\|\cdot\|_{L^2(I;H^1(B_{2R},\mu_t))}$. By weak compactness, there exists a subsequence, still denoted by $\{h_m\}$, and a finite-energy limit $h_\lambda$ such that

$$h_m \rightharpoonup h_\lambda \quad \text{weakly in the finite-energy Hilbert space.}$$

The constraint set is weakly closed. Indeed, weak convergence in the finite-energy Hilbert space implies weak convergence in $L^2([t_0, T] \times B_{2R})$, and the set $\{h \in L^2([t_0, T] \times B_{2R}) : |h| \leq M_R \text{ a.e.}\}$ is closed and convex in $L^2([t_0, T] \times B_{2R})$, hence weakly closed. Thus $h_\lambda \in \mathcal{H}_{M_R}([t_0, T] \times B_{2R})$.

The cross-entropy part is convex and weakly lower semicontinuous on the bounded logit interval $[-M_R, M_R]$, and the quadratic Sobolev penalty is weakly lower semicontinuous. Therefore

$$J_{\lambda,2R}(h_\lambda) \leq \liminf_{m \to \infty} J_{\lambda,2R}(h_m),$$

so the minimum is attained.

Uniqueness follows from the strict convexity of $J_{\lambda,2R}$ on $\mathcal{H}_{M_R}([t_0, T] \times B_{2R})$. Indeed, on the interval $[-M_R, M_R]$, the conditional logistic risk has curvature bounded below by

$$c_{\min}(R) = \frac{1}{4\cosh^2(M_R/2)} > 0.$$

Since $p_t$ and $q_t$ are Gaussian convolutions, $\rho_t > 0$ on $B_{2R}$ for every $t \in [t_0, T]$. Hence equality in the strict convexity inequality forces two minimizers to agree $dt\,dx$-a.e. on $[t_0, T] \times B_{2R}$. Thus $J_{\lambda,2R}$ has at most one minimizer. We denote the unique minimizer by $f_{\lambda,2R}$.

It remains to prove the clipping property. Let

$$M_R^\star := \|f^\star\|_{L^\infty([t_0,T] \times B_{2R})}.$$

By the global choice of $C_M$, we have $M_R^\star \leq M_R$. For any $h \in \mathcal{H}_{M_R}([t_0, T] \times B_{2R})$, define its clipped version by

$$h^{\text{clip}}(t, x) := (-M_R^\star) \vee \big(h(t,x) \wedge M_R^\star\big).$$

Then $h^{\text{clip}} \in \mathcal{H}_{M_R}([t_0, T] \times B_{2R})$. Indeed,

$$|h^{\text{clip}}| \leq M_R^\star \leq M_R,$$

and the chain rule for Lipschitz truncations gives

$$\|\nabla_x h^{\text{clip}}\|_{L^2([t_0,T] \times B_{2R})} \leq \|\nabla_x h\|_{L^2([t_0,T] \times B_{2R})}.$$

For fixed $(t, x)$, the conditional cross-entropy risk

$$G_{t,x}(u) := -\eta_t(x) \log \sigma(u) - (1 - \eta_t(x)) \log(1 - \sigma(u))$$

is convex and is uniquely minimized at $u = f^\star(t, x)$, because $\eta_t(x) = \sigma(f^\star(t, x))$. Since $|f^\star(t, x)| \leq M_R^\star$, the projection of any $u$ onto $[-M_R^\star, M_R^\star]$ cannot increase $G_{t,x}(u)$. Hence

$$L_{\text{CE},2R}(h^{\text{clip}}) \leq L_{\text{CE},2R}(h).$$

Combining this with the contraction of the spatial-gradient seminorm yields

$$J_{\lambda,2R}(h^{\text{clip}}) \leq J_{\lambda,2R}(h).$$

Applying this to $h = f_{\lambda,2R}$, we find that $f_{\lambda,2R}^{\mathrm{cl}}$ is also a minimizer. By uniqueness,

$$f_{\lambda,2R}^{\mathrm{cl}} = f_{\lambda,2R}.$$

Therefore

$$\|f_{\lambda,2R}\|_{L^\infty([t_0,T]\times B_{2R})} \leq M_R^\star = \|f^\star\|_{L^\infty([t_0,T]\times B_{2R})} \leq M_R,$$

which proves (71). $\qquad\square$

**Lemma C.3** (Bias on $[t_0,T]\times B_{2R}$). *Let $f_{\lambda,2R}$ be the unique minimizer from Lemma C.2. For any smooth $h$, define the time-dependent operator*

$$\mathcal{K}_t h(x) := \Delta_x h(x) + \nabla_x h(x) \cdot \nabla_x \log \rho_t(x).$$

*Set*

$$\Theta(R) := \int_{t_0}^T \|\mathcal{K}_t f^\star(t,\cdot)\|_{L^2(B_{2R},\mu_t)}^2 \, dt.$$

*Let the boundary remainder be*

$$\mathfrak{b}_R := 2M_R \int_{t_0}^T \int_{\partial B_{2R}} \rho_t(x)|\partial_n f^\star(t,x)| \, dS(x) \, dt.$$

*Then*

$$\|f_{\lambda,2R} - f^\star\|_{L^2([t_0,T]\times B_{2R})}^2 \leq C c_{\min}(R)^{-2}\Theta(R)\lambda^2 + C c_{\min}(R)^{-1}\lambda\mathfrak{b}_R, \tag{72}$$

*and*

$$\|\nabla_x(f_{\lambda,2R} - f^\star)\|_{L^2([t_0,T]\times B_{2R})}^2 \leq C c_{\min}(R)^{-1}\Theta(R)\lambda + C\mathfrak{b}_R. \tag{73}$$

*In particular, with*

$$\beta(R) := C c_{\min}(R)^{-2}\Theta(R), \tag{74}$$

*we have*

$$\|f_{\lambda,2R} - f^\star\|_{L^2([t_0,T]\times B_{2R})}^2 \leq \beta(R)\lambda^2 + C c_{\min}(R)^{-1}\lambda\mathfrak{b}_R, \tag{75}$$

*and*

$$\|\nabla_x(f_{\lambda,2R} - f^\star)\|_{L^2([t_0,T]\times B_{2R})}^2 \leq \beta(R)\lambda + C\mathfrak{b}_R. \tag{76}$$

*Moreover,*

$$\Theta(R) \leq C(1+R)^K, \qquad \mathfrak{b}_R \leq C(1+R)^K e^{-cR^2}, \tag{77}$$

*and therefore*

$$\beta(R) \leq C e^{CR}(1+R)^K. \tag{78}$$

*Proof.* Let

$$\delta_\lambda := f_{\lambda,2R} - f^\star.$$

Since $f_{\lambda,2R}$ minimizes $J_{\lambda,2R}$ over the convex set $\mathcal{H}_{M_R}([t_0,T]\times B_{2R})$, and since $f^\star \in \mathcal{H}_{M_R}([t_0,T]\times B_{2R})$ by Lemma C.2, convexity implies that, for every $\varepsilon \in [0,1]$,

$$f_{\lambda,2R} + \varepsilon(f^\star - f_{\lambda,2R}) \in \mathcal{H}_{M_R}([t_0,T]\times B_{2R}).$$

Therefore the one-sided directional derivative of $J_{\lambda,2R}$ at $f_{\lambda,2R}$ along the feasible direction $f^\star - f_{\lambda,2R}$ is nonnegative:

$$DJ_{\lambda,2R}(f_{\lambda,2R})[f^\star - f_{\lambda,2R}] \geq 0.$$

Equivalently,

$$\int_{t_0}^T \int_{B_{2R}} (\sigma(f_{\lambda,2R}) - \eta_t)\delta_\lambda \, d\mu_t \, dt + 2\lambda \int_{t_0}^T \int_{B_{2R}} \nabla_x f_{\lambda,2R} \cdot \nabla_x \delta_\lambda \, d\mu_t \, dt \leq 0. \tag{79}$$

The logistic curvature lower bound on $[-M_R, M_R]$ gives

$$\int_{t_0}^{T} \int_{B_{2R}} (\sigma(f_{\lambda, 2R}) - \eta_t) \delta_\lambda \, d\mu_t \, dt \geq c_{\min}(R) \|\delta_\lambda\|_{L^2([t_0, T] \times B_{2R})}^2. \tag{80}$$

Writing $f_{\lambda, 2R} = f^\star + \delta_\lambda$, we obtain from (79) and (80)

$$c_{\min}(R) \|\delta_\lambda\|_{L^2([t_0, T] \times B_{2R})}^2 + 2\lambda \|\nabla_x \delta_\lambda\|_{L^2([t_0, T] \times B_{2R})}^2 \leq -2\lambda \int_{t_0}^{T} \int_{B_{2R}} \nabla_x f^\star \cdot \nabla_x \delta_\lambda \, d\mu_t \, dt. \tag{81}$$

For each fixed $t$, Green's identity on $B_{2R}$ yields

$$-\int_{B_{2R}} \nabla_x f^\star \cdot \nabla_x \delta_\lambda \, \rho_t \, dx = \int_{B_{2R}} \mathcal{K}_t f^\star(t, \cdot) \, \delta_\lambda \, d\mu_t - \int_{\partial B_{2R}} \rho_t \delta_\lambda \partial_n f^\star \, dS. \tag{82}$$

For smooth $\delta_\lambda$, this is the usual integration-by-parts formula. For general $\delta_\lambda \in H^1(B_{2R})$, the identity follows by approximating $\delta_\lambda$ in $H^1(B_{2R})$ by smooth functions and using the trace theorem, since $f^\star$ and $\rho_t$ are smooth on $B_{2R}$. Since both $f_{\lambda, 2R}$ and $f^\star$ are bounded by $M_R$ on $[t_0, T] \times B_{2R}$, we have

$$|\delta_\lambda| \leq |f_{\lambda, 2R}| + |f^\star| \leq 2M_R.$$

The trace of $\delta_\lambda$ on $\partial B_{2R}$ satisfies the same $L^\infty$-bound, because $\delta_\lambda \in H^1(B_{2R}) \cap L^\infty(B_{2R})$ and the trace is obtained by approximation with bounded truncations. Therefore, the boundary contribution in (82) is bounded by $\mathfrak{b}_R$, and Cauchy–Schwarz gives

$$c_{\min}(R) \|\delta_\lambda\|_{L^2([t_0, T] \times B_{2R})}^2 + 2\lambda \|\nabla_x \delta_\lambda\|_{L^2([t_0, T] \times B_{2R})}^2 \leq 2\lambda \sqrt{\Theta(R)} \|\delta_\lambda\|_{L^2([t_0, T] \times B_{2R})} + 2\lambda \mathfrak{b}_R. \tag{83}$$

Applying Young's inequality,

$$2\lambda \sqrt{\Theta(R)} \|\delta_\lambda\|_{L^2([t_0, T] \times B_{2R})} \leq \frac{c_{\min}(R)}{2} \|\delta_\lambda\|_{L^2([t_0, T] \times B_{2R})}^2 + \frac{C\lambda^2}{c_{\min}(R)} \Theta(R),$$

we obtain

$$\frac{c_{\min}(R)}{2} \|\delta_\lambda\|_{L^2([t_0, T] \times B_{2R})}^2 + 2\lambda \|\nabla_x \delta_\lambda\|_{L^2([t_0, T] \times B_{2R})}^2 \leq \frac{C\lambda^2}{c_{\min}(R)} \Theta(R) + 2\lambda \mathfrak{b}_R.$$

Dropping the gradient term gives

$$\|\delta_\lambda\|_{L^2([t_0, T] \times B_{2R})}^2 \leq C c_{\min}(R)^{-2} \Theta(R) \lambda^2 + C c_{\min}(R)^{-1} \lambda \mathfrak{b}_R,$$

which proves (72). Dropping instead the $L^2$-term and dividing by $2\lambda$ gives

$$\|\nabla_x \delta_\lambda\|_{L^2([t_0, T] \times B_{2R})}^2 \leq C c_{\min}(R)^{-1} \Theta(R) \lambda + C \mathfrak{b}_R,$$

which proves (73). The beta-form bounds (75)–(76) follow immediately from the definition of $\beta(R)$.

It remains to prove the growth estimates in (77). First, we control the mixture score. Since $p_t$ and $q_t$ are the laws of $\alpha_t X_0 + \sigma_t \xi$ and $\alpha_t Y_0 + \sigma_t \xi'$, respectively, with $X_0 \sim p_0$, $Y_0 \sim q_0$, and independent standard Gaussian noises, the same Gaussian-factorization argument used in Lemma C.1 gives

$$\|\nabla_x \log p_t(x)\| + \|\nabla_x \log q_t(x)\| \leq C(1 + \|x\|),$$

uniformly over $t \in [t_0, T]$. Since

$$\nabla_x \log \rho_t(x) = \frac{p_t(x)}{p_t(x) + q_t(x)} \nabla_x \log p_t(x) + \frac{q_t(x)}{p_t(x) + q_t(x)} \nabla_x \log q_t(x),$$

we have

$$\|\nabla_x \log \rho_t(x)\| \leq C(1 + \|x\|).$$

Together with Lemma C.1, which gives polynomial growth of the derivatives of $f^\star$ and a uniform bound on $\nabla_x f^\star$, this implies

$$|(\mathcal{K}_t f^\star(t, \cdot))(x)| \leq C(1 + \|x\|)^K.$$

Therefore

$$\Theta(R) = \int_{t_0}^T \|\mathcal{K}_t f^\star(t, \cdot)\|_{L^2(B_{2R}, \mu_t)}^2 \, dt \leq C(1 + R)^K.$$

Next, we bound the boundary remainder. By Lemma C.1,

$$|\partial_n f^\star(t, x)| \leq \|\nabla_x f^\star(t, x)\| \leq C.$$

Moreover, the compact-support VP representation implies the pointwise density bound

$$\rho_t(x) \leq C \exp\{-c(\|x\| - R_1)_+^2\},$$

uniformly over $t \in [t_0, T]$. Hence, on $\partial B_{2R}$,

$$\rho_t(x)|\partial_n f^\star(t, x)| \leq C e^{-cR^2}.$$

Multiplying by the surface area of $\partial B_{2R}$, the length of the time interval, and $M_R = C_M(1 + R)$, we obtain

$$\mathfrak{b}_R \leq C(1 + R)^K e^{-cR^2}.$$

This proves (77). Finally, (78) follows from

$$\beta(R) = C c_{\min}(R)^{-2} \Theta(R), \qquad \Theta(R) \leq C(1 + R)^K,$$

and the growth bound $c_{\min}(R)^{-2} \leq e^{CR}$, which follows from $M_R = C_M(1 + R)$ and

$$c_{\min}(R) = \frac{1}{4 \cosh^2(M_R/2)}.$$

$\square$

**Lemma C.4** (Approximation by the $R$-dependent clipped class). *Suppose Assumption 5.1 holds and $s \leq 4$. For every $R \geq 1$, there exists a deterministic comparator*

$$f_{0,R} \in \mathcal{F}_{2R}$$

*such that*

$$\|f_{0,R} - f^\star\|_{L^2(I; H^1(B_{2R}, \mu_t))}^2 \leq C \mathcal{A}_s(R) N^{-\frac{2(s-1)}{d+1}}. \tag{84}$$

*Moreover,*

$$\|f_{0,R} - f^\star\|_{L^2(I; H^1(B_{2R}, \mu_t))}^2 \leq C(1 + R)^K N^{-\frac{2(s-1)}{d+1}}. \tag{85}$$

*The comparator $f_{0,R}$ is fixed before observing the data.*

*Proof.* By the choice of $C_M$ in (64), the $W^{1,\infty}$-stability constant of the spline quasi-interpolation operator is already absorbed into the envelope $M_R = C_M(1 + R)$. Hence Proposition A.4 applies on the fixed compact time-space domain $[t_0, T] \times B_1 \subset \mathbb{R}^{d+1}$. Therefore, there exists

$$f_R^{\mathrm{app}} \in \mathcal{F}_R$$

such that

$$\|f_R^{\mathrm{app}} - f_R^\star\|_{H^1(I \times B_1)}^2 \leq C_{\mathrm{app}} \|f_R^\star\|_{H^s(I \times B_1)}^2 N^{-\frac{2(s-1)}{d+1}}. \tag{86}$$

Define

$$f_{0,R} := \pi_R f_R^{\mathrm{app}} \in \mathcal{F}_{2R}.$$

Let

$$e_R := f_R^{\mathrm{app}} - f_R^\star \quad \text{on } [t_0, T] \times B_1, \qquad e := f_{0,R} - f^\star \quad \text{on } [t_0, T] \times B_{2R}.$$

Then, for $x \in B_{2R}$,

$$e(t,x) = e_R\Big(t, \frac{x}{2R}\Big), \qquad \nabla_x e(t,x) = \frac{1}{2R}\nabla_{\bar{x}} e_R\Big(t, \frac{x}{2R}\Big).$$

By the Gaussian lower-noise bound in Assumption 5.1, there exists $\rho_\star < \infty$, independent of $R$, such that

$$\sup_{t\in[t_0,T]} \sup_{x\in\mathbb{R}^d} \rho_t(x) \le \rho_\star.$$

Hence

$$\|e\|^2_{L^2(I;H^1(B_{2R},\mu_t))} = \int_{t_0}^T \int_{B_{2R}} \big(|e(t,x)|^2 + \|\nabla_x e(t,x)\|^2\big)\rho_t(x)\, dx\, dt$$

$$\le \rho_\star \int_{t_0}^T \int_{B_{2R}} \big(|e(t,x)|^2 + \|\nabla_x e(t,x)\|^2\big)\, dx\, dt.$$

Changing variables $x = 2R\bar{x}$, we get

$$\int_{B_{2R}} |e(t,x)|^2\, dx = (2R)^d \int_{B_1} |e_R(t,\bar{x})|^2\, d\bar{x},$$

and

$$\int_{B_{2R}} \|\nabla_x e(t,x)\|^2\, dx = (2R)^{d-2} \int_{B_1} \|\nabla_{\bar{x}} e_R(t,\bar{x})\|^2\, d\bar{x}.$$

Since $R \ge 1$, $(2R)^{d-2} \le (2R)^d$. Therefore,

$$\|e\|^2_{L^2(I;H^1(B_{2R},\mu_t))} \le C(2R)^d\|e_R\|^2_{L^2(I;H^1(B_1))} \le C(2R)^d\|e_R\|^2_{H^1(I\times B_1)}.$$

Combining this with (86) yields

$$\|f_{0,R} - f^\star\|^2_{L^2(I;H^1(B_{2R},\mu_t))} \le C(2R)^d\|f_R^\star\|^2_{H^s(I\times B_1)} N^{-\frac{2(s-1)}{d+1}} = C\mathcal{A}_s(R)N^{-\frac{2(s-1)}{d+1}},$$

which proves (84).

It remains to bound $\mathcal{A}_s(R)$. For every $a + |\gamma| \le s$,

$$\partial_t^a D_{\bar{x}}^\gamma f_R^\star(t,\bar{x}) = (2R)^{|\gamma|}\partial_t^a D_x^\gamma f^\star(t, 2R\bar{x}).$$

By the derivative-growth bound (62) in Lemma C.1,

$$\big|\partial_t^a D_x^\gamma f^\star(t, 2R\bar{x})\big| \le C(1+R)^K, \qquad (t,\bar{x}) \in [t_0,T] \times B_1.$$

Since $[t_0, T] \times B_1$ has finite Lebesgue volume and $|\gamma| \le s$, we obtain, after increasing $K$ if necessary,

$$\mathcal{A}_s(R) = (2R)^d\|f_R^\star\|^2_{H^s(I\times B_1)} \le C(1+R)^K.$$

Substituting this into (84) gives (85). □

**Lemma C.5** (Entropy of the pulled-back vector class). *Let $\mathcal{F}_R = \mathcal{F}_{M_R}^{(d+1)}(L,W,S,B)$ be the $R$-dependent clipped-gradient class on $[t_0, T] \times B_1$, with input dimension $d+1$, depth $L = \mathcal{O}(1)$, sparsity $S = \mathcal{O}(N)$, width $W$, and weight bound $B$. Define*

$$\mathcal{V}_{2R} := \{(h, \nabla_x h) : h = \pi_R f,\ f \in \mathcal{F}_R\},$$

*with norm*

$$\|(u,v)\|_\infty := \|u\|_{L^\infty([t_0,T]\times B_{2R})} + \|v\|_{L^\infty([t_0,T]\times B_{2R})}.$$

*Then there exists a constant $C_{\mathrm{ent}} < \infty$, independent of $R, N, n, \lambda$, such that for every $0 < \varepsilon < 1$,*

$$\log \mathcal{N}\big(\varepsilon, \mathcal{V}_{2R}, \|\cdot\|_\infty\big) \le C_{\mathrm{ent}} N \log\Big(\frac{BWM_R}{\varepsilon}\Big). \tag{87}$$

*Proof.* For $h = \pi_R f$, we have

$$h(t,x) = f\left(t, \frac{x}{2R}\right), \qquad \nabla_x h(t,x) = \frac{1}{2R}\nabla_{\bar{x}} f\left(t, \frac{x}{2R}\right).$$

Since $R \geq 1$, an $\varepsilon$-cover of

$$\{(f, \nabla_{\bar{x}} f) : f \in \mathcal{F}_R\}$$

on $[t_0, T] \times B_1$ induces an $\varepsilon$-cover of $\mathcal{V}_{2R}$ on $[t_0, T] \times B_{2R}$. Thus it suffices to control the entropy of the derivative-augmented clipped network class on $[t_0, T] \times B_1$.

The clipped-gradient class $\mathcal{F}_R$ is a subset of the clipped sparse ReLU$^3$ architecture with $\mathcal{O}(N)$ active parameters.

The same parameter-discretization argument applies to the derivative-augmented class $\{(f, \nabla_{\bar{x}} f) : f \in \mathcal{F}_R\}$, because ReLU$^3$ networks are differentiable in the input and their first input derivatives are piecewise-polynomial functions whose coefficients depend polynomially on the network parameters. The imposed bounds on the weights, the output, and the input gradient ensure that the same covering estimate holds, up to a change of constants. On each fixed sparsity pattern, the network output and its first derivatives depend Lipschitzly on the nonzero parameters on the compact domain $[t_0, T] \times B_1$. The Lipschitz constant is polynomial in $B, W$ and in the deterministic envelope $M_R$, while $L = \mathcal{O}(1)$. The number of sparsity patterns contributes an additional $\mathcal{O}(S \log W)$ term. A standard parameter-discretization argument therefore gives

$$\log \mathcal{N}\big(\varepsilon, \{(f, \nabla_{\bar{x}} f) : f \in \mathcal{F}_R\}, \|\cdot\|_\infty\big) \leq C_{\text{ent}} N \log\left(\frac{BWM_R}{\varepsilon}\right).$$

This proves (87). $\qquad\square$

**Lemma C.6** (Comparator-centered anisotropic oracle inequality on $[t_0, T] \times B_{2R}$). *Let $h_0 \in \mathcal{F}_{2R}$ be any deterministic comparator. Define the scaled truncated empirical objective*

$$\widehat{J}_{\lambda, \mathcal{D}_{\text{ext}}, 2R}(h) := \frac{T-t_0}{2n} \sum_{i=1}^n \left[\ell_{\text{CE}}(0, h(t_i^p, X_i^p)) + \lambda\|\nabla_x h(t_i^p, X_i^p)\|^2\right] \mathbf{1}\{X_i^p \in B_{2R}\}$$

$$+ \frac{T-t_0}{2n} \sum_{i=1}^n \left[\ell_{\text{CE}}(1, h(t_i^q, X_i^q)) + \lambda\|\nabla_x h(t_i^q, X_i^q)\|^2\right] \mathbf{1}\{X_i^q \in B_{2R}\}.$$

*The bracketed terms are only relevant on the events $X_i^p \in B_{2R}$ and $X_i^q \in B_{2R}$; equivalently, each $h \in \mathcal{F}_{2R}$ may be extended arbitrarily outside $[t_0, T] \times B_{2R}$, and the value of this extension does not affect the objective. Let*

$$\widehat{h}_R \in \arg\min_{h \in \mathcal{F}_{2R}} \widehat{J}_{\lambda, \mathcal{D}_{\text{ext}}, 2R}(h).$$

*Then, for every $t > 0$, with probability at least $1 - e^{-t}$,*

$$J_{\lambda, 2R}(\widehat{h}_R) - J_{\lambda, 2R}(f_{\lambda, 2R}) \leq C\left[J_{\lambda, 2R}(h_0) - J_{\lambda, 2R}(f_{\lambda, 2R}) + \Gamma_R \frac{N\log(BWM_R n) + t}{n}\right], \tag{88}$$

*where*

$$\Gamma_R := c_{\min}(R)^{-1} + M_R + \lambda\left(\frac{M_R}{2R}\right)^2. \tag{89}$$

*In particular, since $M_R = C_M(1 + R)$ and $\lambda \leq 1$,*

$$\Gamma_R \leq Ce^{CR}(1 + R)^K. \tag{90}$$

*Proof.* For measurable $\psi$, define the source–target population and empirical averages by

$$P\psi := \frac{1}{2}\mathbb{E}_{\tau \sim \text{Unif}, X \sim p_\tau}[\psi(\tau, X, 0)] + \frac{1}{2}\mathbb{E}_{\tau \sim \text{Unif}, X \sim q_\tau}[\psi(\tau, X, 1)],$$

and

$$P_n\psi := \frac{1}{2n}\sum_{i=1}^n \psi(t_i^p, X_i^p, 0) + \frac{1}{2n}\sum_{i=1}^n \psi(t_i^q, X_i^q, 1).$$

For $h \in \mathcal{F}_{2R}$, define the comparator-centered loss

$$\psi_h(\tau, x, y) := (T - t_0)\mathbf{1}\{x \in B_{2R}\} \left[ \ell_{\mathrm{CE}}(y, h(\tau, x)) - \ell_{\mathrm{CE}}(y, h_0(\tau, x)) + \lambda\left(\|\nabla_x h(\tau, x)\|^2 - \|\nabla_x h_0(\tau, x)\|^2\right) \right].$$

The expression inside the indicator is only relevant for $x \in B_{2R}$, so the value of any extension of $h$ outside $[t_0, T] \times B_{2R}$ is irrelevant. The factor $T - t_0$ is inserted so that the empirical and population objectives are normalized consistently with the unnormalized time integral in $J_{\lambda, 2R}$. Thus

$$P\psi_h = J_{\lambda, 2R}(h) - J_{\lambda, 2R}(h_0),$$

and

$$P_n\psi_h = \widehat{J}_{\lambda, \mathcal{D}_{\mathrm{ext}}, 2R}(h) - \widehat{J}_{\lambda, \mathcal{D}_{\mathrm{ext}}, 2R}(h_0).$$

The fixed factor $T - t_0$ is absorbed into constants below.

For every $h = \pi_R f \in \mathcal{F}_{2R}$,

$$\|h\|_{L^\infty([t_0,T] \times B_{2R})} \le M_R, \qquad \|\nabla_x h\|_{L^\infty([t_0,T] \times B_{2R})} \le \frac{M_R}{2R}.$$

The same bounds hold for $h_0$. Moreover, $M_R/(2R) = \mathcal{O}(1)$ for $R \ge 1$.

For $h_1, h_2 \in \mathcal{F}_{2R}$, the logistic loss is 1-Lipschitz in its logit, and

$$\left| \|a\|^2 - \|b\|^2 \right| \le (\|a\| + \|b\|)\|a - b\|.$$

Therefore, pointwise on $B_{2R}$,

$$|\psi_{h_1} - \psi_{h_2}| \le C\left( |h_1 - h_2| + \lambda\frac{M_R}{R}\|\nabla_x h_1 - \nabla_x h_2\| \right).$$

Applying the vector-valued contraction inequality separately on the $p$-sample and $q$-sample blocks and using subadditivity of the supremum,

$$\begin{aligned}
\mathfrak{R}_n\left( \left\{ \psi_h : \begin{array}{c} J_{\lambda,2R}(h) - J_{\lambda,2R}(f_{\lambda,2R}) \\ + J_{\lambda,2R}(h_0) - J_{\lambda,2R}(f_{\lambda,2R}) \le r \end{array} \right\} \right) \\
\le C\mathfrak{R}_n\left( \left\{ h - h_0 : \begin{array}{c} J_{\lambda,2R}(h) - J_{\lambda,2R}(f_{\lambda,2R}) \\ + J_{\lambda,2R}(h_0) - J_{\lambda,2R}(f_{\lambda,2R}) \le r \end{array} \right\} \right) \\
+ C\lambda\frac{M_R}{2R}\mathfrak{R}_n\left( \left\{ \nabla_x h - \nabla_x h_0 : \begin{array}{c} J_{\lambda,2R}(h) - J_{\lambda,2R}(f_{\lambda,2R}) \\ + J_{\lambda,2R}(h_0) - J_{\lambda,2R}(f_{\lambda,2R}) \le r \end{array} \right\} \right).
\end{aligned}$$

No time derivative is involved because the loss depends only on $(h, \nabla_x h)$.

The localization

$$J_{\lambda,2R}(h) - J_{\lambda,2R}(f_{\lambda,2R}) + J_{\lambda,2R}(h_0) - J_{\lambda,2R}(f_{\lambda,2R}) \le r$$

implies a localized anisotropic radius. Since $f_{\lambda,2R}$ minimizes $J_{\lambda,2R}$ over the convex set $\mathcal{H}_{M_R}([t_0,T] \times B_{2R})$, the variational inequality gives

$$DJ_{\lambda,2R}(f_{\lambda,2R})[h - f_{\lambda,2R}] \ge 0, \qquad h \in \mathcal{F}_{2R} \subseteq \mathcal{H}_{M_R}([t_0,T] \times B_{2R}).$$

Hence the strong convexity inequality on $[t_0,T] \times B_{2R}$ yields

$$J_{\lambda,2R}(h) - J_{\lambda,2R}(f_{\lambda,2R}) \ge \frac{c_{\min}(R)}{2}\|h - f_{\lambda,2R}\|^2_{L^2([t_0,T] \times B_{2R})} + \lambda\|\nabla_x(h - f_{\lambda,2R})\|^2_{L^2([t_0,T] \times B_{2R})},$$

and the same bound holds for $h_0$. Therefore

$$\|h - h_0\|^2_{L^2([t_0,T] \times B_{2R})} \le Cc_{\min}(R)^{-1}r, \qquad \lambda\|\nabla_x(h - h_0)\|^2_{L^2([t_0,T] \times B_{2R})} \le Cr.$$

Consequently,

$$\|h - h_0\|^2_{L^2([t_0,T] \times B_{2R})} + \left\| \lambda\frac{M_R}{2R}(\nabla_x h - \nabla_x h_0) \right\|^2_{L^2([t_0,T] \times B_{2R})} \le C(c_{\min}(R)^{-1} + 1)r,$$

where we used $M_R/(2R) = \mathcal{O}(1)$ and $0 < \lambda < 1$.

Applying Dudley's entropy integral to the localized subset of $\mathcal{V}_{2R}$ (Wainwright, 2019), and using Lemma C.5, gives, for $r \gtrsim n^{-2}$,

$$\mathfrak{R}_n \left( \left\{ \psi_h : \begin{array}{c} J_{\lambda,2R}(h) - J_{\lambda,2R}(f_{\lambda,2R}) \\ + J_{\lambda,2R}(h_0) - J_{\lambda,2R}(f_{\lambda,2R}) \leq r \end{array} \right\} \right) \leq \phi_R(r),$$

where

$$\phi_R(r) := C \sqrt{\frac{(c_{\min}(R)^{-1} + 1) r N \log(BW M_R n)}{n}}.$$

The function $\phi_R$ is sub-root and its critical radius satisfies

$$r_R^\star \leq C(c_{\min}(R)^{-1} + 1) \frac{N \log(BW M_R n)}{n}.$$

We next verify the boundedness and variance conditions for the normalized process. The envelope satisfies

$$|\psi_h| \leq C \left( M_R + \lambda \left( \frac{M_R}{2R} \right)^2 \right).$$

Moreover, the pointwise Lipschitz bound above gives

$$P\psi_h^2 \leq C \left[ \|h - h_0\|_{L^2([t_0,T] \times B_{2R})}^2 + \lambda^2 \left( \frac{M_R}{2R} \right)^2 \|\nabla_x(h - h_0)\|_{L^2([t_0,T] \times B_{2R})}^2 \right].$$

Using the strong-convexity localization bounds, we obtain

$$P\psi_h^2 \leq C \left( c_{\min}(R)^{-1} + \lambda \left( \frac{M_R}{2R} \right)^2 \right) [J_{\lambda,2R}(h) - J_{\lambda,2R}(f_{\lambda,2R}) + J_{\lambda,2R}(h_0) - J_{\lambda,2R}(f_{\lambda,2R})].$$

Let

$$\widetilde{\psi}_h := \frac{P\psi_h - \psi_h}{J_{\lambda,2R}(h) - J_{\lambda,2R}(f_{\lambda,2R}) + J_{\lambda,2R}(h_0) - J_{\lambda,2R}(f_{\lambda,2R}) + r}.$$

Then

$$\|\widetilde{\psi}_h\|_\infty \leq \frac{C \left( M_R + \lambda \left( \frac{M_R}{2R} \right)^2 \right)}{r}, \tag{91}$$

and

$$P\widetilde{\psi}_h^2 \leq \frac{C \left( c_{\min}(R)^{-1} + \lambda \left( \frac{M_R}{2R} \right)^2 \right)}{r}. \tag{92}$$

By the peeling lemma applied to the sub-root bound above,

$$\mathfrak{R}_n \left( \left\{ \frac{\psi_h}{J_{\lambda,2R}(h) - J_{\lambda,2R}(f_{\lambda,2R}) + J_{\lambda,2R}(h_0) - J_{\lambda,2R}(f_{\lambda,2R}) + r} : h \in \mathcal{F}_{2R} \right\} \right) \leq \frac{4\phi_R(r)}{r}.$$

Applying the Symmetrization Lemma (Lemma A.3 in (Lu et al., 2021)) to the independent $p$-sample and $q$-sample empirical processes separately, and then adding the two bounds, gives

$$\mathbb{E} \sup_{h \in \mathcal{F}_{2R}} \frac{P\psi_h - P_n\psi_h}{J_{\lambda,2R}(h) - J_{\lambda,2R}(f_{\lambda,2R}) + J_{\lambda,2R}(h_0) - J_{\lambda,2R}(f_{\lambda,2R}) + r} \leq \frac{8\phi_R(r)}{r}. \tag{93}$$

Using (93), (91), and (92), Talagrand's inequality implies that, with probability at least $1 - e^{-t}$,

$$\sup_{h \in \mathcal{F}_{2R}} \frac{P\psi_h - P_n\psi_h}{J_{\lambda,2R}(h) - J_{\lambda,2R}(f_{\lambda,2R}) + J_{\lambda,2R}(h_0) - J_{\lambda,2R}(f_{\lambda,2R}) + r} \leq \frac{16\phi_R(r)}{r}$$
$$+ C \sqrt{\frac{\left( c_{\min}(R)^{-1} + \lambda \left( \frac{M_R}{2R} \right)^2 \right) t}{nr}}$$
$$+ C \frac{\left( M_R + \lambda \left( \frac{M_R}{2R} \right)^2 \right) t}{nr}.$$

Choose

$$r_0 := C\Gamma_R \frac{N\log(BWM_R n) + t}{n}, \quad \text{with} \quad \Gamma_R = c_{\min}(R)^{-1} + M_R + \lambda\left(\frac{M_R}{2R}\right)^2,$$

with $C$ sufficiently large. Since $c_{\min}(R)^{-1} \geq 1$, this choice makes the right-hand side at most $1/2$. Hence, on the same event, for all $h \in \mathcal{F}_{2R}$,

$$(P - P_n)\psi_h \leq \frac{1}{2}\left[J_{\lambda,2R}(h) - J_{\lambda,2R}(f_{\lambda,2R})\right] + \frac{1}{2}\left[J_{\lambda,2R}(h_0) - J_{\lambda,2R}(f_{\lambda,2R})\right] + \frac{1}{2}r_0.$$

Taking $h = \widehat{h}_R$ and using empirical optimality,

$$P_n\psi_{\widehat{h}_R} = \widehat{J}_{\lambda,\mathcal{D}_{\text{ext}},2R}(\widehat{h}_R) - \widehat{J}_{\lambda,\mathcal{D}_{\text{ext}},2R}(h_0) \leq 0,$$

we obtain

$$\begin{aligned}
J_{\lambda,2R}(\widehat{h}_R) - J_{\lambda,2R}(h_0) &= P\psi_{\widehat{h}_R}\\
&= (P - P_n)\psi_{\widehat{h}_R} + P_n\psi_{\widehat{h}_R}\\
&\leq (P - P_n)\psi_{\widehat{h}_R}\\
&\leq \frac{1}{2}\left[J_{\lambda,2R}(\widehat{h}_R) - J_{\lambda,2R}(f_{\lambda,2R})\right] + \frac{1}{2}\left[J_{\lambda,2R}(h_0) - J_{\lambda,2R}(f_{\lambda,2R})\right] + \frac{1}{2}r_0.
\end{aligned}$$

Rearranging yields

$$J_{\lambda,2R}(\widehat{h}_R) - J_{\lambda,2R}(f_{\lambda,2R}) \leq 3\left[J_{\lambda,2R}(h_0) - J_{\lambda,2R}(f_{\lambda,2R})\right] + r_0.$$

This proves (88) after absorbing numerical constants into $C$.

Finally, $M_R/(2R) = \mathcal{O}(1)$, $M_R = C_M(1 + R)$, and $c_{\min}(R)^{-1} \lesssim e^{CR}$. Thus

$$\Gamma_R \leq Ce^{CR}(1 + R)^K,$$

which proves (90). $\qquad\square$

### C.3. Proof of Theorem 5.2

Recall that

$$\kappa := \frac{s - 1}{(d + 1) + 2s - 2}.$$

We decompose

$$\mathcal{E}(\widetilde{f}^{(R)}) := \int_{t_0}^T \|\widetilde{f}_t^{(R)} - f_t^\star\|_{H^1(\mu_t)}^2 \, dt = \text{Main}(R) + \text{Tail}(R), \tag{94}$$

where

$$\text{Main}(R) := \int_{t_0}^T \int_{B_R} \mathcal{L}(\widetilde{f}^{(R)}, f^\star) \, d\mu_t \, dt,$$

and

$$\text{Tail}(R) := \int_{t_0}^T \int_{B_R^c} \mathcal{L}(\widetilde{f}^{(R)}, f^\star) \, d\mu_t \, dt.$$

**Step 1: Reduction of the main error to $[t_0, T] \times B_{2R}$.** Since $\chi_R \equiv 1$ and $\text{Proj}_{2R}(x) = x$ on $B_R$, the cutoff estimator satisfies

$$\widehat{f}^{(R)} = \pi_R\widehat{f}_R \qquad \text{on } [t_0, T] \times B_R.$$

Therefore,

$$\text{Main}(R) = \int_{t_0}^T \int_{B_R} \mathcal{L}(\pi_R\widehat{f}_R, f^\star) \, d\mu_t \, dt \leq \int_{t_0}^T \int_{B_{2R}} \mathcal{L}(\pi_R\widehat{f}_R, f^\star) \, d\mu_t \, dt. \tag{95}$$

**Step 2: The truncated ERM event.** Define

$$\mathcal{A}_R := \left\{ \max_{1 \le i \le n} \|X_i^p\| \vee \max_{1 \le i \le n} \|X_i^q\| \le 2R \right\}.$$

By Lemma C.1, the marginal distributions $p_t$ and $q_t$ have uniformly sub-Gaussian tails. Therefore there exist constants $C, c > 0$, independent of $n$ and $R$, such that

$$\sup_{t \in [t_0, T]} \max \left\{ p_t(\{x : \|x\| > u\}), q_t(\{x : \|x\| > u\}) \right\} \le C \exp\left( -c(u - R_1)_+^2 \right).$$

Write compactly $t_{1:n}^p := (t_1^p, \ldots, t_n^p)$ and $t_{1:n}^q := (t_1^q, \ldots, t_n^q)$. Conditioning on these sampled times and applying a union bound over the $p$-sample and $q$-sample blocks,

$$\mathbb{P}(\mathcal{A}_R^c \mid t_{1:n}^p, t_{1:n}^q) \le \sum_{i=1}^n \mathbb{P}(\|X_i^p\| > 2R \mid t_i^p) + \sum_{i=1}^n \mathbb{P}(\|X_i^q\| > 2R \mid t_i^q) \le 2nC \exp\left( -c(2R - R_1)_+^2 \right).$$

For $R \ge R_1$, $(2R - R_1)_+ \ge R$. Hence

$$\mathbb{P}(\mathcal{A}_R^c) \le 2nC e^{-cR^2}. \tag{96}$$

On $\mathcal{A}_R$, we have $\mathrm{Proj}_{2R}(X_i^p) = X_i^p$ and $\mathrm{Proj}_{2R}(X_i^q) = X_i^q$.

$$\frac{\lambda}{4R^2} \|\nabla_{\bar{x}} f(t_i^p, X_i^p/(2R))\|^2 = \lambda \|\nabla_x(\pi_R f)(t_i^p, X_i^p)\|^2,$$

and the same identity holds for the $q$-samples. Thus, on $\mathcal{A}_R$, minimizing the projected empirical objective $\widehat{J}_{\lambda, \mathcal{D}_{\mathrm{ext}}, 2R}^{\mathrm{proj}}(f)$ over $f \in \mathcal{F}_R$ is equivalent to minimizing the truncated empirical objective over $h \in \mathcal{F}_{2R}$:

$$\begin{aligned}
\widehat{J}_{\lambda, \mathcal{D}_{\mathrm{ext}}, 2R}(h) &:= \frac{T - t_0}{2n} \sum_{i=1}^n \left[ \ell_{\mathrm{CE}}(0, h(t_i^p, X_i^p)) + \lambda \|\nabla_x h(t_i^p, X_i^p)\|^2 \right] \mathbf{1}\{X_i^p \in B_{2R}\} \\
&\quad + \frac{T - t_0}{2n} \sum_{i=1}^n \left[ \ell_{\mathrm{CE}}(1, h(t_i^q, X_i^q)) + \lambda \|\nabla_x h(t_i^q, X_i^q)\|^2 \right] \mathbf{1}\{X_i^q \in B_{2R}\}.
\end{aligned} \tag{97}$$

Consequently, on $\mathcal{A}_R$, $\pi_R \widehat{f}_R$ is an empirical minimizer of (97) over $\mathcal{F}_{2R}$.

**Step 3: Comparator-centered oracle inequality on $[t_0, T] \times B_{2R}$.** Let $f_{\lambda, 2R}$ be the population minimizer over $\mathcal{H}_{M_R}([t_0, T] \times B_{2R})$. By Lemma C.4, there exists a deterministic comparator

$$f_{0, R} \in \mathcal{F}_{2R}$$

such that

$$\|f_{0, R} - f^\star\|_{L^2(I; H^1(B_{2R}, \mu_t))}^2 \le C \mathcal{A}_s(R) N^{-\frac{2(s-1)}{d+1}} \le C(1 + R)^K N^{-\frac{2(s-1)}{d+1}}. \tag{98}$$

Applying Lemma C.6 with $h_0 = f_{0,R}$ and $t > 0$, we obtain an event $\mathcal{E}_{\mathrm{loc}}(R, t)$, with

$$\mathbb{P}(\mathcal{E}_{\mathrm{loc}}(R, t)^c) \le e^{-t},$$

on which

$$J_{\lambda, 2R}(\pi_R \widehat{f}_R) - J_{\lambda, 2R}(f_{\lambda, 2R}) \le C \left[ J_{\lambda, 2R}(f_{0, R}) - J_{\lambda, 2R}(f_{\lambda, 2R}) + \Gamma_R \frac{N \log(BW M_R n) + t}{n} \right]. \tag{99}$$

Here

$$\Gamma_R = c_{\min}(R)^{-1} + M_R + \lambda \left( \frac{M_R}{2R} \right)^2,$$

and by Lemma C.6,

$$\Gamma_R \le C e^{CR} (1 + R)^K. \tag{100}$$

**Step 4: Bound the comparator objective gap.** We bound

$$J_{\lambda,2R}(f_{0,R}) - J_{\lambda,2R}(f_{\lambda,2R})$$

by comparing both terms to $f^\star$:

$$
\begin{aligned}
J_{\lambda,2R}(f_{0,R}) - J_{\lambda,2R}(f_{\lambda,2R}) &\leq J_{\lambda,2R}(f_{0,R}) - J_{\lambda,2R}(f^\star) \\
&\quad + J_{\lambda,2R}(f^\star) - J_{\lambda,2R}(f_{\lambda,2R}).
\end{aligned}
$$

First, since $f^\star$ is the pointwise Bayes logit, it minimizes the population cross-entropy risk. The logistic loss is $1/4$-smooth in the logit; hence

$$L_{\mathrm{CE},2R}(f_{0,R}) - L_{\mathrm{CE},2R}(f^\star) \leq C\|f_{0,R} - f^\star\|^2_{L^2([t_0,T]\times B_{2R})}.$$

For the Sobolev penalty, using the pulled-back gradient envelope

$$\|\nabla_x f_{0,R}\|_{L^\infty([t_0,T]\times B_{2R})} \leq \frac{M_R}{2R} \lesssim 1$$

and the uniform spatial-gradient bound for $f^\star$ from Lemma C.1, we get

$$
\begin{aligned}
\lambda \left| \|\nabla_x f_{0,R}\|^2_{L^2([t_0,T]\times B_{2R})} - \|\nabla_x f^\star\|^2_{L^2([t_0,T]\times B_{2R})} \right| & \\
\leq C\lambda\|\nabla_x(f_{0,R} - f^\star)\|_{L^2([t_0,T]\times B_{2R})} \leq C\left[ \|f_{0,R} - f^\star\|^2_{L^2(I;H^1(B_{2R},\mu_t))} + \lambda^2 \right]. &
\end{aligned}
$$

Therefore, by (98),

$$J_{\lambda,2R}(f_{0,R}) - J_{\lambda,2R}(f^\star) \leq C\mathcal{A}_s(R)N^{-\frac{2(s-1)}{d+1}} + C\lambda^2. \tag{101}$$

Second, since $f^\star$ minimizes the cross-entropy risk,

$$L_{\mathrm{CE},2R}(f^\star) - L_{\mathrm{CE},2R}(f_{\lambda,2R}) \leq 0.$$

$$J_{\lambda,2R}(f^\star) - J_{\lambda,2R}(f_{\lambda,2R}) \leq \lambda \left( \|\nabla_x f^\star\|^2_{L^2([t_0,T]\times B_{2R})} - \|\nabla_x f_{\lambda,2R}\|^2_{L^2([t_0,T]\times B_{2R})} \right).$$

As in the proof of Lemma C.3, Green's identity gives the bound

$$J_{\lambda,2R}(f^\star) - J_{\lambda,2R}(f_{\lambda,2R}) \leq C\left[ \beta(R)\lambda^2 + c_{\min}(R)^{-1}\lambda\mathfrak{b}_R + \lambda\mathfrak{b}_R \right].$$

Combining this with (101) yields

$$
\begin{aligned}
J_{\lambda,2R}(f_{0,R}) &- J_{\lambda,2R}(f_{\lambda,2R}) \\
&\leq C\left[ \mathcal{A}_s(R)N^{-\frac{2(s-1)}{d+1}} + \lambda^2 + \beta(R)\lambda^2 + c_{\min}(R)^{-1}\lambda\mathfrak{b}_R + \lambda\mathfrak{b}_R \right].
\end{aligned}
\tag{102}
$$

**Step 5: Convert the oracle bound to $L^2(I;H^1(B_{2R},\mu_t))$-error.** The same variational inequality for the constrained minimizer $f_{\lambda,2R}$, combined with strong convexity of $J_{\lambda,2R}$ on $\mathcal{H}_{M_R}([t_0,T]\times B_{2R})$, gives, for every $f \in \mathcal{F}_{2R}$,

$$J_{\lambda,2R}(f) - J_{\lambda,2R}(f_{\lambda,2R}) \geq \frac{c_{\min}(R)}{2}\|f - f_{\lambda,2R}\|^2_{L^2([t_0,T]\times B_{2R})} + \lambda\|\nabla_x(f - f_{\lambda,2R})\|^2_{L^2([t_0,T]\times B_{2R})}.$$

Consequently,

$$\|f - f_{\lambda,2R}\|^2_{L^2(I;H^1(B_{2R},\mu_t))} \leq C(c_{\min}(R)^{-1} \vee \lambda^{-1})\left[ J_{\lambda,2R}(f) - J_{\lambda,2R}(f_{\lambda,2R}) \right].$$

For the final choice $R = A\sqrt{\log n}$ and $\lambda \asymp n^{-\kappa}$, we have

$$c_{\min}(R)^{-1} \lesssim e^{CR} = n^{o(1)},$$

so $\lambda^{-1}$ dominates $c_{\min}(R)^{-1}$ for all sufficiently large $n$. Thus, on $\mathcal{E}_{\mathrm{loc}}(R,t) \cap \mathcal{A}_R$, using (99) and (102),

$$
\begin{aligned}
\|\pi_R \widehat{f}_R &- f_{\lambda,2R}\|^2_{L^2(I;H^1(B_{2R},\mu_t))} \\
&\leq \frac{C}{\lambda}\left[ \Gamma_R \frac{N\log(BWM_R n) + t}{n} + \mathcal{A}_s(R)N^{-\frac{2(s-1)}{d+1}} + \lambda^2 + \beta(R)\lambda^2 + c_{\min}(R)^{-1}\lambda\mathfrak{b}_R + \lambda\mathfrak{b}_R \right].
\end{aligned}
$$

Using Lemma C.3 to pass from $f_{\lambda,2R}$ to $f^\star$, we obtain

$$\int_{t_0}^{T} \int_{B_{2R}} \mathcal{L}(\pi_R \widehat{f}_R, f^\star) \, d\mu_t \, dt$$
$$\leq C\left[\frac{\Gamma_R}{\lambda} \frac{N \log(BWM_R n) + t}{n} + \frac{\mathcal{A}_s(R)}{\lambda} N^{-\frac{2(s-1)}{d+1}} + \lambda + \beta(R)\lambda + (c_{\min}(R)^{-1} + 1)\mathfrak{b}_R\right]. \tag{103}$$

By (95), the same bound holds for $\mathrm{Main}(R)$.

**Step 6: Tail error.** On the annulus $B_{2R} \setminus B_R$,

$$\widetilde{f}^{(R)}(t,x) = \chi_R(x)(\pi_R \widehat{f}_R)(t,x),$$

and therefore

$$\nabla_x \widetilde{f}^{(R)}(t,x) = (\nabla_x \chi_R)(x)(\pi_R \widehat{f}_R)(t,x) + \chi_R(x)\nabla_x(\pi_R \widehat{f}_R)(t,x).$$

Since

$$|\chi_R| \leq 1, \qquad \|\nabla_x \chi_R\|_\infty \lesssim R^{-1},$$

and

$$\|\pi_R \widehat{f}_R\|_{L^\infty([t_0,T]\times B_{2R})} \leq M_R, \qquad \|\nabla_x(\pi_R \widehat{f}_R)\|_{L^\infty([t_0,T]\times B_{2R})} \leq \frac{M_R}{2R} \lesssim 1,$$

we have

$$|\widetilde{f}^{(R)}(t,x)| + \|\nabla_x \widetilde{f}^{(R)}(t,x)\| \leq C(1+R), \qquad x \in B_{2R} \setminus B_R.$$

Together with the value and gradient bounds for $f^\star$ from Lemma C.1, this gives

$$\mathcal{L}(\widetilde{f}^{(R)}, f^\star)(t,x) \leq C(1+R)^K, \qquad x \in B_{2R} \setminus B_R.$$

Therefore, using the tail bound in Lemma C.1, there exist constants $C, K, c_{\mathrm{ann}} > 0$ such that

$$\int_{t_0}^{T} \int_{B_{2R}\setminus B_R} \mathcal{L}(\widetilde{f}^{(R)}, f^\star) \, d\mu_t \, dt \leq C(1+R)^K e^{-c_{\mathrm{ann}}(R-R_1)_+^2}. \tag{104}$$

On $B_{2R}^c$, $\widetilde{f}^{(R)} = 0$. Hence

$$\mathcal{L}(\widetilde{f}^{(R)}, f^\star) = |f^\star|^2 + \|\nabla_x f^\star\|^2.$$

By Lemma C.1, this is bounded by $C(1+\|x\|)^K$. The tail-moment bound in Lemma C.1 gives constants $C, K, c_{\mathrm{out}} > 0$ such that

$$\int_{t_0}^{T} \int_{B_{2R}^c} \mathcal{L}(\widetilde{f}^{(R)}, f^\star) \, d\mu_t \, dt \leq C(1+R)^K e^{-c_{\mathrm{out}}(2R-R_1)_+^2}. \tag{105}$$

Combining (104) and (105), and taking $R \geq 2R_1$, we have

$$(R - R_1)_+ \geq R/2, \qquad (2R - R_1)_+ \geq R.$$

Thus, setting

$$c_{\mathrm{tail}} := \min\{c_{\mathrm{ann}}/4, c_{\mathrm{out}}\} > 0,$$

and enlarging $C, K$ if necessary, we obtain

$$\mathrm{Tail}(R) \leq C(1+R)^K e^{-c_{\mathrm{tail}} R^2}. \tag{106}$$

**Step 7: Choosing $N, \lambda, R$.** Choose

$$N \asymp n^{\frac{d+1}{d+1+2s-2}}, \qquad \lambda \asymp n^{-\kappa}, \qquad \kappa = \frac{s-1}{d+1+2s-2}. \tag{107}$$

Then

$$\frac{N \log(BW M_R n)}{n} \lesssim \lambda^2 \log n, \qquad N^{-\frac{2(s-1)}{d+1}} \lesssim \lambda^2. \tag{108}$$

By Lemma C.1,

$$\mathcal{A}_s(R) \le C(1+R)^K,$$

and by Lemmas C.3 and C.6,

$$\beta(R) \le C e^{CR}(1+R)^K, \qquad c_{\min}(R)^{-1} \le C e^{CR}, \qquad \Gamma_R \le C e^{CR}(1+R)^K.$$

Substituting these estimates into (103) and using (108), with $t = 2 \log n$, gives

$$\mathrm{Main}(R) \le C e^{CR}(1+R)^K (\log n) n^{-\kappa} + C e^{CR}(1+R)^K \mathfrak{b}_R. \tag{109}$$

Since

$$\mathfrak{b}_R \le C(1+R)^K e^{-cR^2},$$

the boundary term is exponentially small for the choice of $R$ below. Combining (109) with (106), we obtain

$$\mathcal{E}(\widetilde{f}^{(R)}) \le C e^{CR}(1+R)^K (\log n) n^{-\kappa} + C e^{CR}(1+R)^K e^{-cR^2}. \tag{110}$$

Finally choose

$$R = A\sqrt{\log n},$$

with $A > 0$ sufficiently large so that $R \ge 2R_1$ for all sufficiently large $n$, the truncation-event failure probability in (96) is at most $n^{-2}$, and the exponential remainder in (110) is negligible relative to the main term. On the event

$$\mathcal{E}_{\mathrm{loc}}(R, 2 \log n) \cap \mathcal{A}_R,$$

whose probability is at least $1 - 2n^{-2}$, and hence at least $1 - 3n^{-2}$, we have

$$\mathcal{E}(\widetilde{f}^{(R)}) \le C e^{C\sqrt{\log n}} (\log n)^K n^{-\kappa}. \tag{111}$$

Since

$$e^{C\sqrt{\log n}} (\log n)^K = n^{o(1)},$$

it follows that for every $\varepsilon > 0$, there exists $C_\varepsilon < \infty$ such that

$$\mathcal{E}(\widetilde{f}^{(R)}) \le C_\varepsilon n^{-\kappa+\varepsilon}. \tag{112}$$

This completes the proof. $\qquad\square$

## D. Supplementary Experiment Results

### D.1. Synthetic Experiments

We provide details for synthetic source–target pairs here.

- **Rotated ridge.** In the $(t, s)$-coordinates the source and target are mixtures of two Gaussians with identical covariance but different offsets along the $t$-axis: the source has components centered at $\pm a$ and the target at $\pm b$ (with $b > a$), while the $s$-coordinate is shared. The pair $(t, s)$ is then rotated by an angle $\mathcal{P} = 30°$ into the observed $(x_1, x_2)$-space. This creates an anisotropic "ridge" structure where $p$ and $q$ differ primarily along a rotated onedimensional direction.

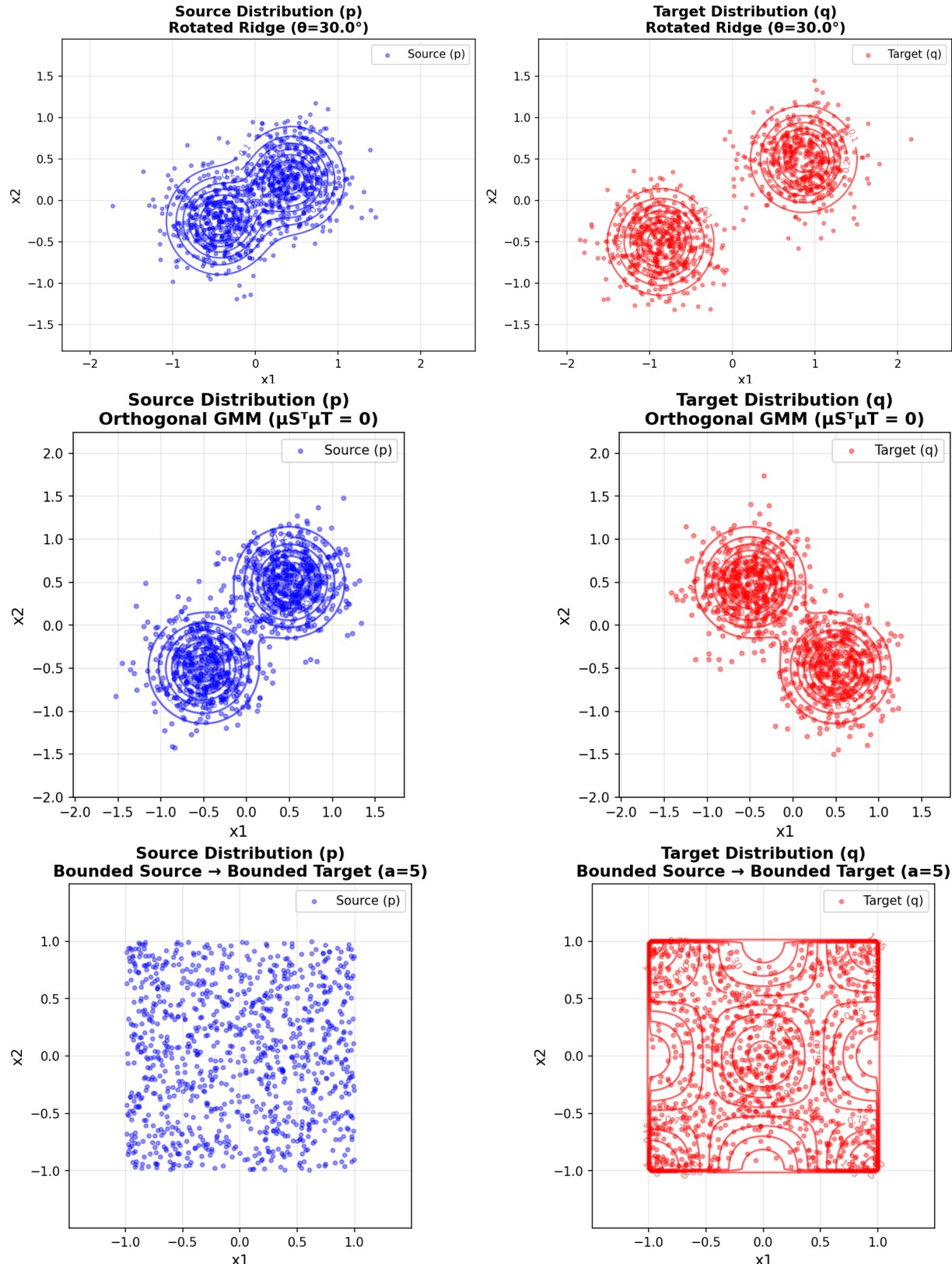

*Figure 4.* Three simulation distributions (left column: source $p$, right column: target $q$). **Top:** Same-line GMM with rotated ridge structure. **Middle:** Orthogonal GMM with misaligned source and target components. **Bottom:** Bounded source-to-target distribution with compact support.

- **Orthogonal GMM.** Both $p$ and $q$ are symmetric two-component Gaussian mixtures with spherical covariance. The source means are $\pm\mu_S$, the target means are $\pm\mu_T$, and we choose $\mu_S^\top \mu_T = 0$, i.e. the source and target mixture axes are orthogonal. This case is adopted in Ouyang et al. (2024) and probes how well the estimator adapts when the main directions of variation in the source and target are misaligned.

- **Bounded source to bounded target.** The source $p$ is the uniform distribution on the square $[-1, 1]^2$. The target $q$ is obtained by smoothly reweighting $p$ with a bounded, oscillatory density ratio $r(x) = \frac{q(x)}{p(x)} = c(1 + b\cos(\pi x_1)\cos(\pi x_2))$, where the parameters are chosen so that $r(x) \in [1/a, a]$ with $a = 5$. This yields a non-Gaussian, compactly supported setting with a spatially varying but uniformly bounded density ratio.

Together, these three examples cover rotated lowdimensional shifts, orthogonal mixture structure, and bounded non-Gaussian modulation, and hence provide a diverse testbed for gradient-of-ratio estimation.

### D.2. WGF experiments

**Setting details.** When a large number of unlabeled target samples $x_q^{(k)}$ are available, (Courty et al., 2016) proposed learning an optimal transport map to align the joint source and target distributions ($P_{XZ}$ and $Q_{XZ}$), and subsequently training a classifier on the aligned source samples. Building on this framework, (Liu et al., 2023) utilized WGF for sample alignment. Their method evolves particles $x_t$ to minimize $\mathrm{KL}[p_t, q]$ between the evolving particle-label distribution $p_t$ (initialized with source samples $\mathcal{D}_{p_t} := \{(x_t^{(i)}, y_p^{(i)})\}_{i=1}^{n_p}$) and the target density $q$. Upon convergence after $T$ iterations, a classifier is trained on the transported source samples $\{(x_T, y_q)\}$ to predict target labels.

Implementing WGF requires estimating the score difference. While (Liu et al., 2023) employed a local method known as "local linear interpolation" (LL) for this estimation, our proposed approach utilizes a classification-based method with Sobolev regularization to estimate the score difference globally.

### D.3. ECG experiments

**Details for two datasets.** PTB-XL contains 21,837 clinical 12-lead ECG recordings of 10-second duration from 18,885 patients, annotated with 71 diagnostic statements in a multi-label setting. The ICBEB2018 dataset consists of 6,877 12-lead ECG recordings with durations ranging from 6 to 60 seconds, each labeled into one of nine diagnostic classes, which form a subset of the PTB-XL label space. We randomly select $10\%$ of the ICBEB2018 samples as the limited target dataset via stratified sampling to preserve label proportions. All ECG signals are resampled to a frequency of 100 Hz.

**Synthetic quality and training speed evaluation for diffusion models.** We evaluate the quality of the generated samples of the target task using the standard Fréchet Inception Distance (FID) metric (Heusel et al., 2017). The FID score measures the Wasserstein-2 distance between the distributions of real and synthetic data within the feature space of an xresnet1d50 classifier (Strodthoff et al., 2020) pre-trained on the target task.

The quantitative results are summarized in Table 6. Our proposed TGDP-SOB not only achieves the best FID of 8.097 but also significantly reduces model parameters to 2.8M and training time to 30 minutes, offering a superior balance between generation quality and computational efficiency.

*Table 6.* The effectiveness of Soblev Penalization on ECG benchmark under synthetic quality.

| Method | FID | Parameters | Training Time |
|---|---|---|---|
| Vanilla Diffusion | 11.171 | 50.2M | 1h |
| Finetune Generator | 8.415 | 50.2M | 40min |
| TGDP | 8.100 | **2.8M** | **30min** |
| **TGDP-SOB** | **8.097** | **2.8M** | **30min** |

