# OpenReview forum: "Sobolev Regularized Score Difference Estimation in Diffusion Models"
_ICML.cc/2026/Conference — ICML 2026 regular_

### Official Review · Reviewer_Np6P · 2026-03-11

**Soundness:** 3
**Presentation:** 3
**Significance:** 3
**Originality:** 3
**Overall Recommendation:** 4
**Confidence:** 3

**Summary:**

This paper proposes a statistical estimator for the difference of two Stein's score functions based on Sobolev regularization and provides the convergence rate and a minimax lower bound depending on the dimension $d$ and the smoothness of the underlying densities $s$. Also, the experiments show that their estimator is more stable and outperforms non-regularized score difference estimators on real-world tasks.

**Compliance With Llm Reviewing Policy:**

Affirmed.

**Final Justification:**

The rebuttal addressed my concerns, but I will keep my current score after weighing the strengths and weaknesses.

**Key Questions For Authors:**

1. This paper focuses on sparse neural network functional space and use the $ReLU^3$ activation. It would be better if the authors can explain more about the motivation to choose $ReLU^3$ and I am wondering if the analysis can be extended to a general activation function.
2. This paper considers Sobolev regularization, can other regularizations be used? How about the performance of Sobolev regularization compared with other regularizations?
3. Which method does "Baseline" in Table 3. refer to?

I would raise my score if all questions have been addressed completely.

I also would like to point 2 typos:
1. Line 178 left: missing { }, should be $\eta_3(x):= $max {$x^3, 0$}
2. Line 386 left: what does "LL" mean in the citation

**Limitations:**

Yes

**Strengths And Weaknesses:**

Strengths:
1. From my perspective, the paper is technically sound.
2. This paper is clearly written and well structured.
3. The framework of the paper can be generalized to diffusion models, which is a contribution in my opinion.

Weaknesses:
1. The title of this paper is a little bit misleading: from the title, it looks like that this paper focuses on the score difference estimation in diffusion models, but only section 5 is about the generalization to diffusion models.

---

> ### Author Rebuttal · Authors · 2026-03-29
>
> ## Question 1 for the choice of $\mathrm{ReLU}^3$
> The choice of $\mathrm{ReLU}^3$ is driven by a specific technical requirement in our proof strategy. To obtain the fast $\mathcal O(1/n)$ generalization rate, rather than the slower $\mathcal O(1/\sqrt n)$ rate, we use localized Rademacher complexity, which in turn requires covering number estimates for the neural network class and its derivatives. Since these covering numbers depend exponentially on depth (for $\mathrm{ReLU}^3$, as $3^L$), keeping the depth bounded is essential. The advantage of $\mathrm{ReLU}^3$ is that it can exactly represent order-4 B-spline bases with $\mathcal O(1)$ depth, which keeps the covering number under control.
>
> By contrast, if one uses global Rademacher complexity with norm-based bounds, then standard ReLU networks of arbitrary depth could also be analyzed, but this would only yield a slower $\mathcal O(1/\sqrt n)$ rate and therefore a strictly weaker final convergence result.
>
> More broadly, the argument is not tied to $\mathrm{ReLU}^3$ itself. It applies to activations that are at least $C^2$ and can efficiently represent degree-3 polynomials, so that the relevant B-spline bases remain implementable with $\mathcal O(1)$ depth. The key requirement is therefore bounded-depth representation, not the specific activation.
>
> We will add a remark in the revised manuscript to clarify this.
>
> ## Question 2 on other regularization.
> We thank the reviewer for this thoughtful question. Our choice of Sobolev regularization is driven by the mathematical nature of the problem, namely, recovering the gradient field $\nabla \log(q/p)$, rather than merely preventing over-parameterization.  We clarify this from two perspectives.
>
> First, unlike standard regularizers such as $L_1$, $L_2$, or weight decay, which act on parameters or function values, Sobolev regularization directly controls derivatives. Thus, a method may estimate the log-density ratio reasonably well while its gradient still oscillates strongly after differentiation. By penalizing $\|\nabla f\|_{L^2(\mu)}^2$, Sobolev regularization is directly aligned with the target quantity and is therefore well suited to stabilizing score-difference estimation.
>
> Second, to the best of our knowledge, there is little work on alternative regularization schemes specifically designed for score-difference estimation, especially with theoretical guarantees. In this sense, Sobolev regularization is a natural choice: it targets the relevant $H^1$ structure and allows us to establish statistical consistency, with rate $\tilde{\mathcal O}(n^{-(s-1)/(d+2s-2)})$, while retaining the scalability of neural network classification frameworks.
>
> If the reviewer has suggestions for alternative regularization schemes, we would be happy to explore them.
>
> ## Question 3 for clarification of “ baseline”
> The "Baseline" refers to a source-only RBF SVM. Because this baseline does not perform transfer learning, it does not require estimating any "gradient," which is why that metric is marked as "N/A" in Table 3. We have updated the table caption to make this clear.
>
> ## Typo ##
> We thank the reviewer for pointing this out. The missing braces have been corrected, and “LL” has now been spelled out as “Local Linear,” following Liu et al.'s terminology.
>
> ## Weakness: suitability of the title
> We thank the reviewer for this observation. The paper is primarily motivated by diffusion-model applications, where score-difference estimation arises naturally in post-training tasks such as discriminator guidance and transfer learning. Sections 2--4 develop the core method and theory on a bounded domain to present the main ideas, including Sobolev regularization, convergence rates, and minimax lower bounds, in a clean and transparent setting, which is standard in nonparametric estimation. Section 5 then extends the analysis to diffusion models with time dependence and unbounded domains, and Section 6.2.2 evaluates the method in a diffusion-model pipeline.
>
> At the same time, the methodology is not limited to diffusion models and applies more broadly to machine learning problems involving density-ratio or score-difference estimation. We nevertheless appreciate the reviewer's suggestion and agree that the current title may overemphasize diffusion models. We would be happy to revise it to ``Sobolev-Regularized Score Difference Estimation with Applications to Diffusion Models.''

---

> > ### Author Rebuttal · Reviewer_Np6P · 2026-04-01
> >
> > Thank you for the detailed rebuttal. I will consider adjust my score.

---

> > > ### Author Response · Authors · 2026-04-05
> > >
> > > We sincerely appreciate your thoughtful response, and we are very glad to know that all of your questions have now been fully addressed!
> > >
> > > We also noted your earlier comment that you would consider raising your score if all concerns were resolved. Given your confirmation that the issues are now “Fully resolved,” we would be very grateful if your final score could reflect this updated assessment as well.
> > >
> > > Should any further questions arise at this final stage, please do not hesitate to let us know!

---

### Official Review · Reviewer_xCJS · 2026-03-12

**Soundness:** 3
**Presentation:** 3
**Significance:** 3
**Originality:** 3
**Overall Recommendation:** 5
**Confidence:** 3

**Summary:**

This paper studies estimation of the score difference between a target and a source distribution, i.e., the gradient of the log-density ratio between the two distributions. This quantity plays an important role in several modern methods such as transfer learning for diffusion models, discriminator guidance, and Wasserstein gradient-flow adaptation.

The paper argues that the common classify-then-differentiate approach can produce unstable gradients because the classifier may overfit high-frequency noise in the log-density ratio. To address this issue, the authors propose a Sobolev-regularized logistic empirical risk minimization objective, which penalizes the gradient of the estimated log-density ratio. Differentiating the resulting logit function then yields a smoother and more stable estimate of the score difference.

On the theoretical side, the paper proves a convergence rate for the Sobolev-regularized estimator in the H1 norm and derives a minimax lower bound for score-difference estimation.  The framework is also extended to diffusion-model transfer learning on time-space domains with unbounded support using a truncation and projection construction.

Empirically, the method is evaluated on synthetic density-ratio estimation problems, Office-Caltech-10 domain adaptation via Wasserstein gradient flow, and ECG diffusion-model transfer from PTB-XL to ICBEB2018. The results show improved stability in score-difference estimation on synthetic data and improved downstream performance in the diffusion transfer setting compared to non-regularized baselines.

**Compliance With Llm Reviewing Policy:**

Affirmed.

**Final Justification:**

The rebuttal has clarified my questions and the additional experiments and the extra discussion that will be added seems sufficient to me.

**Key Questions For Authors:**

please check weaknesses above

**Limitations:**

yes

**Strengths And Weaknesses:**

Strengths
--

1. The paper studies a well-motivated problem. Score-difference estimation is an important primitive in transfer learning and post-training methods for diffusion models, and the paper clearly explains why the standard classify-then-differentiate pipeline can produce unstable gradients due to high-frequency noise in the log-density ratio.

2. The proposed solution is conceptually simple and well aligned with the target quantity. By adding Sobolev regularization to the logistic objective, the method directly controls the smoothness of the gradient of the learned log-density ratio, which is exactly the quantity needed for score-difference estimation.

3. The theoretical analysis is substantial. The paper provides an upper bound for the Sobolev-regularized estimator, a minimax lower bound for score-difference estimation.

4. The extension to diffusion-model transfer learning is technically thoughtful. In particular, the projection and truncation argument used to handle time-dependent score functions on unbounded domains provides a concrete way to connect the theoretical estimator with practical diffusion-model pipelines.

5. The empirical evaluation supports the motivation of the method. The synthetic experiments clearly demonstrate the instability of the unregularized classify-then-differentiate approach and show improved score-difference estimation in low-sample regimes. The downstream experiments on Office-Caltech-10 and the ECG benchmark show consistent improvements over non-regularized baselines.


Weaknesses
---

1. There is a non-trivial gap between the upper and lower bounds for the score-difference estimation problem. The paper attributes this gap to the saturation phenomenon of single-step Tikhonov-type regularization. However, the derived upper rate still depends on the smoothness parameter s, so it is not immediately clear how the saturation mechanism manifests itself in this setting. It would be helpful if the authors could clarify whether the gap reflects a fundamental limitation of the estimator or whether the upper bound might be improvable.

2. The theoretical analysis relies on a Sobolev smoothness parameter s that determines the scaling of the function class and the regularization level in the analysis. In practice, however, the estimator is implemented using neural networks with a Sobolev penalty and the regularization parameter is chosen empirically. It would be helpful if the authors could clarify how the regularization parameter is selected in the experiments and whether the method is sensitive to this choice.

3. The empirical evaluation mainly studies low-dimensional synthetic score-difference estimation problems, while the real-data experiments evaluate downstream task performance rather than the score-difference estimation error itself. Some discussion of how the proposed estimator is expected to behave in higher-dimensional diffusion-model settings would strengthen the practical interpretation of the results.

4. Some implementation and hyperparameter details in the experimental section are not fully specified (for example, network architecture choices and certain optimization settings). Providing additional details would help improve reproducibility.

---

> ### Author Rebuttal · Authors · 2026-03-29
>
> ## Q1 on upper and lower bound gap, and saturation phenomenon
> We sincerely thank the reviewer for this insightful question. The gap between our upper bound and the minimax lower bound is not due to loose analysis, but reflects a fundamental limitation of single-step $H^1$ Tikhonov-type regularization itself.
> Specifically, two parts of our analysis are tight for the current estimator.
> 1. **The saturation of bias:**  In Lemma 3.4, applying Green's Identity to the first-order optimality condition shows that the gradient bias is proportional to $\lambda$ times an inner product involving $R_0^* = \Delta f^* + \nabla f^* \cdot \nabla \log \mu$. Since $R_0^*$ is determined by the underlying data distributions and is non-zero in general, it acts as a non-vanishing coefficient. Therefore, the squared gradient bias is intrinsically of order $\lambda$ and cannot decay faster, regardless of the smoothness level $s$.
> 2. **The variance amplification:** The strong convexity with respect to the gradient norm—which is essential for score-difference estimation— is provided entirely by the regularization term, with modulus of order $\lambda$. Consequently, converting excess risk into a gradient error bound necessarily introduces a factor $1/\lambda$, which amplifies the stochastic error.
>
> The tension between this tight $\mathcal{O}(\lambda)$ bias and the $1/\lambda$ variance amplification bottlenecks the bias-variance trade-off.
>
> In principle, higher-order Sobolev penalties, such as an $H^m$ penalty with $m>1$, could mitigate this saturation effect by weakening the $1/\lambda$ variance amplification through Gagliardo-Nirenberg interpolation and allowing faster bias decay. However, such higher-order regularization is computationally prohibitive in practice.
>
> **Why does the rate still depend on $s$?**
> The confusion is completely understandable. Unlike classical Tikhonov settings, our nonparametric analysis involves not only regularization bias and variance, but also neural network approximation error. Larger smoothness $s$ makes the target easier to approximate, allowing a smaller optimal network size, $N^\star \approx n^{d/(d+2s-2)}$. This permits a smaller choice of $\lambda$ before the saturated bias dominates. It is this additional bias--variance--approximation trade-off that preserves the $s$-dependence in the final rate.
>
> We will add a remark on this in Section 4.
>
> ## Q2 on the choice of $\lambda$ and sensitivity to the selection.
>
> We thank the reviewer for this important question.
>
> **Selection of $\lambda$.**
> In each experiment, $\lambda$ was chosen to balance the classification and Sobolev regularization terms so that both remain active during training. Concretely, we selected $\lambda$ by a coarse grid search over a small candidate set (typically 3 to 5 values), and then fixed it for all runs in that experiment. $\lambda$ was not fine-tuned separately for each dataset pair or each distribution.
>
> **Sensitivity.**
> To assess robustness directly, we performed an ablation study in the simulation setting, varying $\lambda$ over $[10^{-5},10^{-1}]$. As shown in the anonymous figure (see https://imgur.com/a/g5jyzta), Sobolev regularization consistently outperforms the unregularized baseline over a broad range of $\lambda$, typically spanning 2 to 4 orders of magnitude, and degrades only under extreme over-regularization.
>
> ## Q3 on estimation performance in high-d.
> Evaluation of score-difference error requires the ground-truth field $\nabla \log(q_t/p_t)$, which is available only in low-dim synthetic settings. In high-dim real data, this quantity is unobservable, as in prior work such as [1]. Thus, we follow standard practice: synthetic experiments validate against ground truth, while real-data experiments assess downstream performance.
>
> We also expect Sobolev regularization to be more useful in higher dimensions, where the ``classify-then-differentiate'' failure mode is more severe: classifiers can more easily fit high-frequency artifacts, and differentiation amplifies them. Since our regularization acts directly on the gradient field, it targets this instability at its source. We will add a brief discussion in the revised manuscript.
>
> ## Q4 for experiment details
> We thank the reviewer for emphasizing reproducibility. The unspecified details follow established implementations from prior work.
>
> Specifically, Experiment 2 follows [2], including DECAF6 features with PCA to 100 dimensions, WGF with 5 epochs and step size 0.01, and SVM evaluation with the median heuristic for kernel bandwidth. Experiment 3 follows the full SSSD-ECG setup of [1], including the diffusion schedule ($T=200$, $\beta_0=0.0001$, $\beta_T=0.02$), the WaveNet architecture (36 residual layers, 256 channels with S4), and the training protocol (learning rate $2\times 10^{-4}$, batch size 8).
>
> We include these settings here for reference and will document them in the revised manuscript.
>
>
> ***
>
> **Refs:** [1] Ouyang et al. NeurIPS '24. [2] Liu et al. ICML ’24

---

> > ### Author Rebuttal · Reviewer_xCJS · 2026-04-02
> >
> > Thanks for the detailed response and additional experiments. I will raise my score.

---

> > > ### Author Response · Authors · 2026-04-05
> > >
> > > We sincerely thank the Reviewer for the improved score and for the constructive feedback throughout the discussion period！

---

### Official Review · Reviewer_wQ9E · 2026-03-13

**Soundness:** 3
**Presentation:** 3
**Significance:** 2
**Originality:** 3
**Overall Recommendation:** 4
**Confidence:** 3

**Summary:**

This paper addresses the fundamental problem of estimating the difference between two Stein's score functions ($\nabla \log q - \nabla \log p$), a task that is central to transfer learning and post-training methods in diffusion models, such as discriminator guidance. The authors identify that standard classification-based density ratio estimators often suffer from high-frequency noise when differentiated. To resolve this, they propose incorporating a Sobolev penalty into the training objective, which regularizes the $L^2$ norm of the gradient of the log-density ratio. They establish a statistical convergence rate of $\tilde{\mathcal{O}}(n^{-\frac{s-1}{d+2s-2}})$ and a minimax lower bound of $\tilde{\Omega}(n^{-\frac{2(s-1)}{d+2s}})$.

**Compliance With Llm Reviewing Policy:**

Affirmed.

**Final Justification:**

My concerns have been resolved. I believe my initial assessment accurately reflects my evaluation of this paper and I will maintain my score.

**Key Questions For Authors:**

1. Given that recent literature in score-based models (e.g., [1, 2]) has made strides in removing density lower-bound assumptions, could the authors comment on whether alternative analytical techniques could bypass the strict positivity assumption?

2. The Sobolev penalty explicitly enforces smoothness. How does the estimator perform when the true score difference $\nabla \log(q/p)$ is intrinsically high-frequency or contains sharp discontinuities (e.g., highly disjoint target distributions)? Is there a risk of catastrophic underfitting in these scenarios?

[1] K. Zhang, H. Yin, F. Liang, and J. Liu, "Minimax optimality of score-based diffusion models: Beyond the density lower bound assumptions", ICML 2024.

[2] G. Fu and W. Lee "Approximation and generalization abilities of score-based neural network generative models for sub-Gaussian distributions" NeurIPS 2025

**Limitations:**

Yes.

**Strengths And Weaknesses:**

## Strengths

- The paper is technically sound and well-structured.

- The paper has established a statistical convergence rate of $\tilde{\mathcal{O}}(n^{-\frac{s-1}{d+2s-2}})$ on bounded domains and derive a minimax lower bound of $\tilde{\Omega}(n^{-\frac{2(s-1)}{d+2s}})$.


## Weaknesses

- The analysis on bounded domains relies heavily on Assumption 2.1, which requires the source and target densities to be strictly positive and to satisfy a Neumann boundary condition ($\rho(x)\partial_n f^*(x) = 0$). These assumptions are mathematically convenient but practically restrictive.

- There remains a noticeable gap between the achieved upper bound $\tilde{\mathcal{O}}(n^{-\frac{s-1}{d+2s-2}})$ and the minimax lower bound $\tilde{\Omega}(n^{-\frac{2(s-1)}{d+2s}})$.

- The proposed Sobolev penalty is $\lambda ||\nabla f(x)||_2^2$. This penalizes the magnitude of the gradient of the estimated log-density ratio. While this prevents overfitting to high-frequency noise, it also encourages the estimated score difference to be small everywhere. If the true score difference is large (i.e., the optimal transport path between $p$ and $q$ requires strong driving forces ), the penalty will bias the estimator toward zero. This is fundamentally different from score matching, which penalizes the distance to the true score. The authors acknowledge this bias, but the practical impact of shrinking the estimated score difference towards zero, especially in the context of diffusion model guidance, is not fully explored.

- Computing the Sobolev penalty $\lambda \frac{1}{2n} \sum_{j=1}^{2n} ||\nabla f(X_j)||_2^2$  requires calculating the gradient of the neural network output with respect to its inputs during training. This necessitates a double backward pass (or Hessian-vector products) during optimization, which significantly increases the computational cost and memory footprint of training compared to standard classification.

---

> ### Author Rebuttal · Authors · 2026-03-29
>
> ## Question 1 on removing strict positive Assm.
> We thank the reviewer for the constructive suggestion. While the truncation techniques in [1, 2] do not trivially apply to the general score-difference setting, they can be adapted under an **"identical supports"** condition, generalizing our strict positivity assumption.
>
> **1. Structural breakdown in the general case**
> The truncation-based estimator in [1, 2] works by setting the score to zero in low-density tails. This relies on the error metric $\mathbb{E}_{p_t}[|\hat{s}_t - s_t|^2]$ being weighted by $p_t$ itself, ensuring the exponential decay of $p_t$ dominates the diverging score.
>
> However, in score-difference estimation, this breaks down. When $p(x)\to 0$ but $q(x)>0$, the target $f^\star=\log(q/p)\to +\infty$ and $\nabla f^\star$ diverges. Crucially, our error metric $\|\nabla f^\star-\nabla \hat{f}\|_{L^2(\mu)}^2$ is weighted by the mixture $\mu=(p+q)/2\ge q/2>0$, assigning strictly positive mass to these divergent regions. Naive truncation would lead to a non-integrable error.
>
> **2. The "identical supports" extension**
> However, if $p$ and $q$ share **identical supports** , the mixture weight $\mu$ safely vanishes alongside the target. Under this condition, the truncation arguments from [1, 2] can be successfully adapted. Indeed, this identical support condition is a generalization of our original assumption, which is simply the case when the shared support is the whole region.
>
> Inspired by your insightful suggestion, we will incorporate this generalized condition into our revised manuscript, explicitly demonstrating how our framework can use the techniques from [1, 2] to bypass the global strict positivity requirement.
>
>
>
> ## Question 2 on high-frequency ground truth scenario
> We thank the reviewer for this insightful question. Although the Sobolev penalty induces shrinkage and enforces smoothness, catastrophic underfitting is unlikely in our setting for three reasons.
>
> **1. Structural regime of transfer learning.**
> Our method is designed for transfer learning, where the source $p$ and target $q$ are assumed to share meaningful structure. If the true score difference were highly oscillatory or discontinuous, then  $p$ and target $q$ may not share similarity and little transferable information would remain. In that regime, transfer learning itself becomes meaningless and it’s better to learn $q$ from scratch.
>
> **2. Smoothing effect in diffusion models.**
> In our primary application to diffusion models, such high-frequency behavior is not expected. For any $t>0$, the marginals $p_t$ and $q_t$ are Gaussian convolutions of the initial distributions and are therefore immediately smoothed. Consequently, the true score difference $\nabla \log q_t-\nabla \log p_t$ is naturally smooth, so Sobolev regularization matches the intrinsic regularity of the problem rather than imposing an artificial bias.
>
> **3. Empirical robustness.**
> Empirically, even when the true score difference exhibits sharp spatial spikes and steep gradients (e.g., our Bounded source to bounded target experiment), our method avoids catastrophic underfitting. Instead, it reduces the gradient estimation error by 52.86% compared to the collapsing unregularized baseline in the low-sample regime, proving that variance reduction outweighs shrinkage bias.
>
> ## Weakness 4 on double backward pass
> We thank the reviewer for this important point. We agree that computing the Sobolev penalty via automatic differentiation requires a double backward pass.
>
> In our experiments, however, this overhead was acceptable and did not create a practical bottleneck. Moreover, when needed, it can be reduced using a standard finite-difference approximation: the gradient penalty is estimated from the change in the network output under a small random perturbation,
> $$
> \mathbb{E}_{z}\left[\left\|\frac{f(x+\varepsilon z)-f(x)}{\varepsilon}\right\|^2\right],
> $$
> which approximates the squared gradient norm using only one additional forward pass and avoids the double backward pass.
> This approach has been validated in the literature and performs well even in large-scale, high-dimensional settings (e.g., [3]). We will clarify in the revised manuscript that the exact Sobolev penalty is computationally manageable in our setting, and that finite-difference approximations provide a practical alternative in larger-scale applications.
>
>
> ## Weakness 2 on upper and lower bound gap.
> We thank the reviewer for raising this point. For a detailed discussion regarding the theoretical gap and the saturation phenomenon, please refer to our comprehensive responses to Reviewer Zz4G (Weakness 1) and Reviewer xCJS (Question 1).
>
> ***
>  [3] Lin et al. Diffusion Adversarial Post-Training for One-Step Video Generation. '25.

---

> > ### Author Rebuttal · Reviewer_wQ9E · 2026-04-04
> >
> > Thank you for the clear and thorough response. I believe my initial assessment accurately reflects my evaluation of this paper and I will maintain my score.

---

> > > ### Author Response · Authors · 2026-04-05
> > >
> > > We sincerely appreciate your response and are glad to hear that our rebuttal adequately addressed your concerns!

---

### Official Review · Reviewer_Zz4G · 2026-03-14

**Soundness:** 2
**Presentation:** 2
**Significance:** 2
**Originality:** 3
**Overall Recommendation:** 3
**Confidence:** 3

**Summary:**

This paper tackles score difference estimation, i.e. estimating nabla log q - nabla log p for p,q two probability distributions.

Considering the log density ratio f*=log p/q, they assume f* lives in Hs (sobolev space of smoothness s for s>=2), their goal is to learn nabla f* given access to finite samples of p and q. They see it as the minimizer of a cross entropy risk.
The function class is defined as a set of neural networks with some fixed/max number of layers, width, sparsity, norm constraint.
Their method estimates the log density ratio from samples by minimizing an empirical version of the cross entropy risk, regularized by a Sobolev seminar, over this class. They prove a rate of O(n^{ - (s-1)/(d+2s-2)} in Th 3.1 (which relies on three intermediate results, Lemma 3.,3.4 and Th 3.5). Then they prove a lower bound of order n^{-2(s-1)/(2s+d)}.
In Section 5 they extend their analysis to the diffusion model case, by estimating log pt/qt where pt and qt are time marginals of two diffusion processes with different probability distribution initialization.
Then, they test the performance of their estimator in different settings. The first one is a toy one where they know the true density ratio, and evaluate the MSE of the estimator against the ground truth. The second one uses the estimator along a WGF (involving nabla log pt/qt) to transport labelled samples from a source to a target distribution. The last one is a transfer learning task with diffusion model.

**Compliance With Llm Reviewing Policy:**

Affirmed.

**Key Questions For Authors:**

Questions:
- can the authors comment on their theoretical result, more specifically on the dependence of the class wrt n and s?
- could the author compare in the simulation environment the performance of a Kernel density estimator?

**Limitations:**

This paper does not clearly discusses its limitations.

**Strengths And Weaknesses:**

Strengths
- the results are novel and tackle a relevant problem
- their estimator seem to beat the kernel based estimator of Liu et al 2023 in the WGF setting, involving the estimation of nabla log pt\qt,. Similarly in the transfer learning with diffusion model experiment, they beat the competitor Ouyang et al 2024. Overall the experimental results are promising.

Weaknesses
- as acknowledged by the authors, the upper and lower bound rates do not match
- the class  of Th 3.1 grows as n goes to infinity, so the rate does not hold for a fixed estimator class. It also depends on the knowledge of the smoothness s, which is unknown to the user.
- the estimator of section 5 depends on a radius R that has to be chosen by the user (containing the support of the initial distribution of the diffusion processes, p and q, assuming they have compact support)

Minor comments
Soblevl 168 and Th 3.1  ->Sobolev

Overall, I think the problem tackled by the paper is interesting and the ideas are novel. However I am not fully convinced by the sharpness of the theoretical results, hence in the current state I am not sure that the paper meets ICML standards.

---

> ### Author Rebuttal · Authors · 2026-03-29
>
> ## Question 1 on dependence of s and n.
> We thank the reviewer for this observation. The reviewer is correct that the network architecture grows w.r.t. $n$ and depends on $s$; however, this is a standard feature of nonparametric estimation with deep neural networks, not a limitation specific to our analysis. We offer the following clarifications:
>
> **a) Why the function class must grown w.r.t. n**
> Allowing the hypothesis class to grow with $n$ is standard and necessary in
> nonparametric estimation. A fixed network architecture generally yields a non-vanishing approximation error, preventing consistency. Increasing the network size with $n$ (the sieve method) is canonical to balance approximation and estimation errors and achieve optimal rates [1, 2, 3].
>
> **b) Dependence on the unknown smoothness s**
> The optimal architecture depends on the smoothness parameter $s$, which is typically unknown; this is standard in minimax theory. In practice, the architecture and regularization are chosen by data-driven procedures such as cross-validation, which can adapt effectively to unknown smoothness.
>
> We hope this clarifies that our theoretical construction is both necessary for nonparametric consistency and fully aligned with the canonical neural network theory literature.
>
> ## Question 2 on KDE performance.
> We thank the reviewer for this great suggestion! We followed the suggestion and did the experiment with KDE and found interesting results. Results are in https://imgur.com/a/2cHi9qn and we include them in the revised manuscript.
>
> Our observations are the following:
>
> **Sample-size dependence and bias–variance tradeoff**
> In the extremely small-sample regime ($n=20$), kernel estimators can occasionally outperform deep learning methods because their higher bias but lower variance makes them more stable when data are scarce. Neural estimators, even with Sobolev regularization, may have higher variance in this regime. However, such low-data settings are uncommon in modern ML.
>
> For moderate to large sample sizes ($n=200,2000$), our Sobolev-regularized estimator consistently outperforms the kernel baseline. The reason is that kernel methods are ultimately limited by intrinsic bias and cannot capture global structure, while Sobolev regularization controls the variance of the neural estimator and allows it to better exploit its expressivity.
>
>
> **Computation**
> Kernel methods are also substantially more expensive, since each evaluation depends on the full dataset. In contrast, our approach amortizes this cost: once trained, score differences are computed by a single forward/backward pass of a fixed network; suitable for large-scale tasks.
>
> ## Weakness 1 on upper and lower bound gap.
> We acknowledge the gap between our upper and lower bounds. However, obtaining matching rates for derivative recovery from function-value observations with scalable methods is a well-known challenge. Even in classical Sobolev regression, single-step Tikhonov-type methods typically exhibit such gaps [4,5], while closing them generally requires multi-step iterative procedures, such as iterated Tikhonov or Lepski-type methods, which are computationally prohibitive in deep learning settings.
>
> Our goal is therefore to balance theoretical rigor with practical scalability, rather than pursue statistical optimality alone. Specifically, we develop a method that (a) provides an explicit convergence rate for score-difference estimation, unlike prior methods such as [6,7], and (b) remains scalable and performs well in practice. Empirically, the gap does not appear to impair performance: across all experiments, our estimator consistently outperforms the baselines (Tables 1, 2, and 4).
>
> We agree that closing this gap through computationally tractable iterative schemes is an interesting direction for future work. However, this does not diminish the value of our current contribution as a method that is both theoretically grounded and practically effective. More discussion, particularly on the saturation phenomenon, is provided in our response to Reviewer xCJS(question 1).
>
> ## Weakness 3 on dependence of radius $R$
> We thank the reviewer for the question. The radius $R$ is introduced only for technical control of tail behavior on $\mathbb{R}^d$; such truncation or compact-support assumptions are standard in diffusion model theory (e.g., [8, 9]).
>
> In practice, $R$ need not be specified. Real-world data are effectively bounded, and training on finite samples already induces an implicit bounded support, so no explicit tuning of $R$ is needed.
>
> ## Few typos
> Thank you and corrected.
>
> ***
>
> **Refs:** [1] Schmidt-Hieber. Ann. Stat. '20. [2] Farrell et al. Econometrica '21. [3] Lu et al. arXiv:2110.06897 '21. [4] Bauer et al. J. Complexity '07. [5] Engl et al. Springer '96. [6] Ouyang et al. NeurIPS '24. [7] Kim et al. arXiv:2211.17091 '22. [8] Han et al. arXiv:2401.15604 '24. [9] Chen et al. ICLR '23.

---

> > ### Author Rebuttal · Reviewer_Zz4G · 2026-04-03
> >
> > I sincerlu thank the authors for their response. Yet I am not fully convinced by several of the answers (e.g. on how KDE baseline was tuned, or some theoretial assumptions), so I would like to keep my score.

---

> > > ### Author Response · Authors · 2026-04-05
> > >
> > > Thank you again for your careful reading and for raising these points. We would like to take this opportunity to further clarify the two main concerns, which we do not believe “are not easily addressed in a short rebuttal.”
> > >
> > > **(1) KDE baseline and NO hyperparameter tuning.**
> > >
> > > For the KDE-based method, we follow EXACTLY the implementation and hyperparameter choices in [1]. In particular:
> > >
> > > 1. The **bandwidth selection** is determined via the standard median heuristic and does not involve tunable parameters.
> > >
> > > 2. The **kernel matrices** are constructed using the Gaussian RBF kernel.
> > >
> > > 3. The method parameterizes a local linear model at each evaluation point $x_i$, learning $(W_i, b_i)$ where $W_i\in \mathbb{R}^d$ represents the gradient estimate.
> > >
> > > 4. The **optimization objective** is the variational form of the KL divergence, as described in [1].
> > >
> > > 5. The **optimizer settings** (AdaGrad, learning rate $10^{-1}$; with other hyperparameters at their PyTorch defaults) are taken unchanged from [1].
> > >
> > > This is consistent with what we used in Experiment 2 (see our responses to reviewer xCJS Q4). Therefore, there is effectively **NO additional tuning beyond what is prescribed in the original reference**, and we have adhered to this protocol consistently across experiments.
> > >
> > > **(2) On the theoretical assumptions.**
> > >
> > >  We understand the reviewer’s concern regarding the dependence of the function class on $n$ and $s$. We would like to emphasize that such scaling is **standard in nonparametric estimation theory**, where the function class **must** grow with sample size to control the bias–variance tradeoff. This is precisely the regime studied in the literature we cited in the rebuttal.
> > >
> > > For comparison, restricting to a fixed-capacity (parametric) class—such as a neural network with fixed width and depth—generally leads to a **non-vanishing approximation error**, which is not suitable for the consistency guarantees targeted in our results.
> > >
> > > We therefore believe our assumptions are well-aligned with existing literature to achieve the optimal approximation rate, and we hope the cited work can also provide helpful context and clarification on this point.
> > >
> > > We sincerely hope the above addresses your concerns. Please let us know if there is any aspect we could clarify further.

---

### Decision · Program_Chairs · 2026-04-30

**Decision:**

Accept (regular)

**Comment:**

The paper proposes a Sobolev-regularized estimator for the score difference between two distributions,
 addressing instability.  By penalizing gradients of the estimated log-density ratio, the method yields smoother score estimates. The paper also provides some theoretical guarantees on the  estimator.

 Most reviewers found that the paper brings in original contributions and novel ideas. While one reviewer raises some relevant concerns
I believe that the strenghts of the paper outweight its weaknesses and as such, the paper could be accepted.